# Eigencurve: Optimal Learning Rate Schedule for SGD on Quadratic Objectives with Skewed Hessian Spectrums

**Rui Pan**[1*], **Haishan Ye**[2 *†], **Tong Zhang**[1 *‡]
[1]The Hong Kong University of Science and Technology
[2]Xi'an Jiaotong University
`rpan@connect.ust.hk, yehaishan@xjtu.edu.cn`

## Abstract

Learning rate schedulers have been widely adopted in training deep networks. Despite their practical importance, there is a discrepancy between its practice and its theoretical analysis. For instance, it is not known what schedules of SGD achieve best convergence, even for simple problems such as optimizing quadratic objectives. In this paper, we propose Eigencurve, the first family of learning rate schedules that can achieve minimax optimal convergence rates (up to a constant) for SGD on quadratic objectives when the eigenvalue distribution of the underlying Hessian matrix is skewed. The condition is quite common in practice. Experimental results show that Eigencurve can significantly outperform step decay in image classification tasks on CIFAR-10, especially when the number of epochs is small. Moreover, the theory inspires two simple learning rate schedulers for practical applications that can approximate eigencurve. For some problems, the optimal shape of the proposed schedulers resembles that of cosine decay, which sheds light to the success of cosine decay for such situations. For other situations, the proposed schedulers are superior to cosine decay.

## 1 Introduction

Many machine learning models can be represented as the following optimization problem:

$$\min_w f(w) \triangleq \frac{1}{n} \sum_{i=1}^n f_i(w), \tag{1.1}$$

such as logistic regression, deep neural networks. To solve the above problem, stochastic gradient descent (SGD) (Robbins & Monro, 1951) has been widely adopted due to its computation efficiency in large-scale learning problems (Bottou & Bousquet, 2008), especially for training deep neural networks.

Given the popularity of SGD in this field, different learning rate schedules have been proposed to further improve its convergence rates. Among them, the most famous and widely used ones are inverse time decay, step decay (Goffin, 1977), and cosine scheduler (Loshchilov & Hutter, 2017). The learning rates generated by the inverse time decay scheduler depends on the current iteration number inversely. Such a scheduling strategy comes from the theory of SGD on strongly convex functions, and is extended to non-convex objectives like neural networks while still achieving good performance. Step decay scheduler keeps the learning rate piecewise constant and decreases it by a factor after a given amount of epochs. It is theoretically proved in Ge et al. (2019) that when the objective is quadratic, step decay scheduler outperforms inverse time decay. Empirical results are also provided in the same work to demonstrate the better convergence property of step decay in training neural networks when compared with inverse time decay. However, even step decay

---

[*]Equal contribution.
[†]Corresponding author is Haishan Ye.
[‡]Jointly with Google Research.

is proved to be near optimal on quadratic objectives, it is *not* truly optimal. There still exists a $\log T$ gap away from the minimax optimal convergence rate, which turns out to be non-trivial in a wide range of settings and may greatly impact step decay's empirical performance. Cosine decay scheduler (Loshchilov & Hutter, 2017) generates cosine-like learning rates in the range $[0, T]$, with $T$ being the maximum iteration. It is a heuristic scheduling strategy which relies on the observation that good performance in practice can be achieved via slowly decreasing the learning rate in the beginning and "refining" the solution in the end with a very small learning rate. Its convergence property on smooth non-convex functions has been shown in Li et al. (2021), but the provided bound is still not tight enough to explain its success in practice.

Except cosine decay scheduler, all aforementioned learning rate schedulers have (or will have) a tight convergence bound on quadratic objectives. In fact, studying their convergence property on quadratic objective functions is quite important for understanding their behaviors in general non-convex problems. Recent studies in Neural Tangent Kernel (NTK) (Arora et al., 2019; Jacot et al., 2020) suggest that when neural networks are sufficiently wide, the gradient descent dynamic of neural networks can be approximated by NTK. In particular, when the loss function is least-square loss, neural network's inference is equivalent to kernel ridge regression with respect to the NTK in expectation. In other words, for regression tasks, the non-convex objective in neural networks resembles quadratic objectives when the network is wide enough.

The existence of $\log T$ gap in step decay's convergence upper bound, which will be proven to be tight in a wide range of settings, implies that there is still room for improvement in theory. Meanwhile, the existence of cosine decay scheduler, which has no strong theoretical convergence guarantees but possesses good empirical performance in certain tasks, suggests that its convergence rate may depend on some specific properties of the objective determined by the network and dataset in practice. Hence it is natural to ask what those key properties may be, and whether it is possible to find theoretically-optimal schedulers whose empirical performance are comparable to cosine decay if those properties are available. In this paper, we offer an answer to these questions. We first derive a novel eigenvalue-distribution-based learning rate scheduler called `eigencurve` for quadratic functions. Combining with eigenvalue distributions of different types of networks, new neural-network-based learning rate schedulers can be generated based on our proposed paradigm, which achieve better convergence properties than step decay in Ge et al. (2019). Specifically, `eigencurve` closes the $\log T$ gap in step decay and reaches minimax optimal convergence rates if the Hessian spectrum is skewed. We summarize the main contributions of this paper as follows.

1. To the best of our knowledge, this is the first work that incorporates the eigenvalue distribution of objective function's Hessian matrix into designing learning rate schedulers. Accordingly, based on the eigenvalue distribution of the Hessian, we propose a novel eigenvalue distribution based learning rate scheduler named `eigencurve`.

2. Theoretically, `eigencurve` can achieve optimal convergence rate (up to a constant) for SGD on quadratic objectives when the eigenvalue distribution of the Hessian is skewed. Furthermore, even when the Hessian is not skewed, `eigencurve` can still achieve no worse convergence rate than the step decay schedule in Ge et al. (2019), whose convergence rate are proven to be sub-optimal in a wide range of settings.

3. Empirically, on image classification tasks, `eigencurve` achieves optimal convergence rate for several models on CIFAR-10 and ImageNet if the loss can be approximated by quadratic objectives. Moreover, it obtains much better performance than step decay on CIFAR-10, especially when the number of epochs is small.

4. Intuitively, our learning rate scheduler sheds light on the theoretical property of cosine decay and provides a perspective of understanding the reason why it can achieve good performance on image classification tasks. The same idea has been used to inspire and discover several simple families of schedules that works in practice.

**Problem Setup** For the theoretical analysis and the aim to derive our eigenvalue-dependent learning rate schedulers, we mainly focus on the quadratic function, that is,

$$\min_w f(w) \triangleq \mathbb{E}_\xi \left[ f(w, \xi) \right], \text{ where } f(w, \xi) = \frac{1}{2} w^\top H(\xi) w - b(\xi)^\top w, \qquad (1.2)$$

where $\xi$ denotes the data sample. Hence, the Hessian of $f(w)$ is

$$H = \mathbb{E}_\xi \left[ H(\xi) \right]. \qquad (1.3)$$

Letting us denote $b = \mathbb{E}_\xi[b(\xi)]$, we can obtain the optima of problem (1.2)

$$w_* = H^{-1}b. \tag{1.4}$$

Given an initial iterate $w_0$ and the learning rate sequence $\{\eta_t\}$, the stochastic gradient update is

$$w_{t+1} = w_t - \eta_t \nabla f(w_t, \xi) = w_t - \eta_t(H(\xi)w_t - b(\xi)). \tag{1.5}$$

We denote that

$$n_t = Hw_t - b - (H(\xi)w_t - b(\xi)), \quad \mu \triangleq \lambda_{\min}(H), \quad L \triangleq \lambda_{\max}(H), \quad \text{and,} \quad \kappa \triangleq L/\mu. \tag{1.6}$$

In this paper, we assume that

$$\mathbb{E}_\xi\left[n_t n_t^\top\right] \preceq \sigma^2 H. \tag{1.7}$$

The reason for this assumption is presented in Appendix G.5.

**Related Work** In convergence analysis, one key property that separates SGD from vanilla gradient descent is that in SGD, noise in gradients dominates. In gradient descent (GD), constant learning rate can achieve linear convergence $\mathcal{O}(c^T)$ with $0 < c < 1$ for strongly convex objectives, i.e. obtaining $f(w^{(t)}) - f(w^*) \le \epsilon$ in $\mathcal{O}(\log(\frac{1}{\epsilon}))$ iterations. However, in SGD, $f(w^{(t)})$ cannot even be guaranteed to converge to $f(w^*)$ due to the existence of gradient noise (Bottou et al., 2018). Intuitively, this noise leads to a variance proportional to the learning rate size, so constant learning rate will always introduce a $\Omega(\eta_t) = \Omega(\eta_0)$ gap when compared with the convergence rate of GD.

Table 1: Convergence rate of SGD with common schedulers on quadratic objectives.

| Scheduler | Convergence rate of SGD in quadratic objectives |
|---|---|
| Constant | Not guaranteed to converge |
| Inverse Time Decay | $\Theta\left(\frac{d\sigma^2}{T} \cdot \kappa\right)$ |
| Step Decay | $\Theta\left(\frac{d\sigma^2}{T} \cdot \log T\right)$ 
 (Ge et al. (2019); Wu et al. (2021); This work - Theorem 4) |
| Eigencurve | $\mathcal{O}\left(\frac{d\sigma^2}{T}\right)$ with skewed Hessian spectrums, 
 $\mathcal{O}\left(\frac{d\sigma^2}{T} \cdot \log \kappa\right)$ in worst case 
 (This work - Theorem 1, Corollary 2, 3) |

Fortunately, inverse time decay scheduler solves the problem by decaying the learning rate inversely proportional to the iteration number $t$, which achieves $\mathcal{O}(\frac{1}{T})$ convergence rate for strongly convex objectives, specifically, $\mathcal{O}(\frac{d\sigma^2}{T} \cdot \kappa)$. However, this is sub-optimal since the minimax optimal rate for SGD is $\mathcal{O}(\frac{d\sigma^2}{T})$ (Ge et al., 2019; Jain et al., 2018). Moreover, in practice, $\kappa$ can be very big for large neural networks, which makes inverse time decay scheduler undesirable for those models. This is when step decay (Goffin, 1977) comes to play. Empirically, it is widely adopted in tasks such as image classification and serves as a baseline for a lot of models. Theoretically, it has been proven that step decay can achieve nearly optimal convergence rate $\mathcal{O}(\frac{d\sigma^2}{T} \cdot \log T)$ for strongly convex least square regression (Ge et al., 2019). A tighter set of instance-dependent bounds in a recent work (Wu et al., 2021), which is carried out independently from ours, also proves its near optimality. Nevertheless, step decay is not always the best choice for image classification tasks. In practice, cosine decay (Loshchilov & Hutter, 2017) can achieve comparable or even better performance, but the reason behind this superior performance is still unknown in theory (Gotmare et al., 2018). All the aforementioned results are summarized in Table 1, along with our results in this paper. It is worth mentioning that the minimax optimal rate $\mathcal{O}(\frac{d\sigma^2}{T})$ can be achieved by iterate averaging methods (Jain et al., 2018; Bach & Moulines, 2013; Défossez & Bach, 2015; Frostig et al., 2015; Jain et al., 2016; Neu & Rosasco, 2018), but it is not a common practice to use them in training deep neural networks, so only the final iterate (Shamir, 2012) behavior of SGD is analyzed in this paper, i.e. the point right after the last iteration.

**Paper organization:** Section 2 describes the motivation of our `eigencurve` scheduler. Section 3 presents the exact form and convergence rate of the proposed `eigencurve` scheduler, along with the lower bound of step decay. Section 4 shows the experimental results. Section 5 discusses the discovery and limitation of `eigencurve` and Section 6 gives our conclusion.

## 2 MOTIVATION

In this section, we will give the main motivation and intuition of our `eigencurve` learning rate scheduler. We first give the scheduling strategy to achieve the optimal convergence rate in the case that the Hessian is diagonal. Then, we show that the inverse time learning rate is sub-optimal in most cases. Comparing these two scheduling methods brings up the reason why we should design eigenvalue distribution dependent learning rate scheduler.

Letting $H$ be a diagonal matrix $\text{diag}(\lambda_1, \lambda_2, \ldots, \lambda_d)$ and reformulating Eqn. (1.5), we have

$$
\begin{aligned}
w_{t+1} - w_* &= w_t - w_* - \eta_t(H(\xi)w_t - b(\xi)) \\
&= w_t - w_* - \eta_t(Hw_t - b) + \eta_t\left(Hw_t - b - (H(\xi)w_t - b(\xi))\right) \\
&= w_t - w_* - \eta_t(Hw_t - b - (Hw_* - b)) + \eta_t\left(Hw_t - b - (H(\xi)w_t - b(\xi))\right) \\
&= (I - \eta_t H)\left(w_t - w_*\right) + \eta_t n_t.
\end{aligned}
$$

It follows,

$$
\begin{aligned}
\mathbb{E}\left[\lambda_j\left(w_{t+1,j} - w_{*,j}\right)^2\right] &\overset{\mathbb{E}[n_t]=0}{=} \lambda_j\left\{(1 - \eta_t\lambda_j)^2\mathbb{E}\left[(w_{t,j} - w_{*,j})^2\right] + \eta_t^2\mathbb{E}\left\|[n_t]_j\right\|^2\right\} \\
&\overset{(1.7)}{\leq} (1 - \eta_t\lambda_j)^2 \cdot \lambda_j\mathbb{E}\left[(w_{t,j} - w_{*,j})^2\right] + \eta_t^2\lambda_j^2\sigma^2.
\end{aligned}
\tag{2.1}
$$

Since $H$ is diagonal, we can set step size scheduling for each dimension separately. Letting us choose step size $\eta_t$ coordinately with the step size $\eta_{t,j} = \frac{1}{\lambda_j(t+1)}$ being optimal for the $j$-th coordinate, then we have the following proposition.

**Proposition 1.** *Assume that $H$ is diagonal matrix with eigenvalues $\lambda_1 \geq \lambda_2 \geq \ldots, \lambda_d \geq 0$ and Eqn. (1.7) holds. If we set step size $\eta_{t,j} = \frac{1}{\lambda_j(t+1)}$, it holds that*

$$
2\mathbb{E}\left[f(w_{t+1}) - f(w_*)\right] = \mathbb{E}\left[\sum_{j=1}^{d}\lambda_j\left(w_{t+1,j} - w_{*,j}\right)^2\right] \leq \frac{\sum_{j=1}^{d}\lambda_j(w_{1,j} - w_{*,j})^2}{(t+1)^2} + \frac{t}{(t+1)^2}\cdot d\sigma^2.
\tag{2.2}
$$

The leading equality here is proved in Appendix G.1, with the followed inequality proved in Appendix E. From Eqn. (2.2), we can observe that choosing proper step sizes coordinately can achieve the optimal convergence rate (Ge et al., 2019; Jain et al., 2018). Instead, if the widely used inverse time scheduler $\eta_t = 1/(L + \mu t)$ is chosen, we can show that only a sub-optimal convergence rate can be obtained, especially when $\lambda_j$'s vary from each other.

**Proposition 2.** *If we set the inverse time step size $\eta_t = \frac{1}{(L+\mu t)}$, then we have*

$$
2\mathbb{E}\left[f(w_{t+1}) - f(w_*)\right] = \mathbb{E}\left[\sum_{j=1}^{d}\lambda_j\left(w_{t+1,j} - w_{*,j}\right)^2\right]
$$

$$
\leq \left(\frac{L+\mu}{L+\mu t}\right)^2\left(\sum_{j=1}^{d}\lambda_j(w_{1,j} - w_{*,j})^2\right) + \sum_{j=1}^{d}\left(\frac{\lambda_j^2}{2\lambda_j - \mu}\cdot\frac{1}{L+\mu t}\cdot\sigma^2 + \frac{\lambda_j^2\sigma^2}{(L+\mu t)^2}\right).
\tag{2.3}
$$

**Remark 1.** *Eqn. (2.2) shows that if one can choose step size coordinate-wise with step size $\eta_{t,j} = \frac{1}{\lambda_j(t+1)}$, then SGD can achieve a convergence rate*

$$
\mathbb{E}\left[f(w_{T+1}) - f(w_*)\right] \leq \mathcal{O}\left(\frac{d}{T}\cdot\sigma^2\right).
\tag{2.4}
$$

*which matches the lower bound (Ge et al., 2019; Jain et al., 2018). In contrast, replacing $L = \lambda_1$ and $\mu = \lambda_d$ in Proposition 2, we can obtain that the convergence rate of SGD being*

$$
\mathbb{E}\left[f(w_{T+1}) - f(w_*)\right] \leq \mathcal{O}\left(\frac{1}{T}\sum_{j=1}^{d}\frac{\lambda_j}{\lambda_d}\cdot\sigma^2\right).
\tag{2.5}
$$

*Since it holds that $\lambda_j \geq \lambda_d$, the convergence rate in Eqn. (2.4) is better than the one in Eqn. (2.5), especially when the eigenvalues of the Hessian ($H$ matrix) decay rapidly. In fact, the upper bound in Eqn. (2.5) is tight for the inverse time decay scheduling, as proven in Ge et al. (2019).*

**Main Intuition** The diagonal case $H = \text{diag}(\lambda_1, \lambda_2, \ldots, \lambda_d)$ provides an important intuition for designing eigenvalue dependent learning rate scheduling. In fact, for general non-diagonal $H$, letting $H = U\Lambda U^\top$ be the spectral decomposition of the Hessian and setting $w' = U^\top w$, then the Hessian becomes a diagonal matrix from perspective of updating $w'$, with the variance of the stochastic gradient being unchanged since $U$ is a unitary matrix. This is also the core idea of Newton's method and many second-order methods (Huang et al., 2020). However, given our focus in this paper being learning rate schedulers only, we move the relevant discussion of their relationship to Appendix H.

Proposition 1 and 2 imply that a good learning rate scheduler should decrease the error of each coordinate. The inverse time decay scheduler is only optimal for the coordinate related to the smallest eigenvalue. That's the reason why it is sub-optimal overall. Thus, we should reduce the learning rate gradually such that we can run a optimal learning rate associated to $\lambda_j$ to sufficiently drop the error of $j$-th coordinate. *Furthermore, given the total iteration $T$ and the eigenvalue distribution of the Hessian, we should allocate the running time for each optimal learning rate associated to $\lambda_j$.*

## 3 EIGENVALUE DEPENDENT STEP SCHEDULING

Just as discussed in Section 2, to obtain better convergence rate for SGD, we should consider Hessian's eigenvalue distribution and schedule the learning rate based on the distribution. In this section, we propose a novel learning rate scheduler for this task, which can be regarded as piecewise inverse time decay (see Figure 1). The method is very simple, we group eigenvalues according to their value and denote $s_i$ to be the number of eigenvalues lie in the range $\mathcal{R}_i = [\mu \cdot 2^i, \mu \cdot 2^{i+1})$, that is,

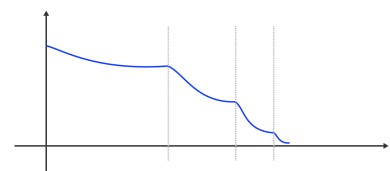

Figure 1: `Eigencurve` : piecewise inverse time decay scheduling.

$$s_i = \#\lambda_j \in [\mu \cdot 2^i, \mu \cdot 2^{i+1}). \tag{3.1}$$

Then, there are at most $I_{\max} = \log_2 \kappa$ such ranges. By the inverse time decay theory, the optimal learning rate associated to eigenvalues in the range $\mathcal{R}_i$ should be

$$\eta_t = \mathcal{O}\left(\frac{1}{2^{i-1}\mu \cdot (t - t_i)}\right), \text{ with } 0 = t_0 < t_1 < t_2 < \cdots < t_{I_{\max}} = T. \tag{3.2}$$

Our scheduling strategy is to run the optimal learning rate for eigenvalues in each $\mathcal{R}_i$ for a period to sufficiently decrease the error associated to eigenvalues in $\mathcal{R}_i$.

To make the step size sequence $\{\eta_t\}_{t=1}^T$ monotonely decreasing, we define the step sizes as

$$\eta_t = \frac{1}{L + \mu\sum_{j=1}^{i-1}\Delta_j 2^{j-1} + 2^{i-1}\mu(t - t_{i-1})} \quad \text{if} \quad t \in [t_{i-1}, t_i) \tag{3.3}$$

where

$$0 = t_0 < t_1 < t_2 < \cdots < t_{I_{\max}} = T, \ \Delta_i = t_i - t_{i-1}, \text{ and } I_{\max} = \log_2 \kappa. \tag{3.4}$$

To make the total error, that is the sum of error associated with $\mathcal{R}_i$, to be small, we should allocate $\Delta_i$ according to $s_{i-1}$'s. Intuitively, a large portion of eigenvalues lying in the range $\mathcal{R}_i$ should allocate a large portion of iterations. Specifically, we propose to allocate $\Delta_i$ as follows:

$$\Delta_i = \frac{\sqrt{s_{i-1}}}{\sum_{i'=0}^{I_{\max}-1}\sqrt{s_{i'}}} \cdot T, \text{ with } s_i = \#\lambda_j \in [\mu \cdot 2^i, \mu \cdot 2^{i+1}). \tag{3.5}$$

In the rest of this section, we will show that the step size scheduling according to Eqn. (3.3) and (3.5) can achieve better convergence rate than the one in Ge et al. (2019) when $s_i$ is non-uniformly distributed. In fact, a better $\Delta_i$ allocation can be calculated using numerical optimization.

### 3.1 THEORETICAL ANALYSIS

**Lemma 1.** *Let objective function $f(x)$ be quadratic and Assumption (1.7) hold. Running SGD for $T$-steps starting from $w_0$ and a learning rate sequence $\{\eta_t\}_{t=1}^T$ defined in Eqn. (3.3), the final iterate*

$w_{T+1}$ *satisfies*

$$\mathbb{E}\left[f(w_{T+1}) - f(w_*)\right] \leq (f(w_0) - f(w_*)) \cdot \frac{\kappa^2}{\Delta_1^2} + \frac{15}{2} \cdot \sigma^2 \mu \sum_{\tilde{i}=0}^{I_{\max}-1} \frac{2^{\tilde{i}+1} s_{\tilde{i}}}{L + \mu \sum_{j=1}^{\tilde{i}+1} \Delta_j 2^{j-1}}.$$
(3.6)

Since the bias term is a high order term, the variance term in Eqn. (3.6) dominates the error for $w_{T+1}$. For simplicity, instead of using numerical methods to find the optimal $\{\Delta_i\}$, we propose to use $\Delta_i$ defined in Eqn. (3.5). The value of $\Delta_i$ is linear to square root of the number of eigenvalues lying in the range $[\mu \cdot 2^{i-1}, \mu \cdot 2^i)$. Using such $\Delta_i$, `eigencurve` has the following convergence property.

**Theorem 1.** *Let objective function $f(x)$ be quadratic and Assumption (1.7) hold. Running SGD for $T$-steps starting from $w_0$, a learning rate sequence $\{\eta_t\}_{t=1}^T$ defined in Eqn. (3.3) and $\Delta_i$ defined in Eqn. (3.5), the final iterate $w_{T+1}$ satisfies*

$$\mathbb{E}\left[f(w_{T+1}) - f(w_*)\right] \leq (f(w_0) - f(w_*)) \cdot \frac{\kappa^2 \cdot \left(\sum_{i=0}^{I_{\max}-1} \sqrt{s_i}\right)^2}{s_0 T^2} + \frac{15 \left(\sum_{i=0}^{I_{\max}-1} \sqrt{s_i}\right)^2}{T} \cdot \sigma^2.$$

Please refer to Appendix D, F and G for the full proof of Lemma 1 and Theorem 1. The variance term $\frac{15\left(\sum_{i=0}^{I_{\max}-1} \sqrt{s_i}\right)^2}{T} \cdot \sigma^2$ in above theorem shows that when $s_i$'s vary largely from each other, then the variance can be close to $\mathcal{O}\left(\frac{d}{T} \cdot \sigma^2\right)$ which matches the lower bound (Ge et al., 2019). For example, letting $I_{\max} = 100$, $s_0 = 0.99d$ and $s_i = \frac{0.01}{99}d$, we can obtain that

$$\frac{\left(\sum_{i=0}^{99} \sqrt{s_i}\right)^2}{T} \cdot \sigma^2 = (\sqrt{0.99} + 99 \times \sqrt{0.01/99})^2 \cdot \frac{d}{T} \cdot \sigma^2 < \frac{4d}{T} \cdot \sigma^2.$$

We can observe that if the variance of $s_i$'s is large, the variance term in Theorem 1 can be close to $d\sigma^2/T$. More generally, as rigorously stated in Corollary 2, `eigencurve` achieves minimax optimal convergence rate if the Hessian spectrum satisfies an extra assumption of "power law": the density of eigenvalue $\lambda$ is exponentially decaying with increasing value of $\lambda$ in log scale, i.e. $\ln(\lambda)$. This assumption comes from the observation of estimated Hessian spectrums in practice (see Figure 2), which will be further illustrated in Section 4.1.

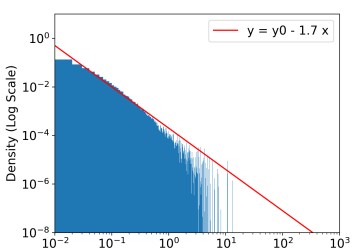

Figure 2: Power law observed in ResNet-18 on ImageNet, both eigenvalue (x-axis) and density (y-axis) are plotted in log scale.

**Corollary 2.** *Given the same setting as in Theorem 1, when Hessian $H$'s eigenvalue distribution $p(\lambda)$ satisfies "power law", i.e.*

$$p(\lambda) = \frac{1}{Z} \cdot \exp(-\alpha(\ln(\lambda) - \ln(\mu))) = \frac{1}{Z} \cdot \left(\frac{\mu}{\lambda}\right)^\alpha$$
(3.7)

*for some $\alpha > 1$, where $Z = \int_\mu^L (\mu/\lambda)^\alpha d\lambda$, there exists a constant $C(\alpha)$ which only depends on $\alpha$, such that the final iterate $w_{T+1}$ satisfies*

$$\mathbb{E}\left[f(w_{T+1}) - f(w_*)\right] \leq \left((f(w_0) - f(w_*)) \cdot \frac{\kappa^2}{T^2} + \frac{d\sigma^2}{T}\right) \cdot C(\alpha).$$

Please refer to Appendix G.3 for the proof. As for the worst-case guarantee, it is easy to check that only when $s_i$'s are equal to each other, that is, $s_i = d/I_{\max} = d/\log_2(\kappa)$, the variance term reaches its maximum.

**Corollary 3.** *Given the same setting as in Theorem 1, when $s_i = d/\log_2(\kappa)$ for all $0 \leq i \leq I_{\max} - 1$, the variance term in Theorem 1 reaches its maximum and $w_{T+1}$ satisfies*

$$\mathbb{E}\left[f(w_{T+1}) - f(w_*)\right] \leq (f(w_0) - f(w_*)) \cdot \frac{\kappa^2 \log^2 \kappa}{T^2} + \frac{15d \cdot \log \kappa}{T} \sigma^2.$$

**Remark 2.** *When $s_i$'s vary from each other, our eigenvalue dependent learning rate scheduler can achieve faster convergence rate than eigenvalue independent scheduler such as step decay which suffers from an extra $log(T)$ term (Ge et al., 2019). Only when $s_i$'s are equal to each other, Corollary 3 shows that the bound of variance matches to lower bound up to $\log \kappa$ which is same to the one in Proposition 3 of Ge et al. (2019).*

Furthermore, we show that this $\log T$ gap between step decay and `eigencurve` certainly exists for problem instances of skewed Hessian spectrums. For simplicity, we only discuss the case where $H$ is diagonal.

**Theorem 4.** *Let objective function $f(x)$ be quadratic. We run SGD for $T$-steps starting from $w_0$ and a step decay learning rate sequence $\{\eta_t\}_{t=1}^T$ defined in Algorithm 1 of Ge et al. (2019) with $\eta_1 \leq 1/L$. As long as (1) $H$ is diagonal, (2) The equality in Assumption (1.7) holds, i.e. $\mathbb{E}_\xi \left[ n_t n_t^\top \right] = \sigma^2 H$ and (3) $\lambda_j \left( w_{0,j} - w_{*,j} \right)^2 \neq 0$ for $\forall j = 1, 2, \ldots, d$, the final iterate $w_{T+1}$ satisfies,*

$$\mathbb{E}\left[f(w_{T+1}) - f(w_*)\right] = \Omega \left( \frac{d\sigma^2}{T} \cdot \log T \right)$$

The proof is provided in Appendix G.4. Removing this extra $\log T$ term may not seem to be a big deal in theory, but experimental results suggest the opposite.

## 4 EXPERIMENTS

To demonstrate `eigencurve`'s practical value, empirical experiments are conducted on the task of image classification [1]. Two well-known dataset are used: CIFAR-10 (Krizhevsky et al., 2009) and ImageNet (Deng et al., 2009). For full experimental results on more datasets, please refer to Appendix A.

### 4.1 HESSIAN SPECTRUM'S SKEWNESS IN PRACTICE

According to estimated[2] eigenvalue distributions of Hessian on CIFAR-10 and ImageNet, as shown in Figure 3, it can be observed that all of them are highly skewed and share a similar tendency: *A large portion of small eigenvalues and a tiny portion of large eigenvalues*. This phenomenon has also been observed and explained by other researchers in the past(Sagun et al., 2017; Arjevani & Field, 2020). On top of that, when we plot both eigenvalues and density in log scale, the "power law" arises. Therefore, if the loss surface can be approximated by quadratic objectives, then `eigencurve` has already achieved optimal convergence rate for those practical settings. The exact values of the extra constant terms are presented in Appendix A.2.

### 4.2 IMAGE CLASSIFICATION ON CIFAR-10 WITH EIGENCURVE SCHEDULING

This optimality in theory induces `eigencurve`'s superior performance in practice, which is demonstrated in Table 2 and Figure 4. The full set of figures are available in Appendix A.8. All models are trained with stochastic gradient descent (SGD), no momentum, batch size 128 and weight decay $wd = 0.0005$. For full details of the experiment setup, please refer to Appendix B.

### 4.3 INSPIRED PRACTICAL SCHEDULES WITH SIMPLE FORMS

By simplifying the form of `eigencurve` and capturing some of its key properties, two simple and practical schedules are proposed: Elastic Step Decay and Cosine-power Decay, whose empirical performance are better than or at least comparable to cosine decay. Due to page limit, we leave all the experimental results in Appendix A.5, A.6, A.7.

$$\text{Elastic Step Decay: } \eta_t = \eta_0/2^k \text{, if } t \in \left[(1 - r^k)T, (1 - r^{k+1})T\right) \tag{4.1}$$

$$\text{Cosine-power Decay: } \eta_t = \eta_{\min} + (\eta_0 - \eta_{\min}) \left[\frac{1}{2}(1 + \cos(\frac{t}{t_{\max}}\pi))\right]^\alpha \tag{4.2}$$

---

[1]Code: https://github.com/opensource12345678/why_cosine_works/tree/main
[2]Please refer to Appendix B.2 for details of the estimation and preprocessing procedure.

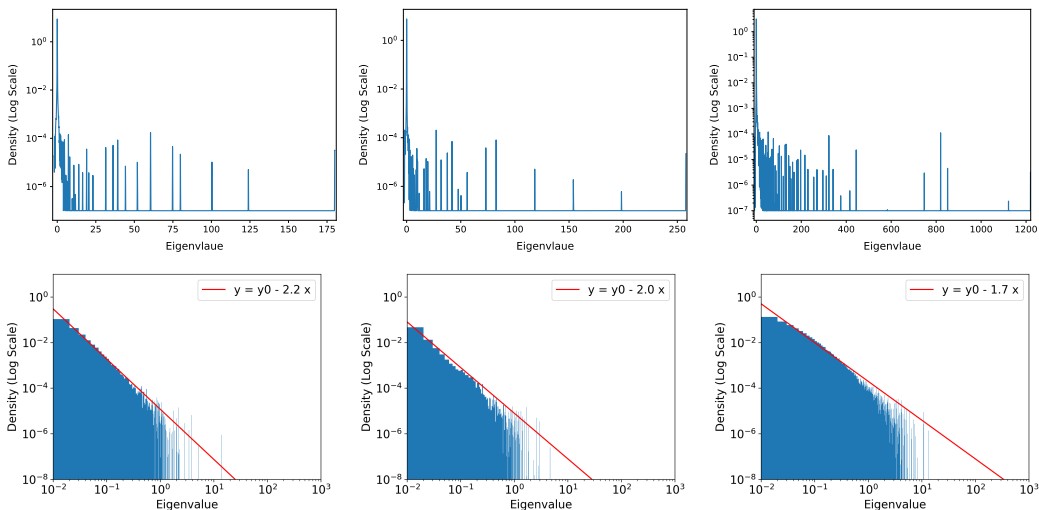

Figure 3: The estimated eigenvalue distribution of Hessian for ResNet-18 on CIFAR-10, GoogLeNet on CIFAR-10 and ResNet-18 on ImageNet respectively. Notice that the density here is all shown in log scale. First row: original scale for eigenvalues. Second row: log scale for preprocessed eigenvalues.

Table 2: CIFAR-10: training losses and test accuracy of different schedules. Step Decay denotes the scheduler proposed in Ge et al. (2019) and General Step Decay means the same type of scheduler with searched interval numbers and decay rates. "*" before a number means at least one occurrence of loss explosion among all 5 trial experiments.

| #Epoch | Schedule | ResNet-18 | | GoogLeNet | | VGG16 | |
|---|---|---|---|---|---|---|---|
| | | Loss | Acc(%) | Loss | Acc(%) | Loss | Acc(%) |
| =10 | Inverse Time Decay | 1.58±0.02 | 79.45±1.00 | 2.61±0.00 | 86.54±0.94 | 2.26±0.00 | 84.47±0.74 |
| | Step Decay | 1.82±0.04 | 73.77±1.48 | 2.59±0.02 | 87.04±0.48 | 2.42±0.45 | 82.98±0.27 |
| | General Step Decay | 1.52±0.02 | 81.99±0.35 | 1.93±0.03 | 88.32±1.32 | 2.14±0.42 | 86.79±0.36 |
| | Cosine Decay | 1.42±0.01 | 84.23±0.07 | 1.94±0.00 | 90.56±0.31 | 2.03±0.00 | 87.99±0.13 |
| | Eigencurve | **1.36±0.01** | **85.62±0.28** | **1.33±0.00** | **90.65±0.15** | **1.87±0.00** | **88.73±0.11** |
| =100 | Inverse Time Decay | 0.73±0.00 | 90.82±0.43 | 0.62±0.02 | 92.05±0.69 | 1.32±0.62 | *76.24±13.77 |
| | Step Decay | 0.26±0.01 | 91.39±1.03 | 0.28±0.00 | 92.83±0.15 | 0.59±0.00 | 91.37±0.20 |
| | General Step Decay | 0.17±0.00 | 93.97±0.21 | 0.13±0.00 | 94.18±0.18 | 0.20±0.00 | *92.36±0.46 |
| | Cosine Decay | 0.17±0.00 | 94.04±0.21 | 0.12±0.00 | 94.62±0.11 | 0.20±0.00 | **93.17±0.05** |
| | Eigencurve | **0.14±0.00** | **94.05±0.18** | **0.12±0.00** | **94.75±0.15** | **0.18±0.00** | 92.88±0.24 |

Figure 4: Example: CIFAR-10 results for ResNet-18, with #Epoch = 100. Left: training losses. Right: test accuracy. For full figures of this experiment, please refer to Appendix A.8.

## 5  DISCUSSION

**Cosine Decay and Eigencurve**   For ResNet-18 on CIFAR-10 dataset, `eigencurve` scheduler presents an extremely similar learning rate curve to cosine decay, especially when the number of training epochs is set to 100, as shown in Figure 5. This directly links cosine decay to our theory: the empirically superior performance of cosine decay is very likely to stem from the utilization of the "skewness" among Hessian matrix's eigenvalues. For other situations, especially when the number of iterations is small, as shown in Table 2, `eigencurve` presents a better performance than cosine decay .

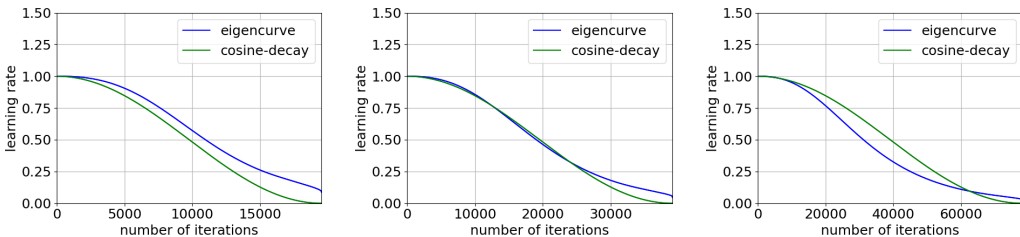

Figure 5: `Eigencurve` 's learning rate curve generated by the estimated eigenvalue distribution for ResNet-18 on CIFAR-18 after training 50/100/200 epochs. The cosine decay's learning rate curve (green) is also provided for comparison.

**Sensitiveness to Hessian's Eigenvalue Distributions**   One limitation of `eigencurve` is that it requires a precomputed eigenvalue distribution of objective functions's Hessian matrix, which can be time-consuming for large models. This issue can be overcome by reusing the estimated eigenvalue distribution from similar settings. Further experiments on CIFAR-10 suggest the effectiveness of this approach. Please refer to Appendix A.3 for more details. This evidence suggests that `eigencurve`'s performance is not very sensitive to estimated eigenvalue distributions.

**Relationship with Numerically Near-optimal Schedulers**   In Zhang et al. (2019), a dynamic programming algorithm was proposed to find almost optimal schedulers if the exact loss of the quadratic objective is accessible. While it is certainly the case, `eigencurve` still possesses several additional advantages over this type of approaches. First, `eigencurve` can be used to find simple-formed schedulers. Compared with schedulers numerically computed by dynamic programming, `eigencurve` provides an analytic framework, so it is able to bypass the Hessian spectrum estimation process if some useful assumptions of the Hessian spectrum can be obtained, such as "power law". Second, `eigencurve` has a clear theoretical convergence guarantee. Dynamic programming can find almost optimal schedulers, but the convergence property of the computed scheduler is still unclear. Our work fills this gap.

## 6  CONCLUSION

In this paper, a novel learning rate schedule named `eigencurve` is proposed, which utilizes the "skewness" of objective's Hessian matrix's eigenvalue distribution and reaches minimax optimal convergence rates for SGD on quadratic objectives with skewed Hessian spectrums. This condition of skewed Hessian spectrums is observed and indeed satisfied in practical settings of image classification. Theoretically, `eigencurve` achieves no worse convergence guarantee than step decay for quadratic functions and reaches minimax optimal convergence rate (up to a constant) with skewed Hessian spectrums, e.g. under "power law". Empirically, experimental results on CIFAR-10 show that `eigencurve` significantly outperforms step decay, especially when the number of epochs is small. The idea of `eigencurve` offers a possible explanation for cosine decay's effectiveness in practice and inspires two practical families of schedules with simple forms.

ACKNOWLEDGEMENT

This work is supported by GRF 16201320. Rui Pan acknowledges support from the Hong Kong PhD Fellowship Scheme (HKPFS). The work of Haishan Ye was supported in part by National Natural Science Foundation of China under Grant No. 12101491.

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

## A  MORE EXPERIMENTAL RESULTS

### A.1  RIDGE REGRESSION

We compare different types of schedulings on ridge regression

$$f(w) = \frac{1}{n}||Xw - Y||_2^2 + \alpha||w||_2^2.$$

This experiment is only an empirical proof of our theory. In fact, the optima of ridge regression has a closed form and can be directly computed with

$$w_* = \left(X^\top X + n\alpha I\right)^{-1} X^T Y.$$

Thus the optimal training loss $f(w_*)$ can be calculated accordingly. In all experiments, we use the loss gap $f(w^T) - f(w_*)$ as our performance metric.

Experiments are conducted on a4a datasets (Chang & Lin, 2011; Dua & Graff, 2017) (`https://www.csie.ntu.edu.tw/~cjlin/libsvmtools/datasets/binary.html#a4a/`), which contains $4,781$ samples and $123$ features. This dataset is chosen majorly because it has a moderate number of samples and features, which enables us to compute the exact Hessian matrix $H = 2(X^\top X/n + \alpha I)$ and its corresponding eigenvalue distribution in acceptable time and space consumption.

In all of our experiments, we set $\alpha = 10^{-3}$. The model is optimized via SGD without momentum, with batch size 1, initial learning rate $\eta_0 \in \{0.1, 0.06, 0.03, 0.02, 0.01, 0.006, 0.003, 0.002, 0.001, 0.0006, 0.0003, 0.0002, 0.0001\}$ and learning rate of last iteration $\eta_{\min} \in \{0.1, 0.01, 0.001, 0.0001, 0.00001, 0, \text{"UNRESTRICTED"}\}$. Here "UNRESTRICTED" denotes the case where $\eta_{\min}$ is not set, which is useful for `eigencurve`, who can decide the learning rate curve without setting $\eta_{\min}$. Given $\eta_0$ and $\eta_{\min}$, we adjust all schedulers as follows. For inverse time decay $\eta_t = \eta_0/(1 + \gamma\eta_0 t)$ and exponential decay $\eta_t = \gamma^t\eta_0$, the hyperparameter $\gamma$ is computed accordingly based on $\eta_0$ and $\eta_{\min}$. For cosine decay, $\eta_0$ and $\eta_{\min}$ is directly used, with no restart adopted. For `eigencurve`, the learning rate curve is linearly scaled to match the given $\eta_{\min}$.

In addition, for `eigencurve`, we use the eigenvalue distribution of the Hessian matrix, which is directly computed via eigenvalue decomposition, as shown in Figure 6.

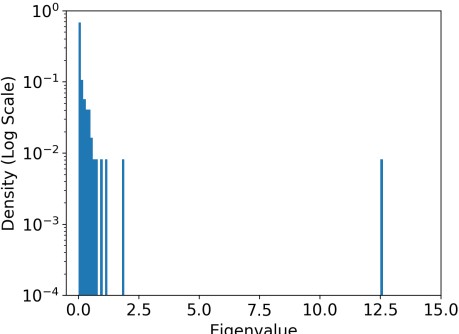 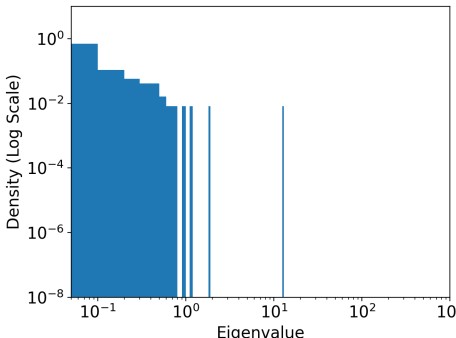

Figure 6: The eigenvalue distribution of Hessian for ridge regression on a4a. Left: original scale for eigenvalues. Right: log scale for eigenvalues. Notice that the density here is shown in log scale.

All experimental results demonstrate that `eigencurve` can obtain similar or better training losses when compared with other schedulers, as shown in Table 3.

Table 3: Ridge regression: training loss gaps of different schedules over 5 trials.

| Schedule | Training loss - optimal training loss: $f(w^T) - f(w_*)$ | |
| --- | --- | --- |
| | #Epoch = 1 | #Epoch = 5 |
| Constant | 0.014963±0.001369 | 0.004787±0.000175 |
| Inverse Time Decay | 0.007284±0.000190 | 0.002098±0.000160 |
| Exponetial Decay | 0.008351±0.000360 | 0.000931±0.000100 |
| Cosine Decay | 0.007767±0.000006 | 0.001167±0.000142 |
| Eigencurve | **0.006977±0.000197** | **0.000676±0.000069** |
| Schedule | #Epoch = 25 | #Epoch = 250 |
| Constant | 0.001351±0.000179 | 0.000122±0.000009 |
| Inverse Time Decay | 0.000637±0.000143 | 0.000011±0.000001 |
| Exponetial Decay | 0.000048±0.000007 | **0.000000±0.000000** |
| Cosine Decay | 0.000054±0.000005 | **0.000000±0.000000** |
| Eigencurve | **0.000045±0.000008** | **0.000000±0.000000** |

## A.2 EXACT VALUE OF THE EXTRA TERM ON CIFAR-10 EXPERIMENTS

In Section 4.1, we have already given the qualitative evidence that shows `eigencurve` 's optimality for practical settings on CIFAR-10. Here we strengthen this argument by providing the quantitative evidence as well. The exact value of the extra term is presented in Table 4, where we assume CIFAR-10 has batch size 128, number of epochs 200 and weight decay $5 \times 10^{-4}$, while ImageNet has batch size 256, number of epochs 90 and weight decay $10^{-4}$.

Table 4: Convergence rate of SGD with common schedulers given the estimated eigenvalue distribution of Hessian, assuming the objective is quadratic.

| Scheduler | Convergence rate of SGD in quadratic functions | Value of the extra term | | | |
| --- | --- | --- | --- | --- | --- |
| | | CIFAR-10 + ResNet18 | CIFAR-10 + GoogLeNet | CIFAR-10 + VGG16 | ImageNet + ResNet18 |
| Inverse Time Decay | $\Theta\left(\frac{d\sigma^2}{T} \cdot \kappa\right)$ | $3.39 \times 10^5$ | $4.92 \times 10^5$ | $6.50 \times 10^5$ | $6.80 \times 10^6$ |
| Step Decay | $\Theta\left(\frac{d\sigma^2}{T} \cdot \log T\right)$ | 16.25 | 16.25 | 16.25 | 18.78 |
| Eigencurve | $\mathcal{O}\left(\frac{d\sigma^2}{T} \cdot \frac{\left(\sum_{i=0}^{I_{\max}-1}\sqrt{s_i}\right)^2}{d}\right)$ where $I_{\max} = \log_2 \kappa$, $s_i = \#\lambda_j \in [\mu \cdot 2^i, \mu \cdot 2^{i+1})$ | **8.15** | **5.97** | **7.12** | **12.61** |
| Minimax optimal rate | $\Omega\left(\frac{d\sigma^2}{T}\right)$ | 1 | 1 | 1 | 1 |

It is worth noticing that the extra term's value of `eigencurve` is independent from the number of iterations $T$, since the value $(\sum_{i=0}^{I_{\max}-1}\sqrt{s_i})^2/d$ only depends on the Hessian spectrum. So basically `eigencurve` has already achieved the minimax optimal rate (up to a constant) for models and datasets listed in Table 4, if the loss landscape around the optima can be approximated by quadratic functions. For full details of the estimation process, please refer to Appendix B.

## A.3 REUSING EIGENCURVE FOR DIFFERENT MODELS ON CIFAR-10

For image classification tasks on CIFAR-10, we check the performance of reusing ResNet-18's eigenvalue distribution for other models. As shown in Table 5, experimental results demonstrate that Hessian's eigenvalue distribution of Resnet-18 on CIFAR-10 can be applied to GoogLeNet/VGG16 and still achieves good peformance. Here the experiment settings are exactly the same as Section 4.2 in main paper.

Table 5: CIFAR-10: training losses and test accuracy of different schedules over 5 trials. Here all `eigencurve` schedules are generated based on ResNet-18's Hessian spectrums. "*" before a number means at least one occurrence of loss explosion among all 5 trial experiments.

| #Epoch | Schedule | GoogLeNet | | VGG16 | |
|--------|----------|-----------|------|-------|------|
| | | Loss | Acc(%) | Loss | Acc(%) |
| =10 | Inverse Time Decay | 2.61±0.00 | 86.54±0.94 | 2.26±0.00 | 84.47±0.74 |
| | Step Decay | 2.59±0.02 | 87.04±0.48 | 2.42±0.45 | 82.98±0.27 |
| | General Step Decay | 1.93±0.03 | 88.32±1.32 | 2.14±0.42 | 86.79±0.36 |
| | Cosine Decay | 1.94±0.00 | 90.56±0.31 | 2.03±0.00 | 87.99±0.13 |
| | Eigencurve (transferred) | **1.65±0.00** | **91.17±0.20** | **1.89±0.00** | **88.17±0.32** |
| =100 | Inverse Time Decay | 0.62±0.02 | 92.05±0.69 | 1.32±0.62 | *76.24±13.77 |
| | Step Decay | 0.28±0.00 | 92.83±0.15 | 0.59±0.00 | 91.37±0.20 |
| | General Step Decay | 0.13±0.00 | 94.18±0.18 | 0.20±0.00 | *92.36±0.46 |
| | Cosine Decay | 0.12±0.00 | 94.62±0.11 | 0.20±0.00 | **93.17±0.05** |
| | Eigencurve (transferred) | **0.11±0.00** | **94.81±0.19** | **0.20±0.00** | **93.17±0.09** |

## A.4 COMPARISON WITH EXPONENTIAL MOVING AVERAGE ON CIFAR-10

Besides learning rate schedules, Exponential Moving Averaging (EMA) method

$$\overline{w}_t = \alpha \sum_{k=0}^{t} (1-\alpha)^{t-k} w_k \quad \Longleftrightarrow \quad \overline{w}_t = \alpha w_t + (1-\alpha)\overline{w}_{t-1}$$

is another competitive practical method that is commonly adopted in training neural networks with SGD. Thus, it is natural to ask whether `eigencurve` can beat this method as well. The answer is yes. In Table 6, we present additional experimental results on CIFAR-10 to compare the performance of `eigencurve` and exponential moving averaging. It can be observed that there is a large performance gap between those two methods.

Table 6: CIFAR-10: training losses and test accuracy of Exponential Moving Average (EMA) and `eigencurve` with #Epoch = 100 over 5 trials. For EMA, we search its constant learning rate $\eta_t = \eta_0 \in \{1.0, 0.6, 0.3, 0.2, 0.1\}$ and decay $\alpha \in \{0.9, 0.95, 0.99, 0.995, 0.999\}$. Other settings remain the same as Section 4.2.

| Method/Schedule | ResNet-18 | | GoogLeNet | | VGG16 | |
|-----------------|-----------|------|-----------|------|-------|------|
| | Loss | Acc(%) | Loss | Acc(%) | Loss | Acc(%) |
| EMA | 0.30±0.01 | 90.09±0.82 | 0.33±0.01 | 93.42±0.26 | 0.49±0.00 | 91.87±0.82 |
| Eigencurve | **0.14±0.00** | **94.05±0.18** | **0.12±0.00** | **94.75±0.15** | **0.18±0.00** | **92.88±0.24** |

## A.5 IMAGENET CLASSIFICATION WITH ELASTIC STEP DECAY

One key observation in CIFAR-10 experiments is the existence of "power law" shown in Figure 3. Also, notice that in the form of `eigencurve`, specifically Eqn. (3.5), iteration interval length $\Delta_i$ is proportional to the square root of eigenvalue density $s_i$ in range $[\mu \cdot 2^i, \mu \cdot 2^{i+1})$. Combining those two facts together, it suggests the length of "learning rate interval" should have lengths exponentially decreasing.

Based on this idea, Elastic Step Decay (ESD) is proposed, which has the following form,

$$\eta_t = \eta_0/2^k \text{ , if } t \in \left[(1-r^k)T, (1-r^{k+1})T\right)$$

Compared to general step decay with adjustable interval lengths, elastic step decay does not require manual adjustment for the length of each interval. Instead, they are all controlled by one hyper-parameter $r \in (0, 1)$, which decides the "shrinking speed" of interval lengths. Experiments on

CIFAR-10, CIFAR-100 and ImageNet demonstrate its superiority in practice, as shown in Table 7, Table 8.

For experiments on CIFAR-10/CIFAR-100, we adopt the same settings as `eigencurve`, except we only use common step decay with three same-length intervals + decay factor 10.

Table 7: Elastic Step Decay on CIFAR-10/CIFAR-100: test accuracy(%) of different schedules over 5 trials. "*" before a number means at least one occurrence of loss explosion among all 5 trial experiments.

| #Epoch | Schedule | ResNet-18 | | GoogLeNet | | VGG16 | |
|---|---|---|---|---|---|---|---|
| | | CIFAR-10 | CIFAR-100 | CIFAR-10 | CIFAR-100 | CIFAR-10 | CIFAR-100 |
| =10 | Inverse Time Decay | 79.45±1.00 | 48.73±1.66 | 86.54±0.94 | 57.90±1.27 | 84.47±0.74 | 50.04±0.83 |
| | Step Decay | 79.67±0.74 | 54.54±0.26 | 88.37±0.13 | 63.05±0.35 | 85.18±0.06 | 45.86±0.31 |
| | Cosine Decay | 84.23±0.07 | 61.26±1.11 | 90.56±0.31 | 69.09±0.27 | 87.99±0.13 | 55.42±0.28 |
| | ESD | **85.38±0.38** | **64.17±0.57** | **91.23±0.33** | **70.46±0.41** | **88.67±0.21** | **57.23±0.39** |
| =100 | Inverse Time Decay | 90.82±0.43 | 69.82±0.37 | 92.05±0.69 | 73.54±0.28 | *76.24±13.77 | 67.70±0.49 |
| | Step Decay | 93.68±0.07 | 73.13±0.12 | 94.13±0.32 | 76.80±0.16 | 92.62±0.15 | 70.02±0.41 |
| | Cosine Decay | 94.04±0.21 | 74.65±0.41 | 94.62±0.11 | 78.13±0.54 | 93.17±0.05 | 72.47±0.28 |
| | ESD | **94.06±0.11** | **74.76±0.33** | **94.65±0.11** | **78.23±0.20** | **93.25±0.12** | **72.50±0.26** |

For experiments on ImageNet, we use ResNet-50 trained via SGD without momentum, batch size 256 and weight decay $wd = 10^{-4}$. Since no momentum is used, the initial learning rate is set to $\eta_0 = 1.0$ instead of $\eta_0 = 0.1$. Two step decay baselines are adopted. "Step Decay [30-60]" is the common choice that decays the learning rate 10 folds at the end of epoch 30 and epoch 60. "Step Decay [30-60-80]" is another popular choice for the ImageNet setting (Goyal et al., 2018), which further decays learning rate 10 folds at epoch 80. For cosine decay scheduler, the hyperparameter $\eta_{\min}$ is set to be 0. As for the dataset, we use the common ILSVRC 2012 dataset, which contains 1000 classes, around 1.2M images for training and 50,000 images for validation. For all experiments, we search $r \in \{1/2, 1/\sqrt{2}\}$ for ESD, with other hyperparameter search and selection process being the same as `eigencurve`.

Table 8: Elastic Step Decay on ImageNet-1k: Losses and validation accuracy of different schedulings for ResNet-50 with #Epoch=90 over 3 trials.

| | Schedule | Training loss | Top-1 validation acc(%) | Top-5 validation acc(%) |
|---|---|---|---|---|
| #Epoch=90 | Step Decay [30-60] | 1.4726±0.0057 | 75.55±0.13 | 92.63±0.08 |
| | Step Decay [30-60-80] | 1.4738±0.0080 | 76.05±0.33 | 92.83±0.15 |
| | Cosine Decay | 1.4697±0.0049 | 76.57±0.07 | 93.25±0.05 |
| | ESD ($r = 1/\sqrt{2}$) | **1.4317±0.0027** | **76.79±0.10** | **93.31±0.05** |

### A.6 LANGUAGE MODELING WITH ELASTIC STEP DECAY

More experiments on language modeling are conducted to further demonstrate Elastic Step Decay's superiority over other schedulers.

For all experiments, we follow almost the same setting in Zaremba et al. (2015), where a large regularized LSTM recurrent neural network (Hochreiter & Schmidhuber, 1997) is trained on Penn Treebank (Marcus et al., 1993) for language modeling task. The Penn Treebank dataset has a training set of 929k words, a validation set of 73k words and a test set of 82k words. SGD without momentum is adopted for training, with batch size 20 and 35 unrolling steps in LSTM.

Other details are exactly the same, except for the number of training epochs. In Zaremba et al. (2015), it uses 55 epochs to train the large regularized LSTM, which is changed to 30 epochs in our setting, since we found that the model starts overfitting after 30 epochs. We conducted hyperparameter search for all schedules, as shown in Table 9.

Table 9: Hyperparameter search for schedulers.

| Scheduler | Form | Hyperparameter choices |
|---|---|---|
| Inverse Time Decay | $\eta_t = \frac{\eta_0}{1 + \lambda \cdot \eta_0 \cdot t}$ | $\eta_0 \in \{10^0, 10^{-1}, 10^{-2}, 10^{-3}\}$, and set $\lambda$, so that $\eta_{\min} \in \{10^{-2}, 10^{-3}, 10^{-4}, 10^{-5}, 10^{-6}\}$ |
| General Step Decay | $\eta_t = \eta_0 \cdot \gamma^k$, if $t \in [k, k+1) \cdot \frac{T}{K}$ | $\eta_0 = 1$, $K \in \{3, 4, 5, \lfloor \log T \rfloor\}, \lfloor \log T \rfloor + 1\}$, $\gamma \in \{\frac{1}{2}, \frac{1}{5}, \frac{1}{10}\}$ |
| Cosine Decay | $\eta_t = \eta_{\min} + \frac{1}{2} (\eta_0 - \eta_{\min}) \left(1 + \cos\left(\frac{t\pi}{T}\right)\right)$ | $\eta_0 \in \{10^0, 10^{-1}, 10^{-2}, 10^{-3}\}$, $\eta_{\min} \in \{10^{-2}, 10^{-3}, 10^{-4}, 0\}$ |
| Elastic Step Decay | $\eta_t = \eta_0/2^k$, if $t \in \left[(1 - r^k)T, (1 - r^{k+1})T\right)$ | $\eta_0 = 1$, $r \in \{2^{-1}, 2^{-1/2}, 2^{-1/3}, 2^{-1/5}, 2^{-1/20}\}$, |
| Baseline | $\eta_t = \begin{cases} \eta_0 & \text{for first 14 epochs} \\ \frac{\eta_0}{1.15^k} & \text{for epoch } k + 14 \end{cases}$ | $\eta_0 = 1$ |

Experimental results show that Elastic Step Decay significantly outperforms other schedulers, as shown in Table 10.

Table 10: Scheduler performance on LSTM + Penn Treebank over 5 trials.

| Scheduler | Validation perplexity | Test perplexity |
|---|---|---|
| Inverse Time Decay | 114.9±1.1 | 112.7±1.1 |
| General Step Decay | 82.4±0.1 | 79.1±0.2 |
| Baseline (Zaremba et al., 2015) | 82.2 | 78.4 |
| Cosine Decay | 82.4±0.4 | 78.5±0.4 |
| Elastic Step Decay | **81.1±0.2** | **77.4±0.3** |

## A.7 IMAGE CLASSIFICATION ON IMAGENET WITH COSINE-POWER SCHEDULING

Another key observation in CIFAR-10 experiments is that `eigencurve` 's learning rate curve shape changes in a fixed tendency: *more "concave" learning rate curves for less training epochs*, which inspire the cosine-power schedule in following form.

$$\text{Cosine-power} : \eta_t = \eta_{\min} + (\eta_0 - \eta_{\min}) \left[ \frac{1}{2}(1 + \cos(\frac{t}{t_{\max}}\pi)) \right]^{\alpha}$$

Results in Table 11 show the schedulings' performance with $\alpha = 0.5/1/2$, which are denoted as $\sqrt{\text{Cosine}}$/Cosine/Cosine$^2$ respectively. Notice that the best scheduler gradually moves from small $\alpha$ to larger $\alpha$ when the number of epochs increases. For #epoch=270, since the number of epochs is large enough to make model converge, it is reasonable that the accuracy gap between all schedulers is small.

For experiments on ImageNet, we use ResNet-18 trained via SGD without momentum, batch size 256 and weight decay $wd = 10^{-4}$. Since no momentum is used, the initial learning rate is set to $\eta_0 = 1.0$ instead of $\eta_0 = 0.1$. The hyperparameters $\eta_{\min}$ is set to be 0 for all cosine-power scheduler. As for the dataset, we use the common ILSVRC 2012 dataset, which contains 1000 classes, around 1.2M images for training and 50,000 images for validation.

Table 11: Cosine-power Decay on ImageNet: training losses and validation accuracy (%) of different schedulings for ResNet-18 over 3 trials. Settings #Epoch$\geq$ 90 only have 1 trial due to constraints of resource and time.

| #Epoch | Schedule | Training loss | Top-1 validation acc (%) | Top-5 validation acc (%) |
|---|---|---|---|---|
| 1 | $\sqrt{\text{Cosine}}$ | **5.4085$\pm$0.0080** | **30.01$\pm$0.21** | **55.26$\pm$0.33** |
| | Cosine | 5.4330$\pm$0.0106 | 26.43$\pm$0.31 | 50.85$\pm$0.43 |
| | Cosine$^2$ | 5.4939$\pm$0.0157 | 21.81$\pm$0.21 | 44.53$\pm$0.09 |
| 5 | $\sqrt{\text{Cosine}}$ | 2.9515$\pm$0.0057 | **57.27$\pm$0.15** | **80.71$\pm$0.12** |
| | Cosine | **2.8389$\pm$0.0061** | 55.67$\pm$0.08 | 79.46$\pm$0.16 |
| | Cosine$^2$ | 2.9160$\pm$0.0099 | 52.75$\pm$0.20 | 77.11$\pm$0.08 |
| 30 | $\sqrt{\text{Cosine}}$ | 2.1739$\pm$0.0046 | 67.56$\pm$0.03 | 87.82$\pm$0.09 |
| | **Cosine** | **2.0402$\pm$0.0031** | **67.97$\pm$0.10** | **88.12$\pm$0.03** |
| | Cosine$^2$ | 2.0525$\pm$0.0032 | 67.41$\pm$0.05 | 87.70$\pm$0.10 |
| 90 | $\sqrt{\text{Cosine}}$ | 1.9056 | 69.85 | 89.46 |
| | **Cosine** | 1.7676 | **70.46** | **89.75** |
| | Cosine$^2$ | **1.7403** | 70.42 | 89.69 |
| 270 | $\sqrt{\text{Cosine}}$ | 1.7178 | 71.37 | 90.31 |
| | Cosine | 1.5756 | **71.93** | 90.33 |
| | **Cosine$^2$** | **1.5250** | 71.69 | **90.37** |

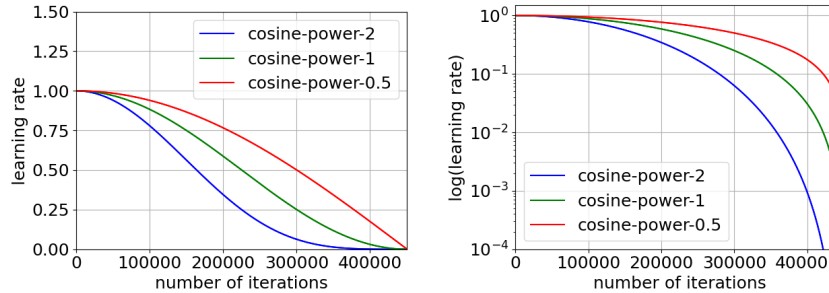

Figure 7: Learning rate curve of three cosine-power schedulers. Top: original scale; Bottom: log scale.

## A.8 FULL FIGURES FOR EIGENCURVE EXPERIMENTS IN SECTION 4.2

Please refer to Figure 8, 9, 10, 11, 12 and 13.

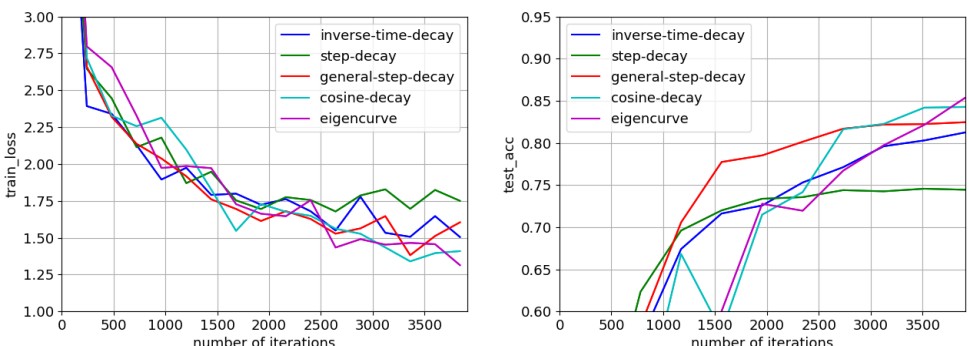

Figure 8: CIFAR-10 results for ResNet-18, with #Epoch = 10. Left: training losses. Right: test accuracy.

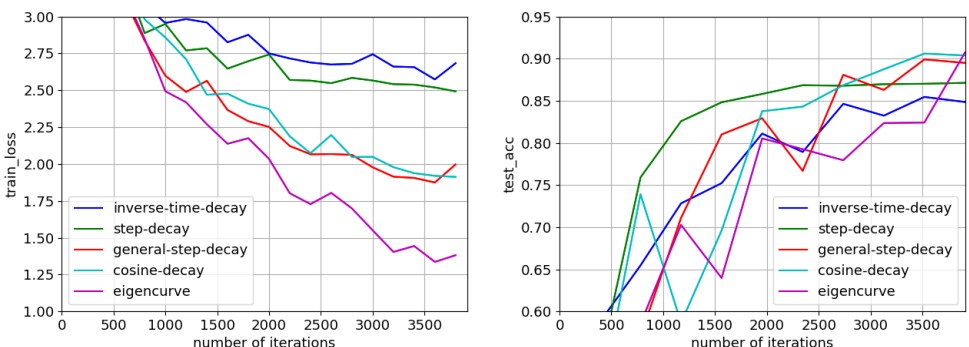

Figure 9: CIFAR-10 results for GoogLeNet, with #Epoch = 10. Left: training losses. Right: test accuracy.

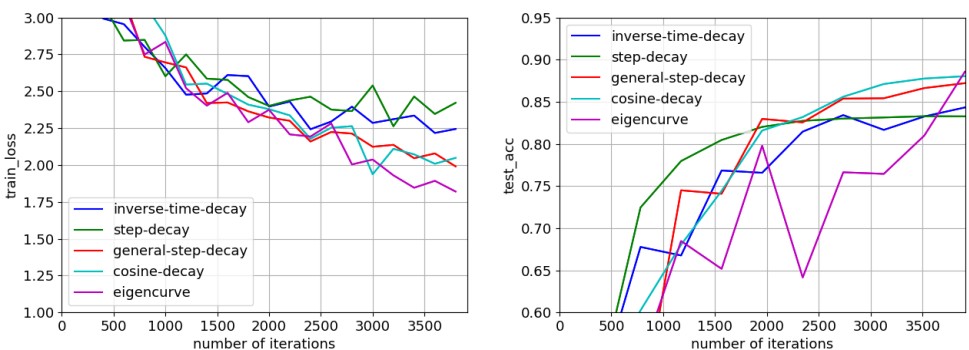

Figure 10: CIFAR-10 results for VGG16, with #Epoch = 10. Left: training losses. Right: test accuracy.

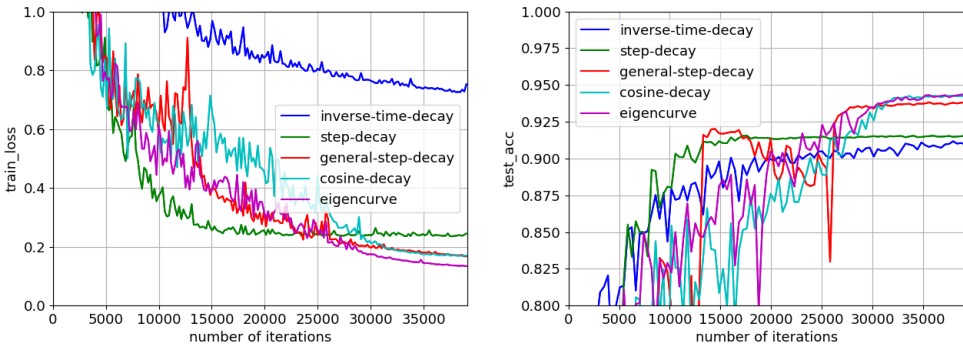

Figure 11: CIFAR-10 results for ResNet-18, with #Epoch = 100. Left: training losses. Right: test accuracy.

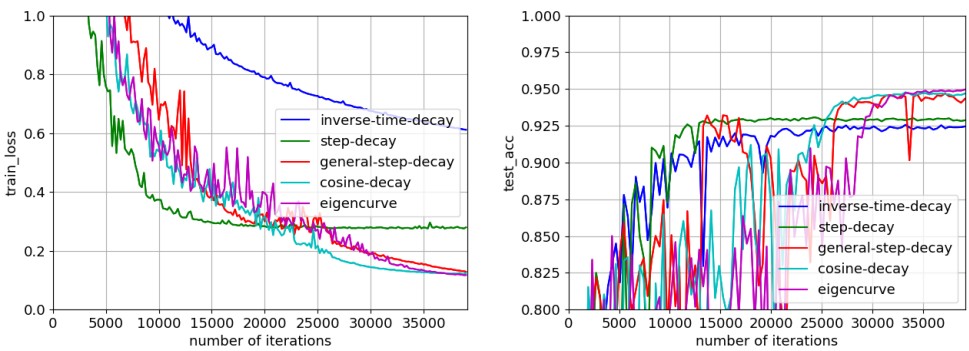

Figure 12: CIFAR-10 results for GoogLeNet, with #Epoch = 100. Left: training losses. Right: test accuracy.

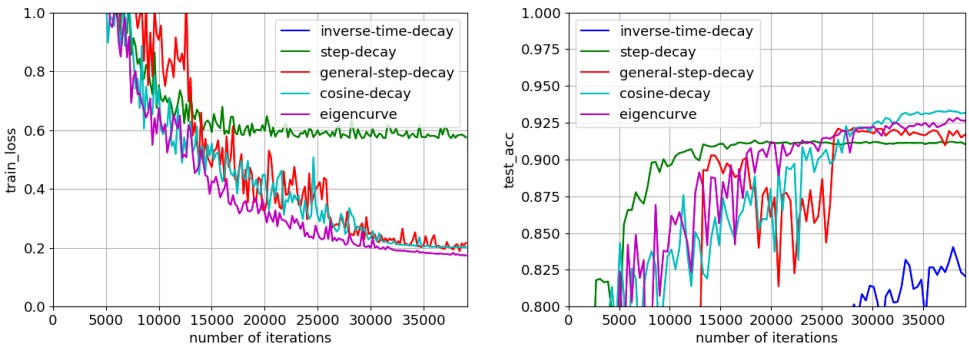

Figure 13: CIFAR-10 results for VGG16, with #Epoch = 100. Left: training losses. Right: test accuracy.

# B DETAILED EXPERIMENTAL SETTINGS FOR IMAGE CLASSIFICATION ON CIFAR-10/CIFAR-100

## B.1 BASIC SETTINGS

As mentioned in the main paper, all models are trained with stochastic gradient descent (SGD), no momentum, batch size $128$ and weight decay $wd = 0.0005$. Furthermore, we perform a grid search to choose the best hyperparameters of all schedulers, with a validation set created from $5,000$

samples in the training set, i.e. one-tenth of the training set. The remaining $45,000$ samples are then used for training the model. After obtaining hyperparameters with the best validation accuracy, we train the model again with the full training set and test the trained model on test set, where 5 trials of experiments are conducted. The mean and standard deviation of the test results are reported.

Here the grid search explores hyperparameters $\eta_0 \in \{1.0, 0.6, 0.3, 0.2, 0.1\}$ and $\eta_{\min} \in \{0.01, 0.001, 0.0001, 0, \text{"UNRESTRICTED"}\}$, where $\eta_0$ denotes the initial learning rate and $\eta_{\min}$ stands for the learning rate of last iteration. "UNRESTRICTED" denotes the case where $\eta_{\min}$ is not set, which is useful for `eigencurve`, who can decide the learning rate curve without setting $\eta_{\min}$. Given $\eta_0$ and $\eta_{\min}$, we adjust all schedulers as follows. For inverse time decay, the hyperparameter $\gamma$ is computed accordingly based on $\eta_0$ and $\eta_{\min}$. For cosine decay, $\eta_0$ and $\eta_{\min}$ is directly used, with no restart adopted. For general step decay, we search the interval number in $\{3, 4, 5, \lfloor \log T \rfloor, \lfloor \log T \rfloor + 1\}$ and decay factor in $\{2, 5, 10\}$. For step decay proposed in Ge et al. (2019), the interval number is fixed to be $\lfloor \log T \rfloor$, along with a decay factor 2. For `eigencurve`, two major modifications are made to make it more suitable for practical settings:

$$\eta_t = \frac{1/L}{1 + \frac{1}{\kappa}\sum_{j=1}^{i-1}\Delta_j 2^{j-1} + \frac{2^{i-1}}{\kappa}(t - t_{i-1})} = \frac{\boldsymbol{\eta_0}}{1 + \frac{1}{\kappa}\sum_{j=1}^{i-1}\Delta_j \boldsymbol{\beta}^{j-1} + \frac{\boldsymbol{\beta}^{i-1}}{\kappa}(t - t_{i-1})}.$$

Here we change $1/L$ to $\eta_0$ so that it is possible to adjust the initial learning rate of `eigencurve`. We also change the fixed constant 2 to a general constant $\beta > 1$, which is aimed at making the learning rate curve smoother. The learning rate curve of `eigencurve` is then linearly scaled to match the given $\eta_{\min}$.

Notice that the learning rate $\eta_0$ can be larger than $1/L$, while the loss still does not explode. There are several explanations for this phenomenon. First, in basic non-smooth analysis of GD and SGD with inverse time decay scheduler, the learning rate can be larger than $1/L$ if the gradient norm is bounded (Shamir & Zhang, 2012). Second, deep learning has a non-convex loss landscape, especially when the parameter is far away from the optima. Hence it is common to use larger learning rate at first. As long as the loss does not explode, it is okay. So we still include large learning rate $\eta_0$ in our grid search process.

## B.2 SETTINGS FOR EIGENCURVE

In addition, for our `eigencurve` scheduler, we use PyHessian (Yao et al., 2020) to generate Hessian matrix's eigenvalue distribution for all models. The whole process consists of three phases, which are illustrated as follows.

**1) Training the model**  Almost all CNN models on CIFAR-10 have non-convex objectives, thus the Hessian's eigenvalue distributions are different for different parameters. Normally, we want the this distribution to reflect the overall tendency of most parts of the training process. According to the phenomenon demonstrated in Appendix E, figure A.11-A.17 of Yao et al. (2020), the eigenvalue distribution of ResNet's Hessian presents similar tendency after training 30 epochs, which suggests that the Hessian's eigenvalue distribution can be used after sufficient training.

In all CIFAR-10 experiments, we use the Hessian's eigenvalue distribution of models after training 180 epochs. Since the goal here is to sufficiently train the model, not to obtain good performance, common baseline settings are adopted for training. For all models used for eigenvalue distribution estimation, we adopt SGD with momentum $= 0.9$, batch size 128, weight decay $wd = 0.0005$ and initial learning rate 0.1. On top of that, we use step decay, which decays the learning rate by a factor of 10 at epoch 80 and 120. All of them are default settings of the PyHessian code (https://github.com/amirgholami/PyHessian/blob/master/training.py, commit: f4c3f77).

ImageNet adopts a similar setting, with training epochs being 90, SGD with momentum $= 0.9$, batch size 256, weight decay $wd = 0.0001$, inital learning rate 0.1 and step decay schedule decays learning rate by a factor of 10 at epoch 30 and 60.

**2) Estimating Hessian matrix's eigenvalue distribution for the trained model**  After obtaining the checkpoint of a sufficiently trained model, we then run PyHessian to estimate the Hessian's

eigenvalue distribution for that checkpoint. The goal here is to obtain the Hessian's eigenvalue distribution with sufficient precision. To be more specific, the length of intervals around each estimated eigenvalue. PyHessian estimates the eigenvalue spectral density (ESD) of a model's Hessian, in other words, the output is a list of eigenvalue intervals, along with the density of each interval, where the whole density adds up to $1$. Precision means the interval length here.

It is natural that the estimation precision is related to the complexity of the PyHessian algorithm, e.g. the better precision it yields, the more time and space it consumes. More specifically, the algorithm has a time complexity of $\mathcal{O}(Nn_v^2d)$ and space complexity $\mathcal{O}(Bd + n_vd)$, where $d$ is the number of model parameters, $N$ is the number of samples used for estimating the ESD, $B$ is the batch size and $n_v$ is the iteration number of Stochastic Lanczos Quadrature used in PyHessian, which controls the estimation precision (see Algorithm 1 of Yao et al. (2020)).

In our experiments, we use $n_v = 5000$ for ResNet-18 and $n_v = 3000$ for GoogLeNet/VGG16, which gives an eigenvalue distribution estimation with precision around $10^{-5}$ to $10^{-4}$. $N$ and $B$ are both set to $200$ due to GPU memory constraint, i.e. we use one mini-batch to estimate the eigenvalue distribution. It turns out that this one-batch estimation is good enough and yields similar results to full dataset settings shown in Yao et al. (2020).

However, space complexity is still a bottleneck here. Due to the large number of $n_v$ and space complexity $\mathcal{O}(Bd + n_vd)$ of PyHessian, the value of $d$ cannot be very large. In practice, with a NVIDIA GeForce 2080 Ti GPU, which has around 11GB memory, the maximum acceptable parameter number $d$ is around $200K - 400K$. This implies that the model has to be compressed. In our experiments, we reduce the number of channels by a factor of $C$ for all models. For ResNet-18, $C = 16$. For GoogLeNet, $C = 4$. For VGG16, $C = 8$. Notice that those compressed models are only used for eigenvalue distribution estimation. In experiments of comparing different scheduling, we still use the original model with no compression.

One may refer to `https://github.com/opensource12345678/why_cosine_works/tree/main/eigenvalue_distribution` for generated eigenvalue distributions.

**3) Generating `eigencurve` scheduler with the estimated eigenvalue distribution**    After obtaining the eigenvalue distribution, we do a preprocessing before plug it into our `eigencurve` scheduler.

First, we notice that there are negative eigenvalues in the final distribution. Theoretically, if the parameter is right at the optimal point, no negative eigenvalues should exist for Hessian matrix. Thus we conjecture that those negative eigenvalues are caused by the fact that the model is closed to optima $w_*$, but not exactly at that point. Furthermore, the estimation precision loss can be another cause. In fact, most of those negative eigenvalues are small, e.g. $98.6\%$ of those negative eigenvalues lie in $[-0.1, 0)$, and can be generally ignored without much loss. In our case, we set them to their absolute values.

Second, for a given weight decay value $wd$, we need to take the implicit L2 regularization into account, since it affects the Hessian matrix as well. Therefore, for all eigenvalues after the first step, we add $wd$ to them.

After preprocessing, we plug the eigenvalue distribution into our `eigencurve` scheduler and generates the exact form of `eigencurve`.

$$\eta_t = \frac{1/L}{1 + \frac{1}{\kappa}\sum_{j=1}^{i-1}\Delta_j 2^{j-1} + \frac{2^{i-1}}{\kappa}(t - t_{i-1})} = \frac{\eta_0}{1 + \frac{1}{\kappa}\sum_{j=1}^{i-1}\Delta_j\beta^{j-1} + \frac{\beta^{i-1}}{\kappa}(t - t_{i-1})}$$

For experiments with 100 epochs, we set $\beta = 1.000005$, so that the learning rate curve is much smoother. For experiments with 10 epochs, we set $\beta = 2.0$. In our experiments, $\beta$ serves as a fixed constant, not hyperparameters. So no hyperparameter search is conducted on $\beta$. One can do that in practice though, if computation resource allows.

### B.3 Compute Resource and Implementation Details

All the code for results in main paper can be found in `https://github.com/opensource12345678/why_cosine_works/tree/main`, which is released under the MIT license.

All experiments on CIFAR-10/CIFAR-100 are conducted on a single NVIDIA GeForce 2080 Ti GPU, where ResNet-18/GoogLeNet/VGG16 takes around 20mins/90mins/40mins to train 100 epochs, respectively. High-precision eigenvalue distribution estimation, e.g. $n_v \geq 3000$, requires around 1-2 days to complete, but this is no longer necessary given the released results.

The ResNet-18 model is implemented in Tensorflow 2.0. We use tensorflow-gpu 2.3.1 in our code. The GoogLeNet and VGG16 model is implemented in Pytorch, specifically, 1.7.0+cu101.

### B.4 License of PyHessian

According to `https://github.com/amirgholami/PyHessian/blob/master/LICENSE`, PyHessian (Yao et al., 2020) is released under the MIT License.

## C Detailed Experimental Settings for Image Classification on ImageNet

One may refer to `https://www.image-net.org/download` for specific terms of access for ImageNet. The dataset can be downloaded from `https://image-net.org/challenges/LSVRC/2012/2012-downloads.php`, with training set being "Training images (Task 1 & 2)" and validation set being "Validation images (all tasks)". Notice that registration and verification of institute is required for successful download.

ResNet-18 experiments on ImageNet are conducted on two NVIDIA GeForce 2080 Ti GPUs with data parallelism, while ResNet-50 experiments are conducted on 4 GPUs in a similar fashion. Both models take around 2 days to train 90 epochs, about 20mins-30mins per epoch. Those ResNet models on ImageNet are implemented in Pytorch, specifically, 1.7.0+cu101.

## D Important Propositions and Lemmas

**Proposition 3.** *Letting $f(x)$ be a monotonically increasing function in the range $[t_0, \tilde{t}]$, then it holds that*

$$\sum_{k=t_0}^{\tilde{t}-1} f(k) \leq \int_{t_0}^{\tilde{t}} f(x) \, dx. \tag{D.1}$$

*If $f(x)$ is monotonically decreasing in the range $[t_0, \tilde{t}]$, then it holds that*

$$\int_{t_0}^{\tilde{t}} f(x) \, dx \leq \sum_{k=t_0}^{\tilde{t}-1} f(k) \leq \sum_{k=t_0}^{\tilde{t}} f(k) \leq f(t_0) + \int_{t_0}^{\tilde{t}} f(x) \, dx. \tag{D.2}$$

**Lemma 2.** *Function $f(x) = \exp(-\alpha x)x^2$ with $0 < \alpha$ and $x \in (0, 1]$ is monotone decreasing in the range $x \in (\frac{2}{\alpha}, +\infty)$ and monotone increasing in the range $x \in [0, \frac{2}{\alpha}]$.*

*Proof.* We can obtain the derivative of $f(x)$ as

$$\nabla f(x) = x \exp(-\alpha x)(2 - \alpha x).$$

Thus, it holds that $\nabla f(x) \geq 0$ when $x \in [0, \frac{2}{\alpha}]$. This implies that $f(x)$ is monotone increasing when $x \in [0, \frac{2}{\alpha}]$. Similarly, we can obtain that $f(x)$ is monotone decreasing when $x \in (\frac{2}{\alpha}, +\infty)$. $\qquad\square$

**Lemma 3.** *It holds that*

$$\int \exp(-\alpha x)x \, dx = -\alpha^{-1}(x \exp(-\alpha x) + \alpha^{-1} \exp(-\alpha x)). \tag{D.3}$$

*Proof.*

$$\int \exp(-\alpha x)x \, dx = -\alpha^{-1} \int x \, d\exp(-\alpha x) = -\alpha^{-1}(x \exp(-\alpha x) - \int \exp(-\alpha x)dx)$$

$$= -\alpha^{-1}(x \exp(-\alpha x) + \alpha^{-1} \exp(-\alpha x)).$$

□

## E   PROOF OF SECTION 2

*Proof of Proposition 1.*   By iteratively applying Eqn. (2.1), we can obtain that

$$\mathbb{E}\left[\lambda_j \left(w_{t+1,j} - w_{*,j}\right)^2\right] \overset{(2.1)}{\leq} \prod_{i=1}^t (1 - \eta_{i,j}\lambda_j)^2 \cdot \lambda_j(w_{1,j} - w_{*,j})^2$$

$$+ \lambda_j^2 \sigma^2 \cdot \sum_{k=1}^t \prod_{i=k+1}^t (1 - \eta_{i,j}\lambda_j)^2 \eta_{k,j}^2$$

$$= \frac{\lambda_j(w_{1,j} - w_{*,j})^2}{(t+1)^2} + \sigma^2 \cdot \sum_{k=1}^t \frac{(k+1)^2}{(t+1)^2} \cdot \frac{1}{(k+1)^2}$$

$$= \frac{\lambda_j(w_{1,j} - w_{*,j})^2}{(t+1)^2} + \frac{t}{(t+1)^2} \cdot \sigma^2.$$

Summing up each coordinate, we can obtain the result.   □

*Proof of Proposition 2.*   Let us denote $\sigma_j^2 = \lambda_j^2 \sigma^2$. By Eqn. (2.1), we can obtain that

$$\mathbb{E}\left[\lambda_j \left(w_{t+1,j} - w_{*,j}\right)^2\right]$$

$$\leq \Pi_{i=1}^t (1 - \eta_{i,j}\lambda_j)^2 \cdot \lambda_j(w_{1,j} - w_{*,j})^2 + \sum_{k=1}^t \Pi_{i=k}(1 - \eta_{i,j}\lambda_j)\eta_{k,j}^2\sigma_j^2$$

$$\leq \exp\left(-2\sum_{i=1}^t \eta_{i,j}\lambda_j\right) \cdot \lambda_j(w_{1,j} - w_{*,j})^2 + \sum_{k=1}^t \exp\left(-2\sum_{i=k}^t \eta_{i,j}\lambda_j\right) \eta_{k,j}^2\sigma_j^2$$

$$= \exp\left(-2\lambda_j \sum_{i=1}^t \frac{1}{L + \mu i}\right) \cdot \lambda_j(w_{1,j} - w_{*,j})^2 + \sum_{k=1}^t \exp\left(-2\lambda_j \sum_{i=k}^t \frac{1}{L + \mu i}\right) \eta_{k,j}^2\sigma_j^2$$

$$\leq \exp\left(\frac{2\lambda_j}{\mu}\ln\left(\frac{L+\mu}{L+\mu t}\right)\right) \cdot \lambda_j(w_{1,j} - w_{*,j})^2 + \sum_{k=1}^t \exp\left(\frac{2\lambda_j}{\mu}\ln\left(\frac{L+\mu k}{L+\mu t}\right)\right) \cdot \frac{\sigma_j^2}{(L+\mu k)^2}$$

$$= \left(\frac{L+\mu}{L+\mu t}\right)^{\frac{2\lambda_j}{\mu}} \cdot \lambda_j(w_{1,j} - w_{*,j})^2 + \sum_{k=1}^t \frac{(L+\mu k)^{\left(\frac{2\lambda_j}{\mu}-2\right)}}{(L+\mu t)^{\frac{2\lambda_j}{\mu}}} \cdot \sigma_j^2$$

$$\leq \left(\frac{L+\mu}{L+\mu t}\right)^{\frac{2\lambda_j}{\mu}} \cdot \lambda_j(w_{1,j} - w_{*,j})^2 + \frac{\sigma_j^2}{2\lambda_j - \mu}\frac{(L+\mu t)^{\frac{2\lambda_j}{\mu}-1}}{(L+\mu t)^{\frac{2\lambda_j}{\mu}}} + \frac{\sigma_j^2}{(L+\mu t)^2}$$

$$= \left(\frac{L+\mu}{L+\mu t}\right)^{\frac{2\lambda_j}{\mu}} \cdot \lambda_j(w_{1,j} - w_{*,j})^2 + \frac{1}{2\lambda_j - \mu} \cdot \frac{1}{L+\mu t} \cdot \sigma_j^2 + \frac{\sigma_j^2}{(L+\mu t)^2}.$$

The third inequality is because function $F(x) = 1/(L + \mu x)$ is monotone decreasing in the range $[1, \infty)$, and it holds that

$$\sum_{i=k}^t \frac{1}{L + \mu i} \geq \int_{i=k}^t \frac{1}{L + \mu i} \, di = \frac{1}{\mu}\ln\left(\frac{L+\mu t}{L+\mu k}\right).$$

The last inequality is because function $F(x) = (L + \mu x)^{2\lambda_j/\mu - 2}$ is monotone increasing in the range $[0, \infty)$, and it holds that

$$\sum_{k=1}^{t}(L + \mu k)^{\frac{2\lambda_j}{\mu} - 2} \leq \int_{k=1}^{t}(L + \mu k)^{\frac{2\lambda_j}{\mu} - 2}dk + (L + \mu t)^{\frac{2\lambda_j}{\mu} - 2}$$

$$= \frac{1}{\mu\left(\frac{2\lambda_j}{\mu} - 1\right)} \cdot \left((L + \mu t)^{\frac{2\lambda_j}{\mu} - 1} - (L + \mu)^{\frac{2\lambda_j}{\mu} - 1}\right) + (L + \mu t)^{\frac{2\lambda_j}{\mu} - 2}$$

$$< \frac{1}{2\lambda_j - \mu}(L + \mu t)^{\frac{2\lambda_j}{\mu} - 1} + (L + \mu t)^{\frac{2\lambda_j}{\mu} - 2}.$$

By $\mu \leq \lambda_j$, $\sigma_j^2 = \lambda_j^2\sigma^2$, and summing up from $i = 1$ to $d$, we can obtain the result. $\qquad\square$

## F  PRELIMINARIES

**Lemma 4.** *Let objective function $f(x)$ be quadratic. Running SGD for $T$-steps starting from $w_0$ and a learning rate sequence $\{\eta_t\}_{t=1}^{T}$, the final iterate $w_{T+1}$ satisfies*

$$\mathbb{E}\left[(w_{T+1} - w_*)^{\top}H(w_{T+1} - w_*)\right]$$
$$= \mathbb{E}\left[(w_0 - w_*)^{\top} \cdot P_T \ldots P_0 H P_0 \ldots P_T \cdot (w_0 - w_*)\right] \tag{F.1}$$
$$+ \sum_{\tau=0}^{T}\mathbb{E}\left[\eta_\tau^2 n_\tau^{\top} \cdot P_T \ldots P_{\tau+1}H P_{\tau+1} \ldots P_T \cdot n_\tau\right],$$

*where $P_t = I - \eta_t H$.*

*Proof.*  Reformulating Eqn. (1.5), we have

$$w_{t+1} - w_* = w_t - w_* - \eta_t(H(\xi)w_t - b(\xi))$$
$$= w_t - w_* - \eta_t(Hw_t - b) + \eta_t(Hw_t - b - (H(\xi)w_t - b(\xi)))$$
$$= w_t - w_* - \eta_t(Hw_t - b - (Hw_* - b)) + \eta_t(Hw_t - b - (H(\xi)w_t - b(\xi)))$$
$$= (I - \eta_t H)(w_t - w_*) + \eta_t n_t$$
$$= P_t(w_t - w_*) + \eta_t n_t.$$

Thus, we can obtain that

$$w_{t+1} - w_* = P_t \ldots P_0(w_0 - w_*) + \sum_{\tau=0}^{t}P_t \ldots P_{\tau+1}\eta_\tau n_\tau. \tag{F.2}$$

We can decompose above stochastic process associated with SGD's update into two simpler processes as follows:

$$w_{t+1}^b - w_* = P_t(w_t^b - w_*), \quad \text{and} \quad w_{t+1}^v - w_* = P_t(w_t^v - w_*) + \eta_t n_t, \text{ with } w_0^v = w_*, \tag{F.3}$$

which entails that

$$w_{t+1}^b - w_* = P_t \ldots P_0(w_0^b - w_*) = P_0 \ldots P_t(w_0^b - w_*) \tag{F.4}$$

$$w_{t+1}^v - w_* = \sum_{\tau=0}^{t}P_t \ldots P_{\tau+1}\eta_\tau n_\tau = \sum_{\tau=0}^{t}P_{\tau+1} \ldots P_t\eta_\tau n_\tau \tag{F.5}$$

$$\overset{\text{(F.2)}}{\Rightarrow} w_{t+1} - w_* = \left(w_{t+1}^b - w_*\right) + \left(w_{t+1}^v - w_*\right) \tag{F.6}$$

where the last equality in Eqn. (F.4) and Eqn. (F.5) is because the commutative property $P_t P_{t'} = (I - \eta_t H)(I - \eta_{t'} H) = (I - \eta_{t'} H)(I - \eta_t H) = P_{t'} P_t$ holds for $\forall t, t'$.

Thus, we have

$$
\mathbb{E}\left[(w_{T+1} - w_*)^\top H (w_{T+1} - w_*)\right]
$$

$$
\overset{\text{(F.6)}}{=} \mathbb{E}\left[(w_{T+1}^b - w_*)^\top H (w_{T+1}^b - w_*) + 2(w_{T+1}^v - w_*)^\top H (w_{T+1}^b - w_*) \right.
$$
$$
\left. + (w_{T+1}^v - w_*)^\top H (w_{T+1}^v - w_*)\right]
$$

$$
\overset{\text{(F.4)}}{=} \mathbb{E}\left[(w_0^b - w_*)^\top P_T \ldots P_0 H P_0 \ldots P_T (w_0^b - w_*) + 2(w_{T+1}^v - w_*)^\top H P_0 \ldots P_T (w_0^b - w_*) \right.
$$
$$
\left. + (w_{T+1}^v - w_*)^\top H (w_{T+1}^v - w_*)\right]
$$

$$
\overset{\text{(F.5)}}{=} \mathbb{E}\left[(w_0^b - w_*)^\top P_T \ldots P_0 H P_0 \ldots P_T (w_0^b - w_*) \right.
$$
$$
+ 2\left(\sum_{\tau=0}^T P_T \ldots P_{\tau+1}\eta_\tau n_\tau\right)^\top H P_0 \ldots P_T (w_0^b - w_*)
$$
$$
\left. + \left(\sum_{\tau=0}^T P_T \ldots P_{\tau+1}\eta_\tau n_\tau\right)^\top H \left(\sum_{\tau=0}^T P_T \ldots P_{\tau+1}\eta_\tau n_\tau\right)\right]
$$

$$
\overset{\mathbb{E}[n_\tau]=0}{=} \mathbb{E}\left[(w_0^b - w_*)^\top P_T \ldots P_0 H P_0 \ldots P_T (w_0^b - w_*)\right]
$$
$$
+ \mathbb{E}\left[\sum_{\tau=0,\tau'=0}^T \eta_\tau \eta_{\tau'} \cdot n_\tau^\top P_T \ldots P_{\tau+1} \cdot H \cdot P_{\tau'+1} \ldots P_T n_{\tau'}\right]
$$

$$
= \mathbb{E}\left[(w_0^b - w_*)^\top P_T \ldots P_0 H P_0 \ldots P_T (w_0^b - w_*)\right]
$$
$$
+ \sum_{\tau=0}^T \mathbb{E}\left[\eta_\tau^2 n_\tau^\top \cdot P_T \ldots P_{\tau+1} H P_{\tau+1} \ldots P_T \cdot n_\tau\right],
$$

where the last equality is because when $\tau$ and $\tau'$ are different, it holds that

$$
\mathbb{E}[n_\tau^\top \cdot P_T \ldots P_{\tau+1} H P_{\tau+1} \ldots P_T \cdot n_{\tau'}] = 0
$$

due to independence between $n_\tau$ and $n_{\tau'}$. $\qquad\square$

**Lemma 5.** *Given the assumption that $\mathbb{E}_\xi\left[n_t n_t^\top\right] \preceq \sigma^2 H$, then the variance term satisfies that*

$$
\sum_{\tau=0}^T \mathbb{E}\left[\eta_\tau^2 n_\tau^\top \cdot P_T \ldots P_{\tau+1} H P_{\tau+1} \ldots P_T \cdot n_\tau\right] \leq \sigma^2 \sum_{j=1}^d \lambda_j^2 \sum_{k=0}^T \eta_k^2 \prod_{i=k+1}^T (1 - \eta_i \lambda_j)^2, \quad \text{(F.7)}
$$

*where $P_t = I - \eta_t H$.*

*Proof.* Denote $A_\tau \triangleq P_T \ldots P_{\tau+1} H^{\frac{1}{2}}$, then

$$
A_\tau^\top = \left(P_T \ldots P_{\tau+1} H^{\frac{1}{2}}\right)^\top = \left(H^{\frac{1}{2}}\right)^\top P_{\tau+1}^\top \ldots P_T^\top = H^{\frac{1}{2}} P_{\tau+1} \ldots P_T, \quad \text{(F.8)}
$$

where the second equality is entailed by the fact that $H^{\frac{1}{2}}, P_{\tau+1}, \ldots, P_T$ are symmetric matrices.

Therefore, we have,

$$\sum_{\tau=0}^{T} \mathbb{E}\left[\eta_\tau^2 n_\tau^\top \cdot P_T \dots P_{\tau+1} H P_{\tau+1} \dots P_T \cdot n_\tau\right]$$

$$\overset{\text{(F.8)}}{=} \sum_{\tau=0}^{T} \mathbb{E}\left[\eta_\tau^2 n_\tau^\top A_\tau A_\tau^\top n_\tau\right] = \sum_{\tau=0}^{T} \eta_\tau^2 \mathbb{E}\left[\text{tr}\left(n_\tau^\top A_\tau A_\tau^\top n_\tau\right)\right] = \sum_{\tau=0}^{T} \eta_\tau^2 \mathbb{E}\left[\text{tr}\left(A_\tau^\top n_\tau n_\tau^\top A_\tau\right)\right]$$

$$= \sum_{\tau=0}^{T} \eta_\tau^2 \text{tr}\left(\mathbb{E}\left[A_\tau^\top n_\tau n_\tau^\top A_\tau\right]\right) = \sum_{\tau=0}^{T} \eta_\tau^2 \text{tr}\left(A_\tau^\top \mathbb{E}\left[n_\tau n_\tau^\top\right] A_\tau\right)$$

$$\leq \sigma^2 \cdot \sum_{\tau=0}^{T} \eta_\tau^2 \text{tr}\left(A_\tau^\top H A_\tau\right) = \sigma^2 \cdot \sum_{\tau=0}^{T} \eta_\tau^2 \text{tr}\left(A_\tau A_\tau^\top H\right)$$

$$= \sigma^2 \cdot \sum_{\tau=0}^{T} \eta_\tau^2 \text{tr}\left(P_T \dots P_{\tau+1} H P_{\tau+1} \dots P_T H\right)$$

$$= \sigma^2 \sum_{j=1}^{d} \lambda_j^2 \sum_{k=0}^{T} \eta_k^2 \prod_{i=k+1}^{T} (1 - \eta_i \lambda_j)^2,$$

where the third and sixth equality come from the cyclic property of trace, while the first inequality is because of the condition $\mathbb{E}_\xi\left[n_t n_t^\top\right] \preceq \sigma^2 H$, where

$$\forall x, \quad x^\top \mathbb{E}[n_\tau n_\tau^\top] x \leq \sigma^2 x^\top H x$$

$$\Rightarrow \quad \forall z, \quad z^\top A_\tau^\top \mathbb{E}[n_\tau n_\tau^\top] A_\tau z = (A_\tau z)^\top \mathbb{E}[n_\tau n_\tau^\top](A_\tau z) \leq \sigma^2 (A_\tau z)^\top H (A_\tau z) = \sigma^2 z^\top A_\tau^\top H A_\tau z$$

$$\Rightarrow \quad A_\tau^\top \mathbb{E}[n_\tau n_\tau^\top] A_\tau \preceq \sigma^2 A_\tau^\top H A_\tau$$

$$\Rightarrow \quad \text{tr}\left(A_\tau^\top \mathbb{E}[n_\tau n_\tau^\top] A_\tau\right) \leq \sigma^2 \text{tr}\left(A_\tau^\top H A_\tau\right).$$

$\square$

**Lemma 6.** *Letting $\lambda_{\tilde{j}}$ be the smallest positive eigenvalue of $H$, then the bias term satisfies that*

$$\mathbb{E}\left[(w_0 - w_*)^\top \cdot P_T \dots P_0 H P_0 \dots P_T \cdot (w_0 - w_*)\right]$$

$$\leq (w_0 - w_*)^\top H (w_0 - w_*) \cdot \exp\left(-2\lambda_{\tilde{j}} \sum_{k=0}^{T} \eta_k\right). \tag{F.9}$$

*Proof.* Letting $H = U\Lambda U^\top$ be the spectral decomposition of $H$ and $u_j$ be $j$-th column of $U$, we can obtain that

$$\mathbb{E}\left[(w_0 - w_*)^\top \cdot P_T \dots P_0 H P_0 \dots P_T \cdot (w_0 - w_*)\right]$$

$$= \sum_{j=1}^{d} \lambda_j \cdot (u_j^\top (w_0 - w_*))^2 \cdot \prod_{k=0}^{T} (1 - \eta_k \lambda_j)^2$$

$$\leq \sum_{j=1}^{d} \lambda_j \cdot (u_j^\top (w_0 - w_*))^2 \cdot \exp\left(-2\lambda_j \sum_{k=0}^{T} \eta_k\right).$$

Since $\lambda_{\tilde{j}}$ is the smallest positive eigenvalue of $H$, it holds that

$$\sum_{j=1}^{d} \lambda_j \cdot (u_j^\top (w_0 - w_*))^2 \cdot \exp\left(-2\lambda_j \sum_{k=0}^{T} \eta_k\right) \leq \sum_{j=1}^{d} \lambda_j \cdot (u_j^\top (w_0 - w_*))^2 \cdot \exp\left(-2\lambda_{\tilde{j}} \sum_{k=0}^{T} \eta_k\right)$$

$$= (w_0 - w_*)^\top H (w_0 - w_*) \cdot \exp\left(-2\lambda_{\tilde{j}} \sum_{k=0}^{T} \eta_k\right).$$

$\square$

## G  PROOF OF THEOREMS

**Lemma 7.** *Let learning rate $\eta_t$ is defined in Eqn. (3.3). Assuming $k \in [t_{i'-1}, t_{i'}]$ with $1 \le i' \le \tilde{i} \le T$, the sequence $\{\eta_t\}_{t=0}^T$ satisfies that*

$$\sum_{t=k}^{t_{\tilde{i}+1}-1} \eta_t \ge \sum_{i=i'+1}^{\tilde{i}+1} \frac{1}{2^{i-1}\mu} \ln \frac{\alpha_i}{\alpha_{i-1}} + \frac{1}{2^{i'-1}\mu} \ln \frac{\alpha_{i'}}{\alpha_{i'-1} + 2^{i'-1}\mu(k - t_{i'-1})}, \tag{G.1}$$

*where $\alpha_i$ is defined as*

$$\alpha_i \triangleq L + \mu \sum_{j=1}^{i} \Delta_j 2^{j-1} = \frac{1}{\eta_{t_i}}. \tag{G.2}$$

*Proof.* First, we divide learning rates into two groups: those who are guaranteed to cover a full interval and those who may not.

$$\sum_{t=k}^{t_{\tilde{i}+1}-1} \eta_t = \sum_{t=t_{i'}}^{t_{\tilde{i}+1}-1} \eta_t + \sum_{t=k}^{t_{i'}-1} \eta_t = \sum_{i=i'+1}^{\tilde{i}+1} \sum_{t=t_{i-1}}^{t_i-1} \eta_t + \sum_{t=k}^{t_{i'}-1} \eta_t$$

Furthermore, because $\eta_t$ is monotonically decreasing with respect to $t$, by Proposition 3, we have

$$\sum_{t=k}^{t_{\tilde{i}+1}-1} \eta_t \overset{(D.2)}{\ge} \sum_{i=i'+1}^{\tilde{i}+1} \int_{t_{i-1}}^{t_i} \eta_t dt + \int_{k}^{t_{i'}} \eta_t dt$$

$$\overset{(3.3)}{=} \sum_{i=i'+1}^{\tilde{i}+1} \int_{t_{i-1}}^{t_i} \frac{1}{L + \mu \sum_{j=1}^{i-1} \Delta_j 2^{j-1} + 2^{i-1}\mu(t - t_{i-1})} dt$$

$$+ \int_{k}^{t_{i'}} \frac{1}{L + \mu \sum_{j=1}^{i'-1} \Delta_j 2^{j-1} + 2^{i'-1}\mu(t - t_{i'-1})} dt$$

$$= \sum_{i=i'+1}^{\tilde{i}+1} \frac{1}{2^{i-1}\mu} \ln \frac{L + \mu \sum_{j=1}^{i-1} \Delta_j 2^{j-1} + 2^{i-1}\mu(t_i - t_{i-1})}{L + \mu \sum_{j=1}^{i-1} \Delta_j 2^{j-1}}$$

$$+ \frac{1}{2^{i'-1}\mu} \ln \frac{L + \mu \sum_{j=1}^{i'-1} \Delta_j 2^{j-1} + 2^{i'-1}\mu(t_{i'} - t_{i'-1})}{L + \mu \sum_{j=1}^{i'-1} \Delta_j 2^{j-1} + 2^{i'-1}\mu(k - t_{i'-1})}$$

$$= \sum_{i=i'+1}^{\tilde{i}+1} \frac{1}{2^{i-1}\mu} \ln \frac{L + \mu \sum_{j=1}^{i} \Delta_j 2^{j-1}}{L + \mu \sum_{j=1}^{i-1} \Delta_j 2^{j-1}}$$

$$+ \frac{1}{2^{i'-1}\mu} \ln \frac{L + \mu \sum_{j=1}^{i'} \Delta_j 2^{j-1}}{L + \mu \sum_{j=1}^{i'-1} \Delta_j 2^{j-1} + 2^{i'-1}\mu(k - t_{i'-1})}$$

$$= \sum_{i=i'+1}^{\tilde{i}+1} \frac{1}{2^{i-1}\mu} \ln \frac{\alpha_i}{\alpha_{i-1}} + \frac{1}{2^{i'-1}\mu} \ln \frac{\alpha_{i'}}{\alpha_{i'-1} + 2^{i'-1}\mu(k - t_{i'-1})}.$$

$\square$

**Lemma 8.** *Letting sequence $\{\alpha_i\}$ be defined in Eqn. (G.2), given $1 \le \tilde{i}$, it holds that*

$$\sum_{i=1}^{\tilde{i}+1} \left[ \prod_{j=i+1}^{\tilde{i}+1} \left( \frac{\alpha_{j-1}}{\alpha_j} \right)^{2^{\tilde{i}-j+2}} \right] \alpha_i^{-2^{\tilde{i}-i+2}} \sum_{k=t_{i-1}}^{t_i-1} (\alpha_{i-1} + 2^{i-1}\mu(k - t_{i-1}))^{2^{\tilde{i}-i+2}-2}$$

$$\le 2 \cdot \frac{1}{2^{\tilde{i}+1}\mu} \cdot \sum_{i=1}^{\tilde{i}+1} \left[ \prod_{j=i+1}^{\tilde{i}+1} \left( \frac{\alpha_{j-1}}{\alpha_j} \right)^{2^{\tilde{i}-j+2}} \right] \left( \alpha_i^{2^{\tilde{i}-i+2}-1} - \alpha_{i-1}^{2^{\tilde{i}-i+2}-1} \right) \alpha_i^{-2^{\tilde{i}-i+2}}. \tag{G.3}$$

*Proof.* Notice that $g(k) := (\alpha_{i-1} + 2^{i-1}\mu(k - t_{i-1}))^{2^{\tilde{i}-i+2}-2}$ is a monotonically increasing function, we have,

$$\sum_{i=1}^{\tilde{i}+1} \left[ \prod_{j=i+1}^{\tilde{i}+1} \left( \frac{\alpha_{j-1}}{\alpha_j} \right)^{2^{\tilde{i}-j+2}} \right] \alpha_i^{-2^{\tilde{i}-i+2}} \sum_{k=t_{i-1}}^{t_i-1} (\alpha_{i-1} + 2^{i-1}\mu(k - t_{i-1}))^{2^{\tilde{i}-i+2}-2}$$

$$\overset{(D.1)}{\leq} \sum_{i=1}^{\tilde{i}+1} \left[ \prod_{j=i+1}^{\tilde{i}+1} \left( \frac{\alpha_{j-1}}{\alpha_j} \right)^{2^{\tilde{i}-j+2}} \right] \alpha_i^{-2^{\tilde{i}-i+2}} \int_{t_{i-1}}^{t_i} (\alpha_{i-1} + 2^{i-1}\mu(t - t_{i-1}))^{2^{\tilde{i}-i+2}-2} dt$$

$$= \sum_{i=1}^{\tilde{i}+1} \left[ \prod_{j=i+1}^{\tilde{i}+1} \left( \frac{\alpha_{j-1}}{\alpha_j} \right)^{2^{\tilde{i}-j+2}} \right] \alpha_i^{-2^{\tilde{i}-i+2}} ((2^{\tilde{i}-i+2} - 1) \cdot 2^{i-1}\mu)^{-1} \left( \alpha_i^{2^{\tilde{i}-i+2}-1} - \alpha_{i-1}^{2^{\tilde{i}-i+2}-1} \right)$$

$$= \sum_{i=1}^{\tilde{i}+1} \left[ \prod_{j=i+1}^{\tilde{i}+1} \left( \frac{\alpha_{j-1}}{\alpha_j} \right)^{2^{\tilde{i}-j+2}} \right] \alpha_i^{-2^{\tilde{i}-i+2}} \frac{1}{(2^{\tilde{i}+1} - 2^{i-1})\mu} \left( \alpha_i^{2^{\tilde{i}-i+2}-1} - \alpha_{i-1}^{2^{\tilde{i}-i+2}-1} \right)$$

$$\leq \sum_{i=1}^{\tilde{i}+1} \left[ \prod_{j=i+1}^{\tilde{i}+1} \left( \frac{\alpha_{j-1}}{\alpha_j} \right)^{2^{\tilde{i}-j+2}} \right] \alpha_i^{-2^{\tilde{i}-i+2}} \frac{1}{(2^{\tilde{i}+1} - 2^{\tilde{i}})\mu} \left( \alpha_i^{2^{\tilde{i}-i+2}-1} - \alpha_{i-1}^{2^{\tilde{i}-i+2}-1} \right)$$

$$= \sum_{i=1}^{\tilde{i}+1} \left[ \prod_{j=i+1}^{\tilde{i}+1} \left( \frac{\alpha_{j-1}}{\alpha_j} \right)^{2^{\tilde{i}-j+2}} \right] \alpha_i^{-2^{\tilde{i}-i+2}} \frac{1}{2^{\tilde{i}+1}\mu} \frac{1}{1 - \frac{1}{2}} \left( \alpha_i^{2^{\tilde{i}-i+2}-1} - \alpha_{i-1}^{2^{\tilde{i}-i+2}-1} \right)$$

$$= 2 \cdot \frac{1}{2^{\tilde{i}+1}\mu} \sum_{i=1}^{\tilde{i}+1} \left[ \prod_{j=i+1}^{\tilde{i}+1} \left( \frac{\alpha_{j-1}}{\alpha_j} \right)^{2^{\tilde{i}-j+2}} \right] \alpha_i^{-2^{\tilde{i}-i+2}} \left( \alpha_i^{2^{\tilde{i}-i+2}-1} - \alpha_{i-1}^{2^{\tilde{i}-i+2}-1} \right)$$

$$= 2 \cdot \frac{1}{2^{\tilde{i}+1}\mu} \sum_{i=1}^{\tilde{i}+1} \left[ \prod_{j=i+1}^{\tilde{i}+1} \left( \frac{\alpha_{j-1}}{\alpha_j} \right)^{2^{\tilde{i}-j+2}} \right] \left( \alpha_i^{2^{\tilde{i}-i+2}-1} - \alpha_{i-1}^{2^{\tilde{i}-i+2}-1} \right) \alpha_i^{-2^{\tilde{i}-i+2}}.$$

$\square$

**Lemma 9.** *Letting* $\{\alpha_i\}$ *be a positive sequence, given* $1 \leq \tilde{i}$, *it holds that*

$$\sum_{i=1}^{\tilde{i}+1} \left[ \prod_{j=i+1}^{\tilde{i}+1} \left( \frac{\alpha_{j-1}}{\alpha_j} \right)^{2^{\tilde{i}-j+2}} \right] \left( \alpha_i^{2^{\tilde{i}-i+2}-1} - \alpha_{i-1}^{2^{\tilde{i}-i+2}-1} \right) \alpha_i^{-2^{\tilde{i}-i+2}} \leq \alpha_{\tilde{i}+1}^{-1}. \tag{G.4}$$

*Proof.* First, we have

$$\sum_{i=1}^{\tilde{i}+1} \left[ \prod_{j=i+1}^{\tilde{i}+1} \left( \frac{\alpha_{j-1}}{\alpha_j} \right)^{2^{\tilde{i}-j+2}} \right] \left( \alpha_i^{2^{\tilde{i}-i+2}-1} - \alpha_{i-1}^{2^{\tilde{i}-i+2}-1} \right) \alpha_i^{-2^{\tilde{i}-i+2}}$$

$$= \sum_{i=1}^{\tilde{i}+1} \left[ \prod_{j=i+1}^{\tilde{i}+1} \left( \frac{\alpha_{j-1}}{\alpha_j} \right)^{2^{\tilde{i}-j+2}} \right] \left( \alpha_i^{-1} - \frac{\alpha_{i-1}^{2^{\tilde{i}-i+2}-1}}{\alpha_i^{2^{\tilde{i}-i+2}}} \right)$$

$$= \sum_{i=1}^{\tilde{i}+1} \left[ \prod_{j=i+1}^{\tilde{i}+1} \left( \frac{\alpha_{j-1}}{\alpha_j} \right)^{2^{\tilde{i}-j+2}} \right] \alpha_i^{-1} - \sum_{i=1}^{\tilde{i}+1} \left[ \prod_{j=i+1}^{\tilde{i}+1} \left( \frac{\alpha_{j-1}}{\alpha_j} \right)^{2^{\tilde{i}-j+2}} \right] \left( \frac{\alpha_{i-1}^{2^{\tilde{i}-i+2}-1}}{\alpha_i^{2^{\tilde{i}-i+2}}} \right)$$

$$= \alpha_{\tilde{i}+1}^{-1} + \sum_{i=1}^{\tilde{i}} \left[ \prod_{j=i+1}^{\tilde{i}+1} \left( \frac{\alpha_{j-1}}{\alpha_j} \right)^{2^{\tilde{i}-j+2}} \right] \alpha_i^{-1} - \sum_{i=1}^{\tilde{i}+1} \left[ \prod_{j=i+1}^{\tilde{i}+1} \left( \frac{\alpha_{j-1}}{\alpha_j} \right)^{2^{\tilde{i}-j+2}} \right] \left( \frac{\alpha_{i-1}^{2^{\tilde{i}-i+2}-1}}{\alpha_i^{2^{\tilde{i}-i+2}}} \right).$$

Furthermore, we reformulate the term $\sum_{i=1}^{\tilde{i}} \left[ \prod_{j=i+1}^{\tilde{i}+1} \left( \frac{\alpha_{j-1}}{\alpha_j} \right)^{2^{\tilde{i}-j+2}} \right] \alpha_i^{-1}$ as follows

$$\sum_{i=1}^{\tilde{i}} \left[ \prod_{j=i+1}^{\tilde{i}+1} \left( \frac{\alpha_{j-1}}{\alpha_j} \right)^{2^{\tilde{i}-j+2}} \right] \alpha_i^{-1} = \sum_{i=1}^{\tilde{i}} \left[ \prod_{j=i+2}^{\tilde{i}+1} \left( \frac{\alpha_{j-1}}{\alpha_j} \right)^{2^{\tilde{i}-j+2}} \right] \left( \frac{\alpha_i}{\alpha_{i+1}} \right)^{2^{\tilde{i}-i+1}} \alpha_i^{-1} \quad \text{(G.5)}$$

$$= \sum_{i=1}^{\tilde{i}} \left[ \prod_{j=i+2}^{\tilde{i}+1} \left( \frac{\alpha_{j-1}}{\alpha_j} \right)^{2^{\tilde{i}-j+2}} \right] \left( \frac{\alpha_i^{2^{\tilde{i}-i+1}-1}}{\alpha_{i+1}^{2^{\tilde{i}-i+1}}} \right) \quad \text{(G.6)}$$

$$\overset{i''=i+1}{=} \sum_{i''=2}^{\tilde{i}+1} \left[ \prod_{j=i''+1}^{\tilde{i}+1} \left( \frac{\alpha_{j-1}}{\alpha_j} \right)^{2^{\tilde{i}-j+2}} \right] \left( \frac{\alpha_{i''-1}^{2^{\tilde{i}-i''+2}-1}}{\alpha_{i''}^{2^{\tilde{i}-i''+2}}} \right) \quad \text{(G.7)}$$

$$\overset{i=i''}{=} \sum_{i=2}^{\tilde{i}+1} \left[ \prod_{j=i+1}^{\tilde{i}+1} \left( \frac{\alpha_{j-1}}{\alpha_j} \right)^{2^{\tilde{i}-j+2}} \right] \left( \frac{\alpha_{i-1}^{2^{\tilde{i}-i+2}-1}}{\alpha_i^{2^{\tilde{i}-i+2}}} \right). \quad \text{(G.8)}$$

Combining above results, we can obtain that

$$\sum_{i=1}^{\tilde{i}+1} \left[ \prod_{j=i+1}^{\tilde{i}+1} \left( \frac{\alpha_{j-1}}{\alpha_j} \right)^{2^{\tilde{i}-j+2}} \right] \left( \alpha_i^{2^{\tilde{i}-i+2}-1} - \alpha_{i-1}^{2^{\tilde{i}-i+2}-1} \right) \alpha_i^{-2^{\tilde{i}-i+2}}$$

$$= \alpha_{\tilde{i}+1}^{-1} + \sum_{i=2}^{\tilde{i}+1} \left[ \prod_{j=i+1}^{\tilde{i}+1} \left( \frac{\alpha_{j-1}}{\alpha_j} \right)^{2^{\tilde{i}-j+2}} \right] \left( \frac{\alpha_{i-1}^{2^{\tilde{i}-i+2}-1}}{\alpha_i^{2^{\tilde{i}-i+2}}} \right)$$

$$- \sum_{i=1}^{\tilde{i}+1} \left[ \prod_{j=i+1}^{\tilde{i}+1} \left( \frac{\alpha_{j-1}}{\alpha_j} \right)^{2^{\tilde{i}-j+2}} \right] \left( \frac{\alpha_{i-1}^{2^{\tilde{i}-i+2}-1}}{\alpha_i^{2^{\tilde{i}-i+2}}} \right)$$

$$= \alpha_{\tilde{i}+1}^{-1} - \left[ \prod_{j=2}^{\tilde{i}+1} \left( \frac{\alpha_{j-1}}{\alpha_j} \right)^{2^{\tilde{i}-j+2}} \right] \left( \frac{\alpha_0^{2^{\tilde{i}+1}-1}}{\alpha_1^{2^{\tilde{i}+1}}} \right)$$

$$\leq \alpha_{\tilde{i}+1}^{-1}.$$

$\square$

**Lemma 10.** *Letting us denote* $v_{t+1,j} \triangleq \sum_{k=0}^t \eta_k^2 \prod_{i=k+1}^t (1 - \eta_i \lambda_j)^2$ *with* $\eta_i$ *defined in Eqn.* (3.3), *for* $1 \leq t \leq t'$, *it holds that*

$$v_{t',j} \leq \max(v_{t,j}, \eta_t/\lambda_j). \quad \text{(G.9)}$$

*Proof.* If $v_{t+1,j} \leq \max(v_{t,j}, \eta_t/\lambda_j)$ holds for $\forall t \geq 1$, then it naturally follows that

$$v_{t',j} \leq \max \left( v_{t'-1,j}, \frac{\eta_{t'-1}}{\lambda_j} \right) \leq \max \left( v_{t'-2,j}, \frac{\eta_{t'-2}}{\lambda_j}, \frac{\eta_{t'-1}}{\lambda_j} \right)$$

$$\leq \dots$$

$$\leq \max \left( v_{t,j}, \frac{\eta_t}{\lambda_j}, \dots \frac{\eta_{t'-2}}{\lambda_j}, \frac{\eta_{t'-1}}{\lambda_j} \right)$$

$$= \max \left( v_{t,j}, \frac{\eta_t}{\lambda_j} \right)$$

where the last equality is entailed by the fact that $t \leq t'$ and $\eta_t$ defined in Eqn. (3.3) is monotonically decreasing. We then prove $v_{t+1,j} \leq \max(v_{t,j}, \eta_t/\lambda_j)$ holds for $\forall t \geq 1$.

For $\forall t \geq 1$, we have

$$
\begin{aligned}
v_{t+1,j} &= \sum_{k=0}^{t} \eta_k^2 \prod_{i=k+1}^{t} (1 - \eta_i \lambda_j)^2 \\
&= \eta_t^2 + \sum_{k=0}^{t-1} \eta_k^2 \prod_{i=k+1}^{t} (1 - \eta_i \lambda_j)^2 \\
&= \eta_t^2 + (1 - \eta_t \lambda_j)^2 \sum_{k=0}^{t-1} \eta_k^2 \prod_{i=k+1}^{t-1} (1 - \eta_i \lambda_j)^2 \\
&= \eta_t^2 + (1 - \eta_t \lambda_j)^2 v_{t,j}
\end{aligned}
\tag{G.10}
$$

1) If $v_{t+1,j} \leq v_{t,j}$, then it naturally follows $v_{t+1,j} \leq \max(v_{t,j}, \eta_t / \lambda_j)$.

2) If $v_{t+1,j} > v_{t,j}$, denote $a \triangleq (1 - \eta_t \lambda_j)^2, b \triangleq \eta_t^2$, we have $v_{t+1,j} = a v_{t,j} + b$, where $a \in [0, 1)$ and $b \geq 0$. It follows,

$$
\begin{aligned}
& v_{t+1,j} > v_{t,j} \\
\Rightarrow\quad & a v_{t,j} + b > v_{t,j} \\
\Rightarrow\quad & v_{t,j} < \frac{b}{1 - a} \\
\Rightarrow\quad & v_{t+1,j} = a v_{t,j} + b < a \cdot \frac{b}{1 - a} + b = \frac{b}{1 - a}
\end{aligned}
$$

Therefore,

$$
v_{t+1,j} < \frac{b}{1 - a} = \frac{\eta_t^2}{1 - (1 - \eta_t \lambda_j)^2} < \frac{\eta_t^2}{1 - (1 - \eta_t \lambda_j)} = \frac{\eta_t}{\lambda_j} \leq \max\left(v_{t,j}, \frac{\eta_t}{\lambda_j}\right),
$$

where the second inequality is entailed by the fact that $1 - \eta_t \lambda_j \in [0, 1)$.

$\square$

**Lemma 11.** *Letting $v_{t,j}$ be defined as Lemma 10 and index $\tilde{i}$ satisfy $\lambda_j \in [\mu \cdot 2^{\tilde{i}}, \mu \cdot 2^{\tilde{i}+1})$, then $v_{\tilde{i}+1,j}$ has the following property*

$$
v_{t_{\tilde{i}+1},j} \leq 15 \cdot \frac{\eta_{t_{\tilde{i}+1}}}{\lambda_j}.
\tag{G.11}
$$

*Proof.* By the fact that $(1 - x) \leq \exp(-x)$, we have

$$
v_{t+1,j} \leq \sum_{k=0}^{t} \exp\left(-2 \sum_{t'=k+1}^{t} \eta_{t'} \lambda_j\right) \eta_k^2.
$$

Setting $t = t_{\tilde{i}+1} - 1$ in above equation, we have

$$
v_{t_{\tilde{i}+1},j} \leq \sum_{k=0}^{t_{\tilde{i}+1}-1} \exp\left(-2 \sum_{t'=k+1}^{t_{\tilde{i}+1}-1} \eta_{t'} \lambda_j\right) \eta_k^2.
\tag{G.12}
$$

Now we bound the variance term. First, we have

$$
\sum_{k=0}^{t_{\tilde{i}+1}-1} \exp\left(-2 \sum_{t=k+1}^{t_{\tilde{i}+1}-1} \eta_t \lambda_j\right) \eta_k^2 = \sum_{k=0}^{t_{\tilde{i}+1}-1} \exp\left(-2 \sum_{t=k}^{t_{\tilde{i}+1}-1} \eta_t \lambda_j\right) \exp(2\eta_k \lambda_j) \eta_k^2
$$

$$
\leq \sum_{k=0}^{t_{\tilde{i}+1}-1} \exp\left(-2 \sum_{t=k}^{t_{\tilde{i}+1}-1} \eta_t \lambda_j\right) \exp\left(\frac{2\lambda_j}{L}\right) \eta_k^2
$$

$$
\leq \exp(2) \cdot \sum_{k=0}^{t_{\tilde{i}+1}-1} \exp\left(-2 \sum_{t=k}^{t_{\tilde{i}+1}-1} \eta_t \lambda_j\right) \eta_k^2,
$$

where the first inequality is because $\eta_k \leq 1/L$. Hence, we can obtain

$$
\sum_{k=0}^{t_{\tilde{i}+1}-1} \exp\left(-2 \sum_{t=k+1}^{t_{\tilde{i}+1}-1} \eta_t \lambda_j\right) \eta_k^2
$$

$$
\leq \exp(2) \cdot \sum_{k=0}^{t_{\tilde{i}+1}-1} \exp\left(-2 \sum_{t=k}^{t_{\tilde{i}+1}-1} \eta_t \lambda_j\right) \eta_k^2
$$

$$
= \exp(2) \cdot \sum_{i=1}^{\tilde{i}+1} \sum_{k=t_{i-1}}^{t_i-1} \exp\left(-2 \sum_{t=k}^{t_{\tilde{i}+1}-1} \eta_t \lambda_j\right) \eta_k^2.
$$

Furthermore, combining with Eqn. (G.1) and the condition $\lambda_j \in [\mu \cdot 2^{\tilde{i}}, \mu \cdot 2^{\tilde{i}+1})$, we can obtain

$$
\sum_{i=1}^{\tilde{i}+1} \sum_{k=t_{i-1}}^{t_i-1} \exp\left(-2 \sum_{t=k}^{t_{\tilde{i}+1}-1} \eta_t \lambda_j\right) \eta_k^2 \leq \sum_{i=1}^{\tilde{i}+1} \sum_{k=t_{i-1}}^{t_i-1} \exp\left(-2 \sum_{t=k}^{t_{\tilde{i}+1}-1} \eta_t \mu \cdot 2^{\tilde{i}}\right) \eta_k^2
$$

$$
\stackrel{(G.1)}{\leq} \sum_{i=1}^{\tilde{i}+1} \sum_{k=t_{i-1}}^{t_i-1} \exp\left(-2 \sum_{j=i+1}^{\tilde{i}+1} 2^{\tilde{i}-j+1} \ln \frac{\alpha_j}{\alpha_{j-1}} - 2 \cdot 2^{\tilde{i}-i+1} \ln \frac{\alpha_i}{\alpha_{i-1} + 2^{i-1}\mu(k - t_{i-1})}\right) \eta_k^2
$$

$$
= \sum_{i=1}^{\tilde{i}+1} \sum_{k=t_{i-1}}^{t_i-1} \exp\left(\sum_{j=i+1}^{\tilde{i}+1} 2^{\tilde{i}-j+2} \ln \frac{\alpha_{j-1}}{\alpha_j} + 2^{\tilde{i}-i+2} \ln \frac{\alpha_{i-1} + 2^{i-1}\mu(k - t_{i-1})}{\alpha_i}\right) \eta_k^2
$$

$$
= \sum_{i=1}^{\tilde{i}+1} \sum_{k=t_{i-1}}^{t_i-1} \left[\prod_{j=i+1}^{\tilde{i}+1} \left(\frac{\alpha_{j-1}}{\alpha_j}\right)^{2^{\tilde{i}-j+2}}\right] \cdot \left(\frac{\alpha_{i-1} + 2^{i-1}\mu(k - t_{i-1})}{\alpha_i}\right)^{2^{\tilde{i}-i+2}} \eta_k^2
$$

$$
= \sum_{i=1}^{\tilde{i}+1} \left[\prod_{j=i+1}^{\tilde{i}+1} \left(\frac{\alpha_{j-1}}{\alpha_j}\right)^{2^{\tilde{i}-j+2}}\right] \sum_{k=t_{i-1}}^{t_i-1} \left(\frac{\alpha_{i-1} + 2^{i-1}\mu(k - t_{i-1})}{\alpha_i}\right)^{2^{\tilde{i}-i+2}} \eta_k^2
$$

$$
\stackrel{(3.3)}{=} \sum_{i=1}^{\tilde{i}+1} \left[\prod_{j=i+1}^{\tilde{i}+1} \left(\frac{\alpha_{j-1}}{\alpha_j}\right)^{2^{\tilde{i}-j+2}}\right]
$$

$$
\cdot \sum_{k=t_{i-1}}^{t_i-1} \left(\frac{\alpha_{i-1} + 2^{i-1}\mu(k - t_{i-1})}{\alpha_i}\right)^{2^{\tilde{i}-i+2}} \left(L + \mu \sum_{j=1}^{i-1} \Delta_j 2^{j-1} + 2^{i-1}\mu(k - t_{i-1})\right)^{-2}
$$

$$
\stackrel{(G.2)}{=} \sum_{i=1}^{\tilde{i}+1} \left[\prod_{j=i+1}^{\tilde{i}+1} \left(\frac{\alpha_{j-1}}{\alpha_j}\right)^{2^{\tilde{i}-j+2}}\right]
$$

$$
\cdot \sum_{k=t_{i-1}}^{t_i-1} \left(\frac{\alpha_{i-1} + 2^{i-1}\mu(k - t_{i-1})}{\alpha_i}\right)^{2^{\tilde{i}-i+2}} \left(\alpha_{i-1} + 2^{i-1}\mu(k - t_{i-1})\right)^{-2}
$$

$$
= \sum_{i=1}^{\tilde{i}+1} \left[ \prod_{j=i+1}^{\tilde{i}+1} \left( \frac{\alpha_{j-1}}{\alpha_j} \right)^{2^{\tilde{i}-j+2}} \right] \alpha_i^{-2^{\tilde{i}-i+2}}
$$

$$
\cdot \sum_{k=t_{i-1}}^{t_i-1} (\alpha_{i-1} + 2^{i-1}\mu(k-t_{i-1}))^{2^{\tilde{i}-i+2}} \left( \alpha_{i-1} + 2^{i-1}\mu(k-t_{i-1}) \right)^{-2}
$$

$$
= \sum_{i=1}^{\tilde{i}+1} \left[ \prod_{j=i+1}^{\tilde{i}+1} \left( \frac{\alpha_{j-1}}{\alpha_j} \right)^{2^{\tilde{i}-j+2}} \right] \alpha_i^{-2^{\tilde{i}-i+2}} \sum_{k=t_{i-1}}^{t_i-1} (\alpha_{i-1} + 2^{i-1}\mu(k-t_{i-1}))^{2^{\tilde{i}-i+2}-2}
$$

$$
\overset{\text{(G.3)}}{\leq} 2 \cdot \frac{1}{2^{\tilde{i}+1}\mu} \cdot \sum_{i=1}^{\tilde{i}+1} \left[ \prod_{j=i+1}^{\tilde{i}+1} \left( \frac{\alpha_{j-1}}{\alpha_j} \right)^{2^{\tilde{i}-j+2}} \right] \left( \alpha_i^{2^{\tilde{i}-i+2}-1} - \alpha_{i-1}^{2^{\tilde{i}-i+2}-1} \right) \alpha_i^{-2^{\tilde{i}-i+2}}
$$

$$
\overset{\text{(G.4)}}{\leq} 2 \cdot \frac{1}{2^{\tilde{i}+1}\mu} \cdot \alpha_{i+1}^{-1} \leq 2 \cdot \frac{\eta_{t_{\tilde{i}+1}}}{\lambda_j},
$$

where the last inequality is because of the condition $\lambda_j \in [\mu \cdot 2^{\tilde{i}}, \mu \cdot 2^{\tilde{i}+1})$ and the definition of $\alpha_i$.

Therefore, we have

$$
v_{t_{\tilde{i}+1},j} \leq 2\exp(2) \cdot \frac{\eta_{t_{\tilde{i}+1}}}{\lambda_j} \leq 15 \cdot \frac{\eta_{t_{\tilde{i}+1}}}{\lambda_j}.
$$

$\square$

**Lemma 12.** *Let objective function $f(x)$ be quadratic and Assumption (1.7) hold. Running SGD for $T$-steps starting from $w_0$ and a learning rate sequence $\{\eta_t\}_{t=1}^T$ defined in Eqn. (3.3), the final iterate $w_{T+1}$ satisfies*

$$
\mathbb{E}\left[ (w_{T+1} - w_*)^\top H (w_{T+1} - w_*) \right] \leq (w_0 - w_*)^\top H (w_0 - w_*) \cdot \exp\left( -2\mu \sum_{k=0}^T \eta_k \right)
$$

$$
+ 15\sigma^2 \mu \sum_{\tilde{i}=0}^{I_{\max}-1} \frac{2^{\tilde{i}+1} s_{\tilde{i}}}{L + \mu \sum_{j=1}^{\tilde{i}+1} \Delta_j 2^{j-1}}.
$$

*Proof.* The target of this lemma is to obtain the explicit form to bound the variance term. By the definition of $v_{t+1,j}$ in Lemma 10, we can obtain that

$$
\sum_{\tau=0}^T \mathbb{E}\left[ \eta_\tau^2 n_\tau^\top \cdot P_T \dots P_{\tau+1} H P_{\tau+1} \dots P_T \cdot n_\tau \right]
$$

$$
\overset{\text{(F.7)}}{\leq} \sigma^2 \sum_{j=1}^d \lambda_j^2 \cdot v_{T+1,j}
$$

$$
\overset{\text{(G.9)}}{\leq} \sigma^2 \sum_{j=1}^d \lambda_j^2 \cdot \max\left( v_{t_{\tilde{i}+1}+1,j}, \frac{\eta_{t_{\tilde{i}+1}+1}}{\lambda_j} \right)
$$

$$
\overset{\text{(G.11)}}{\leq} \sigma^2 \sum_{j=1}^d \lambda_j^2 \cdot \max\left( 15 \cdot \frac{\eta_{t_{\tilde{i}+1}+1}}{\lambda_j}, \frac{\eta_{t_{\tilde{i}+1}+1}}{\lambda_j} \right)
$$

$$
= 15\sigma^2 \sum_{j=1}^d \lambda_j \cdot \eta_{t_{\tilde{i}+1}+1} \leq 15\sigma^2 \sum_{j=1}^d \lambda_j \cdot \eta_{t_{\tilde{i}+1}} \leq 15\sigma^2 \mu \sum_{\tilde{i}=0}^{I_{\max}-1} 2^{\tilde{i}+1} s_{\tilde{i}} \cdot \eta_{t_{\tilde{i}+1}},
$$

where the last inequality is because $\lambda_j \in [2^{\tilde{i}}\mu, \; 2^{\tilde{i}+1}\mu)$ and there are $s_{\tilde{i}}$ such $\lambda_j$'s lie in this range. By Eqn. (3.3), we have

$$
\eta_{t_{\tilde{i}+1}} = \frac{1}{L + \mu \sum_{j=1}^{\tilde{i}+1} \Delta_j 2^{j-1}}. \tag{G.13}
$$

Therefore, we have

$$\sum_{\tau=0}^{T} \mathbb{E}\left[\eta_\tau^2 n_\tau^\top \cdot P_T \dots P_{\tau+1} H P_{\tau+1} \dots P_T \cdot n_\tau\right] \leq 15\sigma^2 \mu \sum_{\tilde{i}=0}^{I_{\max}-1} \frac{2^{\tilde{i}+1} s_{\tilde{i}}}{L + \mu \sum_{j=1}^{\tilde{i}+1} \Delta_j 2^{j-1}}. \quad \text{(G.14)}$$

Combining with Lemma 4 and Lemma 6, we can obtain that

$$\mathbb{E}\left[(w_{T+1} - w_*)^\top H(w_{T+1} - w_*)\right] \leq (w_0 - w_*)^\top H(w_0 - w_*) \cdot \exp\left(-2\mu \sum_{k=0}^{T} \eta_k\right)$$

$$+ 15\sigma^2 \mu \sum_{\tilde{i}=0}^{I_{\max}-1} \frac{2^{\tilde{i}+1} s_{\tilde{i}}}{L + \mu \sum_{j=1}^{\tilde{i}+1} \Delta_j 2^{j-1}}.$$

$\square$

**Lemma 13.** *For $\forall t \geq 0$, the learning rate sequence $\{\eta_t\}_{t=1}^{T}$ defined in Eqn. (3.3) satisfies*

$$\eta_t \leq \frac{1}{L + \mu t} \quad \text{(G.15)}$$

*Proof.* For $\forall t \geq 0$, there $\exists i \geq 1$, where $t \in [t_{i-1}, t_i)$. Given the form defined in Eqn. (3.3), we have,

$$\eta_t = \frac{1}{L + \mu \sum_{j=1}^{i-1} \Delta_j 2^{j-1} + 2^{i-1} \mu(t - t_{i-1})}$$

$$\leq \frac{1}{L + \mu \sum_{j=1}^{i-1} \Delta_j + \mu(t - t_{i-1})}$$

$$\stackrel{(3.4)}{=} \frac{1}{L + \mu \sum_{j=1}^{i-1}(t_j - t_{j-1}) + \mu(t - t_{i-1})}$$

$$= \frac{1}{L + \mu(t_{i-1} - t_0) + \mu(t - t_{i-1})}$$

$$= \frac{1}{L + \mu(t - t_0)}$$

$$= \frac{1}{L + \mu t}$$

$\square$

**Lemma 14.** *Let objective function $f(x)$ be quadratic and Assumption (1.7) hold. Running SGD for $T$-steps starting from $w_0$ and a learning rate sequence $\{\eta_t\}_{t=1}^{T}$ defined in Eqn. (3.3), the final iterate $w_{T+1}$ satisfies*

$$\mathbb{E}\left[(w_{T+1} - w_*)^\top H(w_{T+1} - w_*)\right] \leq (w_0 - w_*)^\top H(w_0 - w_*) \cdot \frac{\kappa^2}{\Delta_1^2}$$

$$+ 15\sigma^2 \mu \sum_{\tilde{i}=0}^{I_{\max}-1} \frac{2^{\tilde{i}+1} s_{\tilde{i}}}{L + \mu \sum_{j=1}^{\tilde{i}+1} \Delta_j 2^{j-1}}.$$

*Proof.* The target of this lemma is to obtain the explicit form to bound the bias term.

First, by Eqn. (G.1) and the condition $\lambda_j \in [\mu \cdot 2^{\tilde{i}}, \mu \cdot 2^{\tilde{i}+1})$, we have

$$
\exp\left(-2\lambda_j \sum_{k=0}^{T} \eta_k\right) \le \exp\left(-2 \sum_{k=0}^{t_{\tilde{i}+1}-1} \eta_k \lambda_j\right) \le \exp\left(-2 \sum_{i=1}^{\tilde{i}+1} \frac{1}{2^{i-1}\mu} \ln \frac{\alpha_i}{\alpha_{i-1}} \lambda_j\right)
$$

$$
\le \exp\left(-\sum_{i=1}^{\tilde{i}+1} 2^{\tilde{i}-i+2} \ln \frac{\alpha_i}{\alpha_{i-1}}\right) = \prod_{i=1}^{\tilde{i}+1} \left(\frac{\alpha_{i-1}}{\alpha_i}\right)^{2^{\tilde{i}-i+2}} \tag{G.16}
$$

$$
\le \prod_{i=1}^{\tilde{i}+1} \left(\frac{\alpha_{i-1}}{\alpha_i}\right)^2 = \left(\frac{\alpha_1}{\alpha_{\tilde{i}+1}}\right)^2 = L^2 \cdot \eta_{t_{\tilde{i}+1}}^2
$$

For $\lambda_j = \mu$, since $\mu \in [\mu \cdot 2^{\tilde{i}}, \mu \cdot 2^{\tilde{i}+1})$ for $\tilde{i} = 0$, it follows,

$$
\exp\left(-2\mu \sum_{k=0}^{T} \eta_k\right) \le L^2 \cdot \eta_{t_1}^2 \overset{(G.15)}{\le} \left(\frac{L}{L+\mu t_1}\right)^2 \overset{(3.4)}{=} \left(\frac{L}{L+\mu\Delta_1}\right)^2 \le \left(\frac{L}{\mu\Delta_1}\right)^2 = \frac{\kappa^2}{\Delta_1^2} \tag{G.17}
$$

Combining with Lemma 12, we obtain that,

$$
\mathbb{E}\left[(w_{T+1} - w_*)^\top H (w_{T+1} - w_*)\right] \le (w_0 - w_*)^\top H (w_0 - w_*) \cdot \frac{\kappa^2}{\Delta_1^2}
$$

$$
+ 15\sigma^2 \mu \sum_{\tilde{i}=0}^{I_{\max}-1} \frac{2^{\tilde{i}+1} s_{\tilde{i}}}{L + \mu \sum_{j=1}^{\tilde{i}+1} \Delta_j 2^{j-1}}.
$$

$\square$

## G.1 PROOF OF LEMMA 1

**Lemma 1.** *Let objective function $f(x)$ be quadratic and Assumption (1.7) hold. Running SGD for $T$-steps starting from $w_0$ and a learning rate sequence $\{\eta_t\}_{t=1}^{T}$ defined in Eqn. (3.3), the final iterate $w_{T+1}$ satisfies*

$$
\mathbb{E}\left[f(w_{T+1}) - f(w_*)\right] \le (f(w_0) - f(w_*)) \cdot \frac{\kappa^2}{\Delta_1^2} + \frac{15}{2} \cdot \sigma^2 \mu \sum_{\tilde{i}=0}^{I_{\max}-1} \frac{2^{\tilde{i}+1} s_{\tilde{i}}}{L + \mu \sum_{j=1}^{\tilde{i}+1} \Delta_j 2^{j-1}}. \tag{3.6}
$$

*Proof.* For $\forall t \ge 0$, we have

$$
f(w_t) - f(w_*) \overset{(1.2)}{=} \mathbb{E}\left[\frac{1}{2} w_t^\top H(\xi) w_t - b(\xi)^\top w_t\right] - \mathbb{E}\left[\frac{1}{2} w_*^\top H(\xi) w_* - b(\xi)^\top w_*\right]
$$

$$
= \left(\frac{1}{2} w_t^\top \mathbb{E}[H(\xi)] w_t - \mathbb{E}[b(\xi)]^\top w_t\right) - \left(\frac{1}{2} w_*^\top \mathbb{E}[H(\xi)] w_* - \mathbb{E}[b(\xi)]^\top w_*\right)
$$

$$
= \left(\frac{1}{2} w_t^\top H w_t - b^\top w_t\right) - \left(\frac{1}{2} w_*^\top H w_* - b^\top w_*\right)
$$

$$
\overset{(1.4)}{=} \left(\frac{1}{2} w_t^\top H w_t - b^\top w_t\right) - \left(\frac{1}{2} b^\top \left(H^\top\right)^{-1} b - b^\top H^{-1} b\right)
$$

$$
= \left(\frac{1}{2} w_t^\top H w_t - b^\top w_t\right) - \left(\frac{1}{2} b^\top H^{-1} b - b^\top H^{-1} b\right)
$$

$$
= \frac{1}{2} w_t^\top H w_t - b^\top w_t + \frac{1}{2} b^\top H^{-1} b
$$

$$
= \frac{1}{2} w_t^\top H w_t - \frac{1}{2} b^\top w_t - \frac{1}{2} b^\top w_t + \frac{1}{2} b^\top H^{-1} b
$$

$$
\begin{aligned}
=&\frac{1}{2}w_t^\top H w_t - \frac{1}{2}w_t^\top b - \frac{1}{2}b^\top w_t + \frac{1}{2}b^\top H^{-1}b \\
=&\frac{1}{2}w_t^\top H w_t - \frac{1}{2}w_t^\top H w_* - \frac{1}{2}w_*^\top H w_t + \frac{1}{2}w_*^\top H w_* \\
=&\frac{1}{2}(w_t - w_*)^\top H(w_t - w_*),
\end{aligned}
$$

where the 5th equality is entailed by the fact that $H^\top = H$ is a symmetric matrix, and the 9th equality uses both $H^\top = H$ and Eqn 1.4.

Combine the above result with Lemma 14, we obtain that

$$
\mathbb{E}\left[f(w_{T+1}) - f(w_*)\right] \leq (f(w_0) - f(w_*)) \cdot \frac{\kappa^2}{\Delta_1^2} + \frac{15}{2} \cdot \sigma^2 \mu \sum_{\tilde{i}=0}^{I_{\max}-1} \frac{2^{\tilde{i}+1}s_{\tilde{i}}}{L + \mu \sum_{j=1}^{\tilde{i}+1}\Delta_j 2^{j-1}}.
$$

$\square$

## G.2 PROOF OF THEOREM 1

**Theorem 1.** *Let objective function $f(x)$ be quadratic and Assumption (1.7) hold. Running SGD for $T$-steps starting from $w_0$, a learning rate sequence $\{\eta_t\}_{t=1}^T$ defined in Eqn. (3.3) and $\Delta_i$ defined in Eqn. (3.5), the final iterate $w_{T+1}$ satisfies*

$$
\mathbb{E}\left[f(w_{T+1}) - f(w_*)\right] \leq (f(w_0) - f(w_*)) \cdot \frac{\kappa^2 \cdot \left(\sum_{i=0}^{I_{\max}-1}\sqrt{s_i}\right)^2}{s_0 T^2} + \frac{15\left(\sum_{i=0}^{I_{\max}-1}\sqrt{s_i}\right)^2}{T} \cdot \sigma^2.
$$

*Proof.* We have

$$
\begin{aligned}
\mu \cdot \sum_{\tilde{i}=0}^{I_{\max}-1} \frac{2^{\tilde{i}+1}s_{\tilde{i}}}{L + \mu \sum_{j=1}^{\tilde{i}+1}\Delta_j 2^{j-1}} &< \mu \cdot \sum_{\tilde{i}=0}^{I_{\max}-1} \frac{2^{\tilde{i}+1}s_{\tilde{i}}}{\mu 2^{\tilde{i}}\Delta_{\tilde{i}+1}} \overset{(3.5)}{=} 2\sum_{\tilde{i}=0}^{I_{\max}-1} \frac{s_{\tilde{i}}}{\frac{\sqrt{s_{\tilde{i}}}}{\sum_{i=0}^{I_{\max}-1}\sqrt{s_i}} \cdot T} \\
&= \frac{2}{T} \cdot \sum_{i=0}^{I_{\max}-1}\sqrt{s_i} \cdot \sum_{\tilde{i}=0}^{I_{\max}-1}\sqrt{s_{\tilde{i}}} = \frac{2\left(\sum_{i=0}^{I_{\max}-1}\sqrt{s_i}\right)^2}{T}.
\end{aligned}
$$

Combining with Lemma 1 and the definition of $\Delta_1$, we can obtain that

$$
\mathbb{E}\left[f(w_{T+1}) - f(w_*)\right] \leq (f(w_0) - f(w_*)) \cdot \frac{\kappa^2 \cdot \left(\sum_{i=0}^{I_{\max}-1}\sqrt{s_i}\right)^2}{s_0 T^2} + \frac{15\left(\sum_{i=0}^{I_{\max}-1}\sqrt{s_i}\right)^2}{T} \cdot \sigma^2.
$$

$\square$

## G.3 PROOF OF COROLLARY 2

**Corollary 2.** *Given the same setting as in Theorem 1, when Hessian $H$'s eigenvalue distribution $p(\lambda)$ satisfies "power law", i.e.*

$$
p(\lambda) = \frac{1}{Z} \cdot \exp(-\alpha(\ln(\lambda) - \ln(\mu))) = \frac{1}{Z} \cdot \left(\frac{\mu}{\lambda}\right)^\alpha \tag{3.7}
$$

*for some $\alpha > 1$, where $Z = \int_\mu^L (\mu/\lambda)^\alpha d\lambda$, there exists a constant $C(\alpha)$ which only depends on $\alpha$, such that the final iterate $w_{T+1}$ satisfies*

$$
\mathbb{E}\left[f(w_{T+1}) - f(w_*)\right] \leq \left((f(w_0) - f(w_*)) \cdot \frac{\kappa^2}{T^2} + \frac{d\sigma^2}{T}\right) \cdot C(\alpha).
$$

*Proof.* According to Theorem 1,

$$\mathbb{E}\left[f(w_{T+1}) - f(w_*)\right]$$

$$\leq (f(w_0) - f(w_*)) \cdot \frac{\kappa^2 \cdot \left(\sum_{i=0}^{I_{\max}-1} \sqrt{s_i}\right)^2}{s_0 T^2} + \frac{15 \left(\sum_{i=0}^{I_{\max}-1} \sqrt{s_i}\right)^2}{T} \cdot \sigma^2$$

$$= (f(w_0) - f(w_*)) \cdot \frac{\kappa^2}{T^2} \cdot \frac{\left(\sum_{i=0}^{I_{\max}-1} \sqrt{s_i}\right)^2}{s_0} + \frac{d\sigma^2}{T} \cdot \frac{15 \left(\sum_{i=0}^{I_{\max}-1} \sqrt{s_i}\right)^2}{d}.$$

The key terms here are $C_1 \triangleq \left(\sum_{i=0}^{I_{\max}-1} \sqrt{s_i}\right)^2 / s_0$ and $C_2 \triangleq 15 \left(\sum_{i=0}^{I_{\max}-1} \sqrt{s_i}\right)^2 / d$. As long as we can bound both terms with a constant $C(\alpha)$, the corollary will be directly proved.

1) If $\kappa < 2$, then there is only one interval with $s_0 = d$. By setting $C(\alpha) = \max(C_1, C_2) = 15$, this completes the proof.

2) If $\kappa \geq 2$, then bounding $C_1$ and $C_2$ be done by computing the value of $s_i$ under power law. For all interval $i$ except the last interval, we have,

$$\frac{s_i}{d} \stackrel{(3.1)}{=} \#\lambda_j \in [\mu \cdot 2^i, \mu \cdot 2^{i+1}) = \int_{\mu \cdot 2^i}^{\mu \cdot 2^{i+1}} p(\lambda) d\lambda$$

$$\stackrel{(3.7)}{=} \int_{\mu \cdot 2^i}^{\mu \cdot 2^{i+1}} \frac{1}{Z} \cdot \left(\frac{\mu}{\lambda}\right)^\alpha d\lambda = \frac{1}{Z} \cdot \mu^\alpha \cdot \int_{\mu \cdot 2^i}^{\mu \cdot 2^{i+1}} \lambda^{-\alpha} d\lambda$$

$$= \left(\int_\mu^L \left(\frac{\mu}{\lambda}\right)^\alpha d\lambda\right)^{-1} \cdot \mu^\alpha \cdot \int_{\mu \cdot 2^i}^{\mu \cdot 2^{i+1}} \lambda^{-\alpha} d\lambda = \left(\int_\mu^L \lambda^{-\alpha} d\lambda\right)^{-1} \cdot \int_{\mu \cdot 2^i}^{\mu \cdot 2^{i+1}} \lambda^{-\alpha} d\lambda$$

$$= \left(\frac{\lambda^{1-\alpha}}{1-\alpha}\bigg|_\mu^L\right)^{-1} \cdot \left(\frac{\lambda^{1-\alpha}}{1-\alpha}\bigg|_{\mu \cdot 2^i}^{\mu \cdot 2^{i+1}}\right) = \left(\lambda^{1-\alpha}\bigg|_\mu^L\right)^{-1} \cdot \left(\lambda^{1-\alpha}\bigg|_{\mu \cdot 2^i}^{\mu \cdot 2^{i+1}}\right)$$

$$= \left(L^{1-\alpha} - \mu^{1-\alpha}\right)^{-1} \cdot \left(\mu^{1-\alpha} \cdot \left(2^{i+1}\right)^{1-\alpha} - \mu^{1-\alpha} \cdot \left(2^i\right)^{1-\alpha}\right)$$

$$= \frac{\mu^{1-\alpha} \cdot \left(2^{i+1}\right)^{1-\alpha} - \mu^{1-\alpha} \cdot \left(2^i\right)^{1-\alpha}}{L^{1-\alpha} - \mu^{1-\alpha}} = \frac{\left(2^{i+1}\right)^{1-\alpha} - \left(2^i\right)^{1-\alpha}}{\kappa^{1-\alpha} - 1}$$

$$= 2^{i(1-\alpha)} \cdot \frac{2^{1-\alpha} - 1}{\kappa^{1-\alpha} - 1}$$

Therefore, we have

$$s_i = d \cdot 2^{i(1-\alpha)} \cdot \frac{2^{1-\alpha} - 1}{\kappa^{1-\alpha} - 1} = d \cdot \frac{2^{1-\alpha} - 1}{\kappa^{1-\alpha} - 1} \cdot 2^{i(1-\alpha)} \tag{G.18}$$

holds for all interval $i$ except the last interval $i' = I_{\max} - 1 = \log_2 \kappa - 1 > 0$. This last interval may not completely covers $[\mu \cdot 2^{i'}, \mu \cdot 2^{i'+1})$ due to the boundary truncated by $L$, but we still have

$$s_{i'} \leq d \cdot \frac{2^{1-\alpha} - 1}{\kappa^{1-\alpha} - 1} \cdot 2^{i'(1-\alpha)}$$

It follows,

$$
\begin{aligned}
\left(\sum_{i=0}^{I_{\max}-1}\sqrt{s_i}\right)^2 &\leq d\cdot\frac{2^{1-\alpha}-1}{\kappa^{1-\alpha}-1}\cdot\left(\sum_{i=0}^{I_{\max}-1}\sqrt{2^{i(1-\alpha)}}\right)^2 = d\cdot\frac{2^{1-\alpha}-1}{\kappa^{1-\alpha}-1}\cdot\left(\sum_{i=0}^{I_{\max}-1}2^{i(1-\alpha)/2}\right)^2 \\
&= d\cdot\frac{2^{1-\alpha}-1}{\kappa^{1-\alpha}-1}\cdot\left(\sum_{i=0}^{I_{\max}-1}\left(\frac{1}{2}\right)^{i(\alpha-1)/2}\right)^2 \\
&\leq d\cdot\frac{2^{1-\alpha}-1}{\kappa^{1-\alpha}-1}\cdot\left(\sum_{i=0}^{\infty}\left(\frac{1}{2}\right)^{i(\alpha-1)/2}\right)^2 = d\cdot\frac{2^{1-\alpha}-1}{\kappa^{1-\alpha}-1}\cdot\left(\sum_{i=0}^{\infty}\left(\frac{1}{2^{(\alpha-1)/2}}\right)^{i}\right)^2 \\
&= d\cdot\frac{2^{1-\alpha}-1}{\kappa^{1-\alpha}-1}\cdot\left(\frac{1}{1-\frac{1}{2^{(\alpha-1)/2}}}\right)^2 = d\cdot\frac{2^{1-\alpha}-1}{\kappa^{1-\alpha}-1}\cdot\left(\frac{1}{1-2^{(1-\alpha)/2}}\right)^2.
\end{aligned}
$$

Thus,

$$
\begin{aligned}
C_1 &=\frac{\left(\sum_{i=0}^{I_{\max}-1}\sqrt{s_i}\right)^2}{s_0} \leq \frac{d\cdot\frac{2^{1-\alpha}-1}{\kappa^{1-\alpha}-1}\cdot\left(\frac{1}{1-2^{(1-\alpha)/2}}\right)^2}{d\cdot\frac{2^{1-\alpha}-1}{\kappa^{1-\alpha}-1}\cdot 2^{0(1-\alpha)}} = \left(\frac{1}{1-2^{(1-\alpha)/2}}\right)^2 \\
C_2 &=\frac{15\left(\sum_{i=0}^{I_{\max}-1}\sqrt{s_i}\right)^2}{d} \leq \frac{15}{d}\cdot d\cdot\frac{2^{1-\alpha}-1}{\kappa^{1-\alpha}-1}\cdot\left(\frac{1}{1-2^{(1-\alpha)/2}}\right)^2 \\
&=15\cdot\frac{1-\left(\frac{1}{2}\right)^{\alpha-1}}{1-\left(\frac{1}{\kappa}\right)^{\alpha-1}}\cdot\left(\frac{1}{1-2^{(1-\alpha)/2}}\right)^2 \\
&\leq 15\cdot\left(\frac{1}{1-2^{(1-\alpha)/2}}\right)^2.
\end{aligned}
$$

Here the last inequality for $C_2$ is entailed by $\kappa\geq 2$ and $\alpha>1$.

By setting $C(\alpha)=\max(C_1,C_2)=15\cdot\left(\frac{1}{1-2^{(1-\alpha)/2}}\right)^2$, we obtain

$$
\begin{aligned}
&\mathbb{E}\left[f(w_{T+1})-f(w_*)\right] \\
&\leq (f(w_0)-f(w_*))\cdot\frac{\kappa^2}{T^2}\cdot\frac{\left(\sum_{i=0}^{I_{\max}-1}\sqrt{s_i}\right)^2}{s_0} + \frac{d\sigma^2}{T}\cdot\frac{15\left(\sum_{i=0}^{I_{\max}-1}\sqrt{s_i}\right)^2}{d}. \\
&=(f(w_0)-f(w_*))\cdot\frac{\kappa^2}{T^2}\cdot C_1 + \frac{d\sigma^2}{T}\cdot C_2 \\
&\leq \left((f(w_0)-f(w_*))\cdot\frac{\kappa^2}{T^2} + \frac{d\sigma^2}{T}\right)\cdot C(\alpha).
\end{aligned}
$$

$\square$

### G.4 PROOF OF THEOREM 4

**Theorem 4.** *Let objective function $f(x)$ be quadratic. We run SGD for $T$-steps starting from $w_0$ and a step decay learning rate sequence $\{\eta_t\}_{t=1}^{T}$ defined in Algorithm 1 of Ge et al. (2019) with $\eta_1\leq 1/L$. **As long as (1)** $H$ is diagonal, **(2)** The equality in Assumption (1.7) holds, i.e. $\mathbb{E}_\xi\left[n_t n_t^\top\right]=\sigma^2 H$ and **(3)** $\lambda_j\left(w_{0,j}-w_{*,j}\right)^2\neq 0$ for $\forall j=1,2,\ldots,d$, the final iterate $w_{T+1}$ satisfies,*

$$
\mathbb{E}\left[f(w_{T+1})-f(w_*)\right]=\Omega\left(\frac{d\sigma^2}{T}\cdot\log T\right)
$$

*Proof.* The lower bound here is an asymptotic bound. Specifically, we require

$$\frac{T}{\log T} \geq \max\left(2^{16}, 16, \frac{1}{256} \cdot \frac{\sigma^2}{\min_j \lambda_j (w_{0,j} - w_{*,j})^2}\right). \tag{G.19}$$

In Ge et al. (2019), step decay has following learning rate sequence:

$$\eta_t = \frac{\eta_1}{2^\ell} \quad \text{if } t \in \left[1 + \frac{T}{\log T} \cdot \ell, \frac{T}{\log T} \cdot (\ell + 1)\right], \tag{G.20}$$

where $\ell = 0, 1, \ldots, \log T - 1$. Notice that the index start from $t = 1$ instead of $t = 0$. For consistency with our framework, we set $\eta_0 = 0$, which produces the exact same step decay scheduler while only adding one extra iteration, thus does not affect the overall asymptotic bound.

We first translate the general notations to diagonal cases so that the idea of the proof can be clearer.

Since $f(x)$ is quadratic, according to the proof of Lemma 1 in Appendix G.1,

$$f(w_{T+1}) - f(w_*) = \frac{1}{2}(w_{T+1} - w_*)^\top H(w_{T+1} - w_*).$$

Furthermore, according to Lemma 4, where $P_t = I - \eta_t H$,

$$\mathbb{E}\left[(w_{T+1} - w_*)^\top H(w_{T+1} - w_*)\right]$$
$$= \mathbb{E}\left[(w_0 - w_*)^\top \cdot P_T \ldots P_0 H P_0 \ldots P_T \cdot (w_0 - w_*)\right]$$
$$+ \sum_{\tau=0}^{T} \mathbb{E}\left[\eta_\tau^2 n_\tau^\top \cdot P_T \ldots P_{\tau+1} H P_{\tau+1} \ldots P_T \cdot n_\tau\right]$$
$$= \sum_{j=1}^{d} \lambda_j (w_{0,j} - w_{*,j})^2 \prod_{k=0}^{T}(1 - \eta_k \lambda_j)^2 + \sum_{\tau=0}^{T} \eta_\tau^2 \sum_{j=1}^{d} \prod_{k=\tau+1}^{T} \lambda_j(1 - \eta_k \lambda_j)^2 \mathbb{E}\left[n_{\tau,j}^2\right]$$
$$= \sum_{j=1}^{d} \lambda_j (w_{0,j} - w_{*,j})^2 \prod_{k=0}^{T}(1 - \eta_k \lambda_j)^2 + \sum_{\tau=0}^{T} \eta_\tau^2 \sum_{j=1}^{d} \prod_{k=\tau+1}^{T} \lambda_j(1 - \eta_k \lambda_j)^2 \cdot \lambda_j \sigma^2$$
$$= \sum_{j=1}^{d} \lambda_j (w_{0,j} - w_{*,j})^2 \prod_{k=0}^{T}(1 - \eta_k \lambda_j)^2 + \sigma^2 \sum_{j=1}^{d} \lambda_j^2 \sum_{\tau=0}^{T} \eta_\tau^2 \prod_{k=\tau+1}^{T}(1 - \eta_k \lambda_j)^2.$$

Here the second equality is entailed by the fact that $H$ and $P_t$ are diagonal, and the third equality comes from $\mathbb{E}_\xi\left[n_t n_t^\top\right] = \sigma^2 H$. Thus, by denoting $b_j \triangleq \lambda_j (w_{0,j} - w_{*,j})^2 \prod_{k=0}^{T}(1 - \eta_k \lambda_j)^2$ and $v_j \triangleq \sum_{\tau=0}^{T} \eta_\tau^2 \prod_{k=\tau+1}^{T}(1 - \eta_k \lambda_j)^2$, we have,

$$\mathbb{E}\left[f(w_{T+1} - f(w_*)\right] = \frac{1}{2}\mathbb{E}\left[(w_{T+1} - w_*)^\top H(w_{T+1} - w_*)\right]$$
$$= \frac{1}{2}\left[\left(\sum_{j=1}^{d} b_j\right) + \left(\sigma^2 \sum_{j=1}^{d} \lambda_j^2 v_j\right)\right]. \tag{G.21}$$

To proceed the analysis, we divide all eigenvalues $\{\lambda_j\}$ into two groups:

$$\mathcal{A} = \left\{j \,\middle|\, \lambda_j > \frac{\log T}{8\eta_1 T}\right\}, \quad \mathcal{B} = \left\{j \,\middle|\, \lambda_j \leq \frac{\log T}{8\eta_1 T}\right\}, \tag{G.22}$$

where group $\mathcal{A}$ are those large eigenvalues that the variance term $v_j$ will finally dominate, and group $\mathcal{B}$ are those small eigenvalues that the bias term $b_j$ will finally dominate. Rigorously speaking,

**a) For $\forall j \in \mathcal{A}$:**

Step decay's bottleneck in variance term actually occurs at the first interval $\ell$ that satisfies

$$2^\ell \geq \lambda_j \eta_1 \cdot \frac{8T}{\log T} \tag{G.23}$$

We first show that interval $\ell$ is well-defined for any dimension $j \in \mathcal{A}$. Since $j \in \mathcal{A}$, it follows from the definition of $\mathcal{A}$ in Eqn. (G.22),

$$\lambda_j > \frac{\log T}{8\eta_1 T} \implies \lambda_j \eta_1 \cdot \frac{8T}{\log T} > 1 = 2^0$$

On the other hand, since we assume $T/\log T \geq 2^{16}$ in Eqn. (G.19), which implies $T \geq 2^{16} \Rightarrow \log T \geq 16$, it follows

$$\lambda_j \eta_1 \cdot \frac{8T}{\log T} \leq \lambda_j \eta_1 \cdot \frac{T}{2} \leq \frac{\lambda_j}{L} \cdot \frac{T}{2} \leq \frac{T}{2} = 2^{\log T - 1},$$

where the second inequality comes from $\eta_1 \leq 1/L$ in assumption (1), and the third inequality is entailed by $\lambda_j \leq L$ given the definition of $L$ in Eqn. (1.6).

As a result, we have

$$\lambda_j \eta_1 \cdot \frac{8T}{\log T} \in \left(2^0, 2^{\log T - 1}\right]$$

thus

$$2^\ell \geq \lambda_j \eta_1 \cdot \frac{8T}{\log T}$$

will guaranteed be satisified for some interval $\ell = 1, \dots, \log T - 1$. Since interval $\ell$ is the first interval satisifies Eqn. (G.23), we also have

$$2^{\ell-1} < \lambda_j \eta_1 \cdot \frac{8T}{\log T} \implies 2^\ell < \lambda_j \eta_1 \cdot \frac{16T}{\log T} \tag{G.24}$$

Back to our analysis for the lower bound, by focusing on the variance produced by interval $\ell$ only, we have,

$$v_j = \sum_{\tau=0}^{T} \eta_\tau^2 \prod_{k=\tau+1}^{T} (1 - \eta_k \lambda_j)^2 \geq \sum_{\tau=\ell \cdot \frac{T}{\log T}+1}^{(\ell+1)\cdot \frac{T}{\log T}} \eta_\tau^2 \prod_{k=\tau+1}^{T} (1 - \eta_k \lambda_j)^2$$

$$\geq \sum_{\tau=\ell \cdot \frac{T}{\log T}+1}^{(\ell+1)\cdot \frac{T}{\log T}} \eta_\tau^2 \prod_{k=\ell \cdot \frac{T}{\log T}+1}^{T} (1 - \eta_k \lambda_j)^2 = \sum_{\tau=\ell \cdot \frac{T}{\log T}+1}^{(\ell+1)\cdot \frac{T}{\log T}} \left(\frac{\eta_1}{2^\ell}\right)^2 \prod_{k=\ell \cdot \frac{T}{\log T}+1}^{T} (1 - \eta_k \lambda_j)^2$$

$$= \frac{T}{\log T} \cdot \left(\frac{\eta_1}{2^\ell}\right)^2 \prod_{k=\ell \cdot \frac{T}{\log T}+1}^{T} (1 - \eta_k \lambda_j)^2$$

$$\overset{(G.24)}{>} \frac{T}{\log T} \cdot \left(\frac{\eta_1}{\lambda_j \eta_1 \cdot \frac{16T}{\log T}}\right)^2 \prod_{k=\ell \cdot \frac{T}{\log T}+1}^{T} (1 - \eta_k \lambda_j)^2$$

$$= \frac{1}{256} \cdot \frac{\log T}{T} \cdot \frac{1}{\lambda_j^2} \cdot \prod_{k=\ell \cdot \frac{T}{\log T}+1}^{T} (1 - \eta_k \lambda_j)^2$$

$$\geq \frac{1}{256} \cdot \frac{\log T}{T} \cdot \frac{1}{\lambda_j^2} \cdot \left(1 - \sum_{k=\ell \cdot \frac{T}{\log T}+1}^{T} 2\eta_k \lambda_j\right) = \frac{1}{256} \cdot \frac{\log T}{T} \cdot \frac{1}{\lambda_j^2} \cdot \left(1 - 2\lambda_j \sum_{k=\ell \cdot \frac{T}{\log T}+1}^{T} \eta_k\right)$$

$$= \frac{1}{256} \cdot \frac{\log T}{T} \cdot \frac{1}{\lambda_j^2} \cdot \left(1 - 2\lambda_j \sum_{i=\ell}^{\log T - 1} \sum_{k=i \cdot \frac{T}{\log T}+1}^{(i+1)\cdot \frac{T}{\log T}} \eta_k\right)$$

$$= \frac{1}{256} \cdot \frac{\log T}{T} \cdot \frac{1}{\lambda_j^2} \cdot \left( 1 - 2\lambda_j \sum_{i=\ell}^{\log T - 1} \sum_{k=i\cdot\frac{T}{\log T}+1}^{(i+1)\cdot\frac{T}{\log T}} \frac{\eta_1}{2^i} \right)$$

$$= \frac{1}{256} \cdot \frac{\log T}{T} \cdot \frac{1}{\lambda_j^2} \cdot \left( 1 - 2\lambda_j \sum_{i=\ell}^{\log T - 1} \frac{T}{\log T} \cdot \frac{\eta_1}{2^i} \right)$$

$$\geq \frac{1}{256} \cdot \frac{\log T}{T} \cdot \frac{1}{\lambda_j^2} \cdot \left( 1 - 2\lambda_j \cdot \frac{T}{\log T} \cdot \frac{\eta_1}{2^{\ell-1}} \right)$$

$$\overset{\text{(G.23)}}{\geq} \frac{1}{256} \cdot \frac{\log T}{T} \cdot \frac{1}{\lambda_j^2} \cdot \left( 1 - 4\lambda_j \cdot \frac{T}{\log T} \cdot \frac{\eta_1}{\lambda_j \eta_1 \cdot \frac{8T}{\log T}} \right)$$

$$= \frac{1}{256} \cdot \frac{\log T}{T} \cdot \frac{1}{\lambda_j^2} \cdot \frac{1}{2}$$

$$= \frac{1}{512} \cdot \frac{\log T}{T} \cdot \frac{1}{\lambda_j^2}$$

Here the first inequality is obtained by focusing variance generated in interval $\ell$ only. The second inequality utilizes $\tau \geq \ell \cdot T/\log T$. The fourth inequality is entailed by $(1-a_1)(1-a_2) = 1 - a_1 - a_2 + a_1 a_2 \geq 1 - a_1 - a_2$ for $\forall a_1, a_2 \in [0,1]$, where by mathematical induction, we can extend this inequality for more terms $\prod_{i=1}^{n}(1-a_i) \geq 1 - \sum_{i=1}^{n} a_i$ as long as $\sum_{i=1}^{n} a_i \leq 1$. The fifth inequality comes from $\sum_{i=\ell}^{\log T - 1} 1/2^i \leq \sum_{i=\ell}^{\infty} 1/2^i = 1/2^{\ell-1}$.

**b) For $\forall j \in \mathcal{B}$:**

Step decay's bottleneck will occur in the bias term. Since $j \in \mathcal{B}$, it follows from the definition of $\mathcal{B}$ in Eqn.(G.22),

$$\lambda_j \leq \frac{\log T}{8\eta_1 T} \quad \Longrightarrow \quad \eta_1 \lambda_j \leq \frac{\log T}{8T},$$

we have

$$b_j = \lambda_j (w_{0,j} - w_{*,j})^2 \prod_{k=0}^{T} (1 - \eta_k \lambda_j)^2$$

$$\geq \lambda_j (w_{0,j} - w_{*,j})^2 \cdot \left( 1 - \sum_{k=0}^{T} 2\eta_k \lambda_j \right) = \lambda_j (w_{0,j} - w_{*,j})^2 \cdot \left( 1 - \sum_{k=1}^{T} 2\eta_k \lambda_j \right)$$

$$= \lambda_j (w_{0,j} - w_{*,j})^2 \cdot \left( 1 - \sum_{i=0}^{\log T - 1} \sum_{k=i\cdot\frac{T}{\log T}+1}^{(i+1)\cdot\frac{T}{\log T}} 2\eta_k \lambda_j \right)$$

$$= \lambda_j (w_{0,j} - w_{*,j})^2 \cdot \left( 1 - \sum_{i=0}^{\log T - 1} \sum_{k=i\cdot\frac{T}{\log T}+1}^{(i+1)\cdot\frac{T}{\log T}} \frac{\eta_1 \lambda_j}{2^{i-1}} \right)$$

$$= \lambda_j (w_{0,j} - w_{*,j})^2 \cdot \left( 1 - \eta_1 \lambda_j \sum_{i=0}^{\log T - 1} \frac{T}{\log T} \cdot \frac{1}{2^{i-1}} \right)$$

$$\geq \lambda_j (w_{0,j} - w_{*,j})^2 \cdot \left( 1 - 4\eta_1 \lambda_j \cdot \frac{T}{\log T} \right)$$

$$\geq \lambda_j (w_{0,j} - w_{*,j})^2 \cdot \left( 1 - 4 \cdot \frac{\log T}{8T} \cdot \frac{T}{\log T} \right)$$

$$= \lambda_j (w_{0,j} - w_{*,j})^2 \cdot \frac{1}{2},$$

where the first inequality is caused by $(1-a_1)(1-a_2) = 1 - a_1 - a_2 + a_1 a_2 \geq 1 - a_1 - a_2$ for $\forall a_1, a_2 \in [0,1]$ and applying mathematical induction for $\{a_n\}$ to obtain $\prod_{i=1}^{n}(1-a_i) \geq$

$1 - \sum_{i=1}^{n} a_i$ as long as $\sum_{i=1}^{n} a_i \leq 1$. The second equality is because $\eta_0 = 0$. The second inequality comes from $\sum_{i=0}^{\log T - 1} 1/2^{i-1} \leq \sum_{i=0}^{\infty} 1/2^{i-1} = 4$. The last inequality follows $\eta_1 \lambda_j \leq \log T/(8T)$.

From assumption (3), we know $\lambda_j \left(w_{0,j} - w_{*,j}\right)^2 > 0$. Furthermore, as we require

$$\frac{T}{\log T} \geq \frac{1}{256} \cdot \frac{\sigma^2}{\min_j \lambda_j \left(w_{0,j} - w_{*,j}\right)^2}$$

in Eqn. (G.19),

$$b_j \geq \lambda_j \left(w_{0,j} - w_{*,j}\right)^2 \cdot \frac{1}{2} \geq \min_j \lambda_j \left(w_{0,j} - w_{*,j}\right)^2 \cdot \frac{1}{2} \geq \frac{1}{512} \cdot \frac{\sigma^2}{T} \cdot \log T.$$

In sum, we have obtained

$$\forall j \in \mathcal{A}, \quad v_j \geq \frac{1}{512} \cdot \frac{\log T}{T} \cdot \frac{1}{\lambda_j^2}$$

$$\forall j \in \mathcal{B}, \quad b_j \geq \frac{1}{512} \cdot \frac{\sigma^2}{T} \cdot \log T$$

By combining with Eqn. (G.21), we have

$$\begin{aligned}
\mathbb{E}\left[f(w_{T+1} - f(w_*)\right] =& \frac{1}{2}\left[\left(\sum_{j=1}^{d} b_j\right) + \left(\sigma^2 \sum_{j=1}^{d} \lambda_j^2 v_j\right)\right] \\
\geq& \frac{1}{2}\left[\left(\sum_{j \in \mathcal{B}} b_j\right) + \left(\sigma^2 \sum_{j \in \mathcal{A}} \lambda_j^2 v_j\right)\right] \\
\geq& |\mathcal{B}| \cdot \left(\frac{1}{1024} \cdot \frac{\sigma^2}{T} \cdot \log T\right) + \sum_{j \in \mathcal{A}} \sigma^2 \cdot \lambda_j^2 \cdot \frac{1}{1024} \cdot \frac{\log T}{T} \cdot \frac{1}{\lambda_j^2} \\
=& |\mathcal{B}| \cdot \left(\frac{1}{1024} \cdot \frac{\sigma^2}{T} \cdot \log T\right) + |\mathcal{A}| \cdot \left(\frac{1}{1024} \cdot \frac{\sigma^2}{T} \cdot \log T\right) \\
=& (|\mathcal{A}| + |\mathcal{B}|) \cdot \left(\frac{1}{1024} \cdot \frac{\sigma^2}{T} \cdot \log T\right) \\
=& d \cdot \frac{1}{1024} \cdot \frac{\sigma^2}{T} \cdot \log T \\
=& \Omega\left(\frac{d\sigma^2}{T} \cdot \log T\right),
\end{aligned}$$

where the first inequality is because both the bias and variance terms are non-negative, given $b_j = \lambda_j \left(w_{0,j} - w_{*,j}\right)^2 \prod_{k=0}^{T}(1 - \eta_k \lambda_j)^2 \geq 0$ and $v_j = \sum_{\tau=0}^{T} \eta_\tau^2 \prod_{k=\tau+1}^{T}(1 - \eta_k \lambda_j)^2 \geq 0$.

$\square$

**Remark 3.** *The requirement* $T/\log T \geq 1/256 \cdot \sigma^2 / \left(\min_j \lambda_j \left(w_{0,j} - w_{*,j}\right)^2\right)$ *and assumption* $\lambda_j \left(w_{0,j} - w_{*,j}\right)^2 \neq 0$ *for* $\forall j = 1, 2, \ldots, d$ *can be replaced with* $T/\log T > 1/(8\eta_1\mu)$, *since in that case* $j \in \mathcal{A}$ *holds for* $\forall j = 1, 2, \ldots, d$ *and* $\mathcal{B} = \emptyset$. *In particular, if* $\eta_1 = 1/L$, *this requirement on* $T$ *becomes* $T/\log T \geq \kappa/8$.

### G.5 THE REASON OF USING ASSUMPTION (1.7)

In all of our analysis, we employ assumption (1.7)

$$\mathbb{E}_\xi\left[n_t n_t^\top\right] \preceq \sigma^2 H \quad \text{where } n_t = H w_t - b - (H(\xi)w_t - b(\xi))$$

which is the same as the one in Appendix C, Theorem 13 of Ge et al. (2019). This key theorem is the major difference between our work and Ge et al. (2019), which directly entails its main theorem by instantiating $\sigma$ with specific values in its assumptions.

On the other hand, it is possible to use the assumptions in Ge et al. (2019); Bach & Moulines (2013); Jain et al. (2016) instead of our assumption (1.7) for least square regression:

$$\min_w f(w) \quad \text{where } f(w) \triangleq \frac{1}{2}\mathbb{E}_{(x,y)\sim\mathcal{D}}\left[(y - w^\top x)^2\right] \tag{G.25}$$

$$y = w_*^\top x + \epsilon \text{ with } \epsilon \text{ satisfying } \mathbb{E}_{(x,y)\sim\mathcal{D}}\left[\epsilon^2 x x^\top\right] \preceq \sigma^2 H \text{ for } \forall(x,y) \sim \mathcal{D} \tag{G.26}$$

$$\mathbb{E}\left[||x||^2 x x^\top\right] \preceq R^2 H \tag{G.27}$$

By combining our Lemma 1 and assumption (1.7) with Lemma 5, Lemma 8 and Lemma 9 in Ge et al. (2019), one can obtain similar results in this paper with their assumptions. For simplicity, we just use assumption (1.7) here.

## H  RELATIONSHIP WITH (STOCHASTIC) NEWTON'S METHOD

Our motivation in Proposition 1 shares a similar idea with (stochastic) Newton's method on quadratic objectives

$$w_{t+1} = w_t - \eta_t H^{-1}\nabla f(w_t, \xi),$$

where the parameters are also updated coordinately in the "rotated space", i.e. given $H = U\Lambda U^\top$ and $w' = U^\top w$. In particular, when the Hessian $H$ is diagonal and $\eta_t = 1/(t+1)$, the update formula is exactly the same as the one for Proposition 1.

Despite of this similarity, our method differ from Newton method's and its practical variants in several aspects. First of all, our method focuses on learning rate schedulers and is a first-order method. This property is especially salient when we consider `eigencurve`'s derivatives in Section 4.3: only hyperparameter search is needed, just like other common learning rate schedulers. In addition, most second-order methods, e.g. Schraudolph (2002); Erdogdu & Montanari (2015); Grosse & Martens (2016); Byrd et al. (2016); Botev et al. (2017); Huang et al. (2020); Yang et al. (2021), approximates the Hessian matrix or the Hessian inverse and exploits the curvature information, while `eigencurve` only utilizes the rough estimation of the Hessian spectrum. On top of that, this estimation is only an one-time effect and can be even further removed for similar models. These key differences highlight `eigencurve`'s advantages over most second-order methods in practice.

