# OpenReview forum: "Eigencurve: Optimal Learning Rate Schedule for SGD on Quadratic Objectives with Skewed Hessian Spectrums"
_ICLR.cc/2022/Conference — ICLR 2022 Poster_

### Official Review · Reviewer_obgH · 2021-10-29

**Correctness:** 3
**Technical Novelty And Significance:** 3
**Empirical Novelty And Significance:** 3
**Recommendation:** 8
**Confidence:** 4

**Main Review:**

Strengths
- Eigencurve is proved to achieve the optimal convergence rate when the eigenvalue distribution is skewed. It does no worse than step decay generally.
- In the appendix, eigencurve is verified on ridge regression setting where the exact Hessian is known. This helps validate the correctness of the theory.
- The empirical results are strong on CIFAR-10 and CIFAR-100. The authors verify that the loss surface is well approximated by quadratics. They show lower training loss and improved validation performance.
- Paper is clearly written and easy to follow.


Weaknesses
- There is no theory analyzing the non-asymptotic case.
- One concern I have with this method is the computational cost. This is somewhat mitigated by experiments in the appendix demonstrating that the distribution of eigenvalues of the Hessian can be re-used across different models on CIFAR-10.  Other learning rate scheduler have little to no additional overhead. Perhaps it would be useful to report the wall clock time of compute the eigenvalues + training.
- The results on ImageNet appear to be less strong than those on CIFAR and they are not emphasized in the main text. I am a bit concerned that the learning rate scheduler offers limited improvement in the standard training regime (200 epochs on CIFAR and ~100 on ImageNet).
- The experiments are on image classification datasets. Extending these results to other domains, such as language modelling, would improve the significance of the work.

Minor
- In the appendix, it's unclear what the extra term is. Adding a sentence to two would make things more clear.

----
### Post-rebuttal
I'm increasing my score to an 8. Small note is that I believe step decay applied on ImageNet typically decays at epoch 30, 60, and 80 (e.g. Goyal, Priya, et al. "Accurate, large minibatch sgd: Training imagenet in 1 hour."), so it'd be best to compare to that as a baseline.

**Summary Of The Paper:**

The authors propose Eigencurve, a new approach to learning rate scheduling that utilizes information form the eigenvalues of the Hessian. They show that this scheduler obtains the minimax optimal rate on the noisy quadratic problem. Empirically, this scheduler demonstrates faster convergence on CIFAR-10 and ImageNet, especially when the number of epochs is small.

**Summary Of The Review:**

Overall this paper offers an interesting perspective and approach to learning rate scheduling on an existing quadratic setting that shows practical implications in image classification. The experiments are thorough (including various ablations in the appendix) and validate the improved performance from the approach, especially when the number of epochs is limited.

---

> ### Author Response · Authors · 2021-11-17
> **Response to reviewer obgH**
>
> First, we would like to offer our thanks for this reviewer's constructive comments and recognition on our contributions.
>
> **Overall:** To answer the reviewer's questions and support our claims, we have added additional step decay baselines on ImageNet for comparison and an extra experiment in LSTM + Penn Treebank + Language Modeling for Elastic Step Decay. More additional results are available in the revised version of our paper. One may also refer to our responses to other reviewers.
>
> Here we will address all the raised questions one by one.
>
> **Q1: There is no theory analyzing the non-asymptotic case.**
>
> **A1:** The results are already non-asymptotic for quadratic objectives. As for NN, we focus on asymptotic convergence in this paper, since more results may make the paper too long to be easily understood, so we leave it to future researches.
>
> **Q2: One concern I have with this method is the computational cost...**
>
> **A2:** As presented in Appendix B.3, we admit the time cost for estimation Hessian matrix is much more expensive when compared with normal training processes due to limited large-scale Hessian estimation tools available so far.
>
> However, as the reviewer has recognized, the heavy estimation process is normally a onetime effort, and can be reused for similar tasks and models, at least in the experiments we demonstrated. In practice, we recommend the practitioner to directly use Elastic Step Decay or Cosine-power, while treating Eigencurve as a general framework for inspiring and discovering new simple-form schedulers for arising new models or tasks.
>
> **Q3: The results on ImageNet appear to be less strong than those on CIFAR and they are not emphasized in the main text...**
>
> **A3:** Our emphasis to is show that Eigencurve can be used to find schedulers that surpass step decay both theoretically and empirically. In that spirit, We have added ImageNet results of the common step decay baseline, which decays the learning rate 10 folds at epoch 30 and epoch 60. The experiments are conducted in ResNet50 + ImageNet, as shown in the following table, with the same settings in Appendix A.4.
>
> | Schedule |  Top-1 Test acc (%) | Top-5 Validation acc (%) |
> | -- | -- | -- |
> | Step Decay         |   75.63    |   92.66    |
> | Cosine Decay       |   76.67    |   93.28    |
> | Elastic Step Decay | **76.93**  | **93.32**  |
>
> The result appears less strong in ImageNet at first glance because the cosine decay scheduler is already a very strong baseline in practice. When we take a look at the result of step decay, the improvement will be more obvious.
>
> **Q4: The experiments are on image classification datasets. Extending these results to other domains, such as language modelling, would improve the significance of the work.**
>
> **A4:** Yes. We have conducted an extra experiment in language modeling task with LSTM + Penn Treebank for Elastic Step Decay. Results support our claims as well. We follow the setting of [1], except the number of epochs, where we set the number of epoch to 30 instead of 55 for large-regularized LSTM since we observe the model starts to overfit after 30 epochs. Please refer to the following table.
>
> | Schedule |  Validation perplexity | Test perplexity |
> | ----- | ----- | ----- |
> | Inverse time decay |   115.9   |   113.6   |
> | Step Decay         |   82.3    |   79.1    |
> | Baseline in [1]    |   82.2    |   78.4    |
> | Cosine Decay       |   82.0    |   78.2    |
> | Elastic Step Decay | **80.9**  | **77.2**  |
>
> Here all scheduler (except baseline) goes through hyperparameter grid search and are selected with best validation perplexity, especially for step decay since its common version is unknown in language modeling. The number of intervals are searched in { 3, 4, 5, [logT], [logT] + 1 } with decay factor searched in { 2, 5, 10 }. Other details will be added in the revised version of our paper.
>
> **Q5: In the appendix, it's unclear what the extra term is. Adding a sentence to two would make things more clear.**
>
> **A5:** Thanks for pointing that out! We will add that in our revised version to make it more clear.
>
> ## References
> [1] Wojciech Zaremba, Ilya Sutskever, and Oriol Vinyals. "Recurrent neural network regularization". arXiv preprint arXiv:1409.2329 (2014).

---

> > ### Comment · Reviewer_obgH · 2021-12-05
> > **Thanks for the reply**
> >
> > I think this work offers an interesting theoretical result while also motivating and improving practical learning rate schedulers. I've read the other discussions, and the authors do a thorough job improving the manuscript and addressing concerns.  I am thus upgrading my score to an 8.

---

> > > ### Author Response · Authors · 2021-12-06
> > > **Thanks for the positive feedback!**
> > >
> > > Thanks very much for the recognition on our contributions and endeavors during the rebuttal process! It means a great deal to us.
> > >
> > > We would also like to offer our special thanks for this reviewer's encouragement in the comment from the very beginning. We really appreciate those kind words and positive comments. Many thanks!
> > >
> > > Regarding the post-rebuttal comment, we will check out the performance of this baseline on ImageNet. The result will be included in our next version of paper, along with the mentioned reference paper cited. If time and resources allow, we will also post the result under this comment.

---

### Official Review · Reviewer_qo4b · 2021-11-01

**Correctness:** 3
**Technical Novelty And Significance:** 3
**Empirical Novelty And Significance:** 3
**Recommendation:** 6
**Confidence:** 4

**Main Review:**

# Pros
+ The paper is pretty clear written.
+ Related literature from the SGD side is well explained.


# Cons
- I am not sure how interesting it is to embed Hessian information into the SGD stepsize design. The stepsize in Prop 1 is nothing but Newton's method in my perspective, which should be counted as a second order method, and it is unfair comparing this to SGD as a first order method.
- Therefore, the authors should explain the difference between the proposed method with Newton's method, and cite/compare with related works from (stochastic) Newton's method.
- More importantly, the empirical gain of using second order information is, in my perspective, marginal (see e.g., Table 2)...  which can hardly justify the extra cost of obtaining the Hessian information.
- The following comments are from the theory side ---- first of all, I would not say this paper contributes novel techniques or surprising results based on my knowledge of SGD literature; therefore I would not consider the presented theory as a sufficient contribution to justify the value of this work. Please correct me if important points are missing here.
- Prop 2 is only an upper bound for poly-decay stepsize, and this along cannot support the claim that poly-decay is sub-optimal. As least a minimax-lower bound is required. For such lower bounds of poly-decay stepsize, [Ge 2019] already presented a minimax version, and a more recent work [Wu 2021] has an instance-dependent lower bound. The authors should use a lower bound here & clearly cite existing results in their next verison.
- I am not sure how useful it is to compare the minimax rate. The authors want to claim one stepsize scheme is better than another. However the argued gain only happens in the worst case, while in benign instances, it is unknown whether or not the proposed stepsize scheme can have benefits. Perhaps conduct a instance-dependent analysis, and show a (nearly) instance-wise domination of one stepsize scheme over the other is more appealing.
- Even let us assume there are truly someone cares about minimax rates, I doubt the importance of removing a logarithmic factor...
- A final comment is that, comparing upper bound to upper bound simply cannot prove the benefit of one method over the other. The gain could be a loose analysis. A lower bound is needed to demonstrate the tightness of the analysis.





[Ge 2019] Ge, Rong, et al. "The step decay schedule: A near optimal, geometrically decaying learning rate procedure for least squares." arXiv preprint arXiv:1904.12838 (2019).

[Wu 2021] Wu, Jingfeng, et al. "Last Iterate Risk Bounds of SGD with Decaying Stepsize for Overparameterized Linear Regression." arXiv preprint arXiv:2110.06198 (2021).

**Summary Of The Paper:**

The paper studies convergence rates of SGD with different stepsize schemes, in the context of linear regression. For convergence of the last iterate of SGD, the best known result still misses a $\\log T$ factor compared to the minimax rate. This work aims to fill this gap with an improved stepsize scheme that utilizes the eigenvalue distribution of the Hessian. When the true Hessian is known, the proposed method successfully fills the gap, provably and in the sense of the worst problem instances. When the true Hessian is hard to know, practical variants are also proposed, and are shown to be comparable to the state-of-the-art stepsize schemes in standard deep neural network benchmarks.

**Summary Of The Review:**

I think the paper is clearly below the bar of acceptance. From the theory side, many points are not well supported (e.g., comparing upper bound to upper bound, comparing the minimax rate, etc.) and little technical novelty can be found. From the practice side, the improvement is too small to justify the cost of computing second order information.

---

> ### Author Response · Authors · 2021-11-17
> **Response to reviewer qo4b (part 1/2)**
>
> **Overall:** First, we would like to thank for all the comments this reviewer has made.
>
> However, we would also like to kindly remind the reviewer for several major incorrect arguments and some missed important aspects in the comment, which makes the overall comments wrong and unjustified.
>
> For example, the claim of us comparing upper bounds between two schedulers is simply wrong, since the lower bound for polynomial decay and step decay have already been mentioned or provided in our paper, in Remark 1 and Theorem 4 respectively. Furthermore, we would like to kindly remind the reviewer that, as we can cite the mentioned paper in our later versions, it _was_ impossible for us to cite it, since it was published in the future (2021-10-12) at the time we submitted (2021-10-06).
>
> Here we will address all the comments in details.
>
> **Q1: The author need to compare Eigencurve with Newton's method...**
>
> **A1:** First, Eigencurve is a general theoretical framework for discovering new scheduler. In practice, one normally directly use its derivatives, e.g. Elastic Step Decay and Cosine-power, which is still a first-order method, so it would be unfair to compare it with Newton's method.
>
> Second, Eigencurve only estimate the Hessian spectrum once, so it is normally an one-time effort. Furthermore, the estimated results can be transferred to similar models, as demonstrated in Appendix A.3 in our paper. On top of that, the estimation is not even happening in the original model, but a compressed model (see Appendix B.2). In the meantime, Newton's methods have the privilege of accessing second-order information from the original model during the training process on the fly. Comparing those two will be actually unfair for Eigencurve.
>
> Third, our paper's focus is to show that Eigencurve is both theoretically and empirically better than step decay, and has the potential to discover good schedulers with simple forms, which would be beneficial for the deep learning community. As we are talking about good/bad schedulers and optimality of schedulers, bringing Newton's method in would be a bit strange.
>
> **Q2: More importantly, the empirical gain of using second order information is...**
>
> **A2:** The results in main paper is simply a proof of concept, showing that Eigencurve can indeed outperform step decay empirically. For stronger results, we recommend the reviewer to check results of Elastic Step Decay in Appendix A.4, which is a derivative of Eigencurve.
>
> We would also like to add extra baselines/experiments for Elastic Step Decay. In ImageNet + ResNet50, with same settings in Appendix A.4, we have the following result,
>
> | Schedule |  Top-1 Test acc (%) | Top-5 Validation acc (%) |
> | -- | -- | -- |
> | Step Decay         |   75.63    |   92.66    |
> | Cosine Decay       |   76.67    |   93.28    |
> | Elastic Step Decay | **76.93**  | **93.32**  |
>
> In language modeling task with LSTM + Penn Treebank, under similar setting of [1], we have following results for large-regularized LSTMs,
>
> | Schedule |  Validation perplexity | Test perplexity |
> | -- | -- | -- |
> | Inverse time decay |   115.9   |   113.6   |
> | Step Decay         |   82.3    |   79.1    |
> | Baseline in [1]    |   82.2    |   78.4    |
> | Cosine Decay       |   82.0    |   78.2    |
> | Elastic Step Decay | **80.9**  | **77.2**  |
>
> One can observe that the gap between step decay and elastic step decay is non-trivial.
>
> **Q3: Prop 2 is only an upper bound for poly-decay stepsize...**
>
> **A3:** This is simply incorrect. The lower bound argument has already been provided in Remark 1 under Prop 2: "...In fact, the upper bound in Eqn. (2.5) is tight for the inverse time decay scheduling (Ge et al., 2019)", we would like to kindly remind the reviewer to check that out.
>
> Moreover, the lower bound of step decay has been provided in Theorem 4, which is not only an instance-wise lower bound, but also a general lower bound of almost all problem instances. The condition of H being diagonal can be further removed by considering the problem in a rotated space, as stated in "Main Intuition" in Section 2. So Eigencurve is actually strictly better than step decay (asymptotically), which has a worst upper bound with an extra log(kappa) term, while step decay has a lower bound with an extra log(T) term.
>
> What's more, when the Hessian spectrum is skewed, which is common for deep learning models, Eigencurve reaches minimax optimal convergence rate (up to a constant), and the gap between Eigencurve and step decay widens.
>
> **Q4: ...and a more recent work [Wu 2021] has an instance-dependent lower bound. The authors should use a lower bound here &...**
>
> **A4:** We would like to cite this paper in our revised version. But we also like to kindly remind the reviewer that it was impossible for us to cite that paper, since it is published in arXiv at 2021.10.12, while the ICLR submission deadline is 2021.10.6, which means we would have been citing a paper in the future.

---

> > ### Author Response · Authors · 2021-11-17
> > **Response to reviewer qo4b (part 2/2)**
> >
> > **Q5: ...The authors want to claim one stepsize scheme is better than another. However the argued gain only happens in the worst case...**
> >
> > **A5:** This is simply wrong. Please refer to our explanation for Q3.
> >
> > **Q6: Even let us assume there are truly someone cares about minimax rates, I doubt the importance of removing a logarithmic factor...**
> >
> > **A6:** The fact is that there indeed exists a gap between practical schedulers, e.g. cosine decay in Computer Vision and theoretically justified schedulers, e.g. step decay. No matter in which way we choose to view or think, this fact itself is unshakeable.
> >
> > Filling this gap and answering this question is not easy, so we make an attempt, in the perspective of thinking about the relationship between minimax rates and Hessian spectrums. Even this attempt may not directly lead to the true answer, and may only shed light to the answer of this question, we find this type of attempts to be necessary, since researches are all about exploring all the possibilities to find the true answer.
> >
> > The improvement in logarithmic factor may not be huge, but in practice, the improvements of cosine decay over step decay is also not as significant as improvements of step decay over inverse time decay, as displayed in Table 2. This does not mean the question itself is not important. In practice, even the smallest improvements in models can result in saving of hundreds of thoundsands of dollars in model training.
> >
> > **Q7: A final comment is that, comparing upper bound to upper bound simply cannot prove the benefit of one method over the other...**
> >
> > **A7:** This is simply wrong, as we have explained in Q3.
> >
> > **Q8: I think the paper is clearly below the bar of acceptance. From the theory side, many points are not well supported (e.g., comparing upper bound to upper bound, comparing the minimax rate, etc.) and little technical novelty can be found. From the practice side, the improvement is too small to justify the cost of computing second order information.**
> >
> > **A8:** We would like to say that most comments offered by the reviewer are incorrect and unjustified. Theoretically, we have already been comparing polynomial decay and step decay's lower bound to Eigencurve's upper bound (see Q3). Empirically, Eigencurve's derivatives are first order methods and can achieve non-trival improvements over step decay (see Q1 & Q2).
> >
> > ## References
> > [1] Wojciech Zaremba, Ilya Sutskever, and Oriol Vinyals. "Recurrent neural network regularization". arXiv preprint arXiv:1409.2329 (2014).

---

> > > ### Comment · Reviewer_qo4b · 2021-11-27
> > > **Thanks for the reply**
> > >
> > > Thanks for the reply.
> > >
> > > * [vs. Newton's method] Note that the main objective studied in this paper is still a **quadratic** loss where the Hessian is a constant, therefore the paper (e.g., Prop 1) is, at least in terms of theory, related to Newton's method for quadratic loss. Therefore I suggest the authors to discuss at least the relation of their method to Newton's method (and its variants, e.g., quasi-newton method where one uses a cheaper version of Hessian similar to this work).
> > >
> > > * [poly-decay stepsize] I think I am absolutely correct that **Prop 2 is only an upper bound**. Moreover, I believe I make no errant assertion that the lower bound for poly-decay stepsize is provided by **[Ge 2019] instead of this work** (also suggested by remark 2 in this paper). The authors need at least be more clear in terms of writing to avoid overclaiming their contribution.
> > >
> > > * [Thm4] In the reply the authors mentioned Theorem 4 is "instance-dependent". I am afraid that I cannot agree with this assertion. The lower bound in Thm 4 is clearly a "worst-case" bound, as there is no dependence on the spectrum of $H$. To be more precise, an "instance-dependent" bound should be understood as a bound that is sharp for every "problem" (i.e., every $H$) in a reasonably large problem class. To be "instance-dependent", at least the bound should reveal certain dependence on $H$.
> > >
> > > * [Wu 2021] To be more clear, I **did not** ask the authors to cite [Wu 2021] **in this version**, and I **did not** account this as a reason to reject the paper. In fact in my initial comment I clearly suggested that "[Wu 2021] is a more recent work". I mentioned [Wu 2021] because it provides an **instance-dependent** comparison between poly-decay stepsize vs. geometrical-decay stepsize, and the authors might find it related to this work. My only suggestion here is that the author may consider citing this work **in their next version**.
> > >
> > > * [comparing minimax rates] The authors did not answer this question. Let me be more precise on this one: a minimax rate refers to a bound for certain algorithm in its **worst-case* problem instance. Importantly, algorithm A in its **worst-case** outperforms algorithm B in its **worst-case** does not suggest algorithm 1 is better than algorithm 2 in every problem instances, not even in most problem instances. There are at least two issues comparing algorithms with their wrose-case performance.
> > >    - First, the worst-case problems for algorithm A and algorithm B might be different problems, and it makes little sense to compare two algorithm in two different problem instance.
> > >    - Second, often in practice, the problems instances to be solved are not the *worst-case ones*, and are more likely to be an **average-case**. Therefore the practice performance of an algorithm might not be accurately reflected by its "worst-case" rate.
> > >
> > >    To understand the actual performance between algorithm A and algorithm B, it is more useful to look at their performance for every problem instance in a reasonable problem class. This is why I think "problem-dependent" bound is important for this type of research.
> > >
> > > * [experiments] I appreciate the authors' efforts on the extensive amount of experiments. I would consider to raise the score due to this but the paper is still not ready to be published due the to reasons mentioned above.
> > >
> > >
> > > To sum up, the current version might not be good enough to be published in my humble perspective. Please consider re-wording/improving at least the theory part.

---

> > > > ### Author Response · Authors · 2021-11-28
> > > > **Thanks for the feedback**
> > > >
> > > > **Overall:**
> > > >
> > > > Thanks for the follow-up response! We really appreciate the reviewer's recognition on our experimental part, along with the effort in making comments clearer. We will definitely improve our writing regarding those mentioned minor problems.
> > > >
> > > > Currently, **the only remaining major issue** is the lower bound proved in Theorem 4, which is still misunderstood by the reviewer, so we would like to make some further responses to clarify this.
> > > >
> > > > **Q1 (minor): [vs. Newton's method]...**
> > > >
> > > > **A1:** As the reviewer no longer asks us to compare with Newton's methods, this comment seems quite fair now and we will certainly add relevant discussions in our next version.
> > > >
> > > > **Q2 (minor): [poly-decay stepsize]...**
> > > >
> > > > **A2:** Our rebuttal does not suggest that the reviewer's leading statement "Prop 2 is only an upper bound" is wrong. Instead, we point out that the main question raised in the corresponding paragraph of comment is unjustified, where the reviewer asks us to provide lower bounds for polynomial decay and step decay, which have already been mentioned or provided in Remark 1 and Theorem 4 in our paper.
> > > >
> > > > **Q3 (MAJOR): In the reply the authors mentioned Theorem 4 is "instance-dependent". I am afraid that I cannot agree with this assertion. The lower bound in Thm 4 is clearly a "worst-case" bound, as there is no dependence on the spectrum of $H$. To be more precise, an "instance-dependent" bound should be understood as a bound that is sharp for every "problem" (i.e., every $H$) in a reasonably large problem class. To be "instance-dependent", at least the bound should reveal certain dependence on $H$.**
> > > >
> > > > **A3:** Our lower bound in Thm 4 is **not** a "worst-case" bound. It indeed does not depend on the spectrum of $H$, but that is actually the strength of this theorem, not a flaw. It means this lower bound is quite general and holds for a wide range of problem instances for step decay without the need of referring to the specific forms of Hessian spectrums. This bound may not be tight in terms of constants, but it is already good enough for us to compare Eigencurve with step decay, since Eigencurve has an upper bound $\log \kappa$ and step decay has a lower bound $\log T$ for a wide range of problem instances.
> > > >
> > > > We recommend the reviewer to check the statement of Theorem 4, which clearly states that the lower bound holds **as long as**:
> > > > - (1) H is diagonal,
> > > > - (2) The equality in Assumption (1.7) holds, i.e. $\mathbb{E}_{\xi}[n_t n_t^\top] = \sigma^2 H$,
> > > > - (3) $T / \log T \ge \max(\kappa, 2^{16}, 16)$,
> > > >
> > > > where (1) can be further removed by considering the problem in a rotated space, as stated in "Main Intuition" in Section 2. And (2) can be further weakened to $\exists 0 < c < 1, \mathbb{E}_{\xi}[n_t n_t^\top] \succeq c \cdot \sigma^2 H$
> > > >
> > > > **Q4 (minor): [Wu 2021]...**
> > > >
> > > > **A4:** Since the mentioned comment is not counted as a reason of rejecting our paper, as made crystal clear by the reviewer here, we deem it as fair and would like to cite the mentioned paper in our next version.
> > > >
> > > > **Q5.1 (MAJOR) [comparing minimax rates]:...First, the worst-case problems for algorithm A and algorithm B might be different problems, and it makes little sense to compare two algorithm in two different problem instance.**
> > > >
> > > > **A5.1:** As we have clarified in Q3/A3, the lower bound provided in Theorem 4 is **not** a worst-case bound, but a general lower bound for step decay on a large set of problem instances. As long as a problem instance belongs to that problem class, we can compare the upper bound of Eigencurve and the lower bound of step decay, where Eigencurve is strictly better or at least no worse.
> > > >
> > > > Therefore, the comments made by the reviewer here are indeed based on incorrect presumptions.
> > > >
> > > > **Q5.2 (MAJOR) [comparing minimax rates]:...Second, often in practice, the problems instances to be solved are not the worst-case ones, and are more likely to be an average-case. Therefore the practice performance of an algorithm might not be accurately reflected by its "worst-case" rate.**
> > > >
> > > > **A5.2:** We are glad that the reviewer mentions practical cases, since in practice, our estimated Hessian spectrums on popular deep learning models turn out to have a high level of skewness and demonstrate "power power law", as shown in Section 4.1. In that case, Eigencurve reaches its **best case** and completely removes the $\log T$ term, while step decay still has an extra $\log T$ term.
> > > >
> > > > **Q6 (minor) [experiments]...**
> > > >
> > > > **A6:** Thanks! We would like to offer our appreciation to the reviewer for the recognition on our experimental part.
> > > >
> > > > **Q.summary (MAJOR): To sum up, the current version might not be good enough to be published in my humble perspective. Please consider re-wording/improving at least the theory part.**
> > > >
> > > > **A.summary:** As we discussed in Q3, Q5.1 and Q5.2, regarding the theory part, the questions raised by the reviewer are still unjustified. If the reviewer wished, we would like to engage in further discussions to make this part clearer.

---

> > > > > ### Comment · Reviewer_qo4b · 2021-11-28
> > > > > **Thanks for clarifying Theorem 4**
> > > > >
> > > > > Thanks for the further clarification of Theorem 4. I can see your point that Theorem 4 is not a lower bound for the worst-case problem instance; instead, it holds for a wide range of problem instances (with some requirement on the algorithm configuration), which is novel in my perspective. I would like to apologize on my previous miss-interpretation of this theorem.
> > > > >
> > > > > However I am still in doubt whether or not the current Theorem 4 is sufficient to claim the benefits of eigencurve over geometrical stepsize decay. Note that Theorem 4 has two important assumptions on the SGD setting:
> > > > > 1. The number of steps must be large $T \gtrsim \kappa $
> > > > > 2. The initial stepsize must be large $\eta_1 = 1/L$
> > > > >
> > > > > And Theorem 4 precisely says that, under these conditions, SGD with geometrical stepsize decay will pick up a large amount of variance $\propto d \sigma^2 \log (T) / T$.
> > > > >
> > > > > With that being said, it is also clear that the SGD variance can be, in general, reduced when one (1) reduces initial stepsize $\eta_1$ or (2) reduces number of steps $T$. Therefore there are at least two issues in Theorem 4 need to be discussed:
> > > > >
> > > > > 1. **A Large $T$** (minor issue)
> > > > >
> > > > >    In many applications the problem is not well conditioned and $\kappa$ can be rather large. Therefore $T\gtrsim \kappa$ limits the applicability of Theorem 4.
> > > > >
> > > > > 2. **A Large $\eta_1$** (major issue)
> > > > >
> > > > >    Note that the initial stepsize $\eta_1$ should be accounted as a **hyperparameter** of the algorithm itself, and $\eta_1 \eqsim 1/L$ might not be a good choice for every problem. When considering the performance of SGD with geometrical stepsize decay, the hyperparameter $\eta_1$ should be properly tuned (for example to balance bias vs. variance error according to the problem instance itself). Otherwise I tend to view Theorem 1 (Corollary 3) vs. Theorem 4 as an unfair comparison between eigencurve and geometrical stepsize decay.
> > > > >
> > > > >
> > > > >
> > > > > In sum, I recognize that, in certain (though perhaps limited) regime, Theorem 1 (Corollary 3) vs. Theorem 4 shows an improvement of eigencurve over geometrical stepsize decay. However the comparison is not fair to the latter algorithm, as its hyperparameter is not properly tuned (at least in term of theoretical analysis). I am raising the score to 5 to appreciate the contribution that I have missed previously. I am happy to consider a further raising of the score if the author could properly comment on the two newly mentioned issues (especially the second one).

---

> > > > > > ### Author Response · Authors · 2021-11-29
> > > > > > **Thanks for the positive feedback**
> > > > > >
> > > > > > **Overall:**
> > > > > >
> > > > > > Thanks for the positive feedback! We really appreciate the effort this reviewer has put into to go through the whole rebuttal/discussion process with us to make things much clearer.
> > > > > >
> > > > > > Even our lower bound in Theorem 4 is already sufficient to demonstrate the fact that Eigencurve is theoretically better than step decay in a large set of problem classes, we would like to provide stronger results to strengthen this argument. Regarding the newly issues, we would like to offer following explanations to address the reviewer's concern.
> > > > > >
> > > > > > **Q1 (minor): A Large $T$...**
> > > > > >
> > > > > > **A1:** This question is similar to reviewer 7X3r's question Q1. When $T \le \kappa$, the variance term will no longer be the dominated term (at least for small eigenvalues), hence comparing only the variance term will be theoretically meaningless. In that case, the bias terms have to be taken into account, which are large constants for both step decay and Eigencurve (at least for small eigenvalues). Therefore theoretically it is more reasonable to focus on the case of $T > \kappa$.
> > > > > >
> > > > > > **Q2 (major): A Large $\eta_1$...**
> > > > > >
> > > > > > **A2:** According to our proof of Theorem 4 in Appendix G.4, we only utilize the fact that the learning rate sequence {$ \eta_t $} includes the bottleneck interval region, which is the first $\left(\frac{T}{\log T}\right)$-length interval $\left[1 + \frac{T}{\log T} \cdot \ell, \frac{T}{\log T} \cdot (\ell + 1) \right]$ in step decay that satisfies
> > > > > >
> > > > > > $2^{\ell} \ge \frac{\lambda_j}{L} \cdot \frac{8T}{\log T}$
> > > > > >
> > > > > > $\Leftrightarrow \frac{\log T}{8 \lambda_j T} \ge \frac{1}{2^{\ell} L}$
> > > > > >
> > > > > > $\Leftrightarrow \frac{\log T}{8 \lambda_j T} \ge \eta_t  \text{ for some } t$
> > > > > >
> > > > > > So with almost no change, our proof can be extended to cases of $\eta_0 < \frac{1}{L}$ as long as $\eta_0 \ge \frac{\log T}{8 \mu T}$. For large $T$, $\eta_0$ can be very small, so this addresses the raised concern to a certain degree. A similar case (without the factor 8) can be also proved by using Theorem 2 in [Wu 2021], with $s=0$ and $\gamma_0 \ge \frac{1}{\mu K}$, which entails $k^* = d$ according to its definition. Moreover, if $\eta_0 < \frac{\log T}{8 \mu T}$, the bias term may become a bottleneck of step decay, at least for small eigenvalues, so further reducing the learning rate may not be a good choice.
> > > > > >
> > > > > > On top of that, this idea can be further extended to prove step decay's lower bound for $\forall \eta_0 < \frac{1}{L}$, where eigenvalues {$\lambda_j$} can be divided into two groups: 1) $\eta_0 \ge \frac{\log T}{8 \lambda_j T}$; 2) $\eta_0 < \frac{\log T}{8 \lambda_j T}$. For the first group, the bottleneck interval $\ell$ is well-defined, and the variance term will have an extra $\log T$ term for that dimension. For the second group, the bias term will be the bottleneck since $\sum_t \lambda_j \eta_t \le \lambda_j \cdot \eta_0 \cdot \frac{T}{\log T} \cdot (1 + \frac{1}{2} + ...) < 1/4$ and a constant
> > > > > >
> > > > > > $\lambda_j (w_{0,j} - w_{*,j})^2 \prod_t (1 - \lambda_j \eta_t)^2$
> > > > > >
> > > > > > $\ge \lambda_j (w_{0,j} - w_{*,j})^2 (1 - \sum_t 2 \lambda_j \eta_t)$
> > > > > >
> > > > > > $\ge \lambda_j (w_{0,j} - w_{*,j})^2 \cdot \frac{1}{2}$
> > > > > >
> > > > > > will remain in the bias term. With sufficiently large $T$, the combination of both will be $\Omega\left(\frac{d \sigma^2}{T} \cdot \log T\right)$.
> > > > > >
> > > > > > As our paper is focused on Eigencurve scheduler, we adopt a simpler setting and proof for step decay's lower bound in Theorem 4, which is provided just to demonstrate that Eigencurve can outperform step decay in a reasonably large problem classes under common settings. For more detailed analysis of step decay's lower bound, the reviewer may refer to [Wu 2021], whose result will be cited and included in our next version.
> > > > > >
> > > > > > ## Reference
> > > > > > [Wu 2021] Wu, Jingfeng, et al. "Last Iterate Risk Bounds of SGD with Decaying Stepsize for Overparameterized Linear Regression." arXiv preprint arXiv:2110.06198 (2021).

---

> > > > > > > ### Comment · Reviewer_qo4b · 2021-11-30
> > > > > > > **Thanks for the reply; tend to stay neutral to this paper**
> > > > > > >
> > > > > > > I am in fact confused with the authors' latest reply. The main issue is about the dependence on $w\^*$.  For simplicity let us set $w_0 = 0$.
> > > > > > >
> > > > > > > In the answer to Q1, the authors seem to suggest when $T < \kappa$ the bias error must a constant. However this is only true for worst $w^*$. When $w^*$ is well distributed, e.g., concentrates along the large eigenvalue directions, I do not think the bias error will be a constant even when $T < \kappa$.
> > > > > > >
> > > > > > > Similarly, in the answer to Q2, the authors seem to suggest the bias error is a bottleneck, however it really depends on how $w^*$ distributes.
> > > > > > >
> > > > > > > On the one hand, Theorem 4 is widely applicable that accommodates a set of data covariances, given strong assumptions on $T$ and $\eta_1$ (which in my perspective is not fair to SGD with geometrical decaying stepsize).
> > > > > > >
> > > > > > > On the other hand, based on my understanding of the authors' reply, Theorem 4 is only tight for worst $w^*$ if SGD with geometric decaying is allowed to be tuned with the hyeprameterater $\eta_1$ or when $T < \kappa$.
> > > > > > >
> > > > > > >
> > > > > > > At this point, I no longer hold a strong position to reject the paper, as the work has its spirit at least empirically. However the theory part seems to be less supportive. I fail to find a strong reason to vote for accepting the paper. Overall I tend to stay in a neutral position.

---

> > > > > > > > ### Author Response · Authors · 2021-11-30
> > > > > > > > **Thanks for your feedback; just want to point out some important aspects to make things clearer**
> > > > > > > >
> > > > > > > > Thanks for the reviewer's feedback. However, we would like to point out several important aspects in our proofs to make it clearer.
> > > > > > > >
> > > > > > > > **First, the lower bound proof is not the main focus of our paper.**
> > > > > > > >
> > > > > > > > We simply need to demonstrate that Eigencurve outperforms step decay in a reasonably large problem class. Furthermore, our assumption of large $T$ and large learning rates are _neither uncommon nor strong_ for practical settings. For example, in applications of pretraining models with large datasets, especially in industry, one will normally have a very large $T$. And in applications of training LSTM on penntreebank or training ResNet on CIFAR-10, the learning rate is usually at least 0.1 to achieve good performance if SGD without momentum is adopted, which is larger than or at least the same magnitude as $1/L$.
> > > > > > > >
> > > > > > > > **Second, as proved in our last reply, we have almost perfectly addressed Q2 in an asymptotic sense, so the reviewer's argument is simply incorrect.**
> > > > > > > >
> > > > > > > > If $T$ is sufficiently large, the assumption of $\eta_1 = 1/L$ is no longer needed and can be extended to $\eta_1 < 1/L$ regardless of the distribution of $w_*$, as long as $\lambda_j (w_{\*,j} - w_{0,j})^2 \not= 0$  for $\forall j = 1, 2, \dots, d$, since $\lambda_j (w_{\*,j} - w_{0,j})^2 \ge \frac{\sigma^2 \log T}{T}$ will always hold for sufficiently large $T$. Therefore, the reviewer's whole "worst $w_*$" argument is _simply incorrect_ for Q2.
> > > > > > > >
> > > > > > > > **Third, as reviewer said in the 3rd round of comments, Q1 is a minor issue when compared to Q2**.
> > > > > > > >
> > > > > > > > Plus, we have offered some reasonable arguments for Q1, which holds at least for small eigenvalues and can be applicable for settings where the Hessian spectrum has a large amount of small eigenvalues, e.g. skewed Hessian spectrums that satisfy "power power law". In those cases, the bias term will be the bottleneck as long as $(w_{\*,j} - w_{0,j})^2 \ge \frac{c \kappa}{d}$ for some constants $c$ when $T < \kappa$. So it is certainly _not_ just those "worst $w_*$" the reviewer mentioned in the last reply.
> > > > > > > >
> > > > > > > > Finally, we would like to kindly remind the reviewer that 5 point is not a neutral position, but favors for rejection.
> > > > > > > >
> > > > > > > > We hope this can make our arguments clearer and highlight our contribution in the paper.

---

> > > > > > > > > ### Comment · Reviewer_qo4b · 2021-12-01
> > > > > > > > > **Disagree with the authors**
> > > > > > > > >
> > > > > > > > > * The authors have devoted Sections 2 and 3 to justify the theoretical advantages of their proposed algorithm. Among them Theorem 4 is **crucial** to justify why eigencurve is better than geometric decaying stepsize. My concern is that the current Theorem 4 is **unfair** to geometric decaying stepsize (see my previous comment). If the authors cannot address this issue but simply walk round it, I would have to value less on the theory part of this work (which in my perspective is one of the main focus of this work as it takes takes at least 3-4 pages).
> > > > > > > > >
> > > > > > > > > * The assumption $T > \kappa$ is not practical, especially for problems that are not strongly convex, e.g.,  neural networks where $\kappa \to 0$. If the authors disagree, perhaps they can list some concrete references to convince me.
> > > > > > > > >
> > > > > > > > > * I do not think the authors addressed my Q2!!! In particular, can you explain why $\\lambda\_i (w\_{0,j} - w\_{\*,j})\^2$ is a constant? Or $> \\sigma\^2 \\log T / T$? First of all, it is possible that $\\lambda\_i $ is small (e.g., there is un-useful features that correspond to zero eigenvalue in the covariance). Second, it is possible that $w\_{0,j} - w\_{\*,j}$ is small (e.g., $w\_0 = 0$ and the label does not depend on certain feature so the corresponding component in $w\^\*$ is zero too).

---

> > > > > > > > > > ### Author Response · Authors · 2021-12-01
> > > > > > > > > > **5th round response - Thanks for the reply (part 1/3)**
> > > > > > > > > >
> > > > > > > > > > We really appreciate that the reviewer can reply to our response even after the discussion period. Please calm down and be less emotional. Let's discuss this rationally to make things clearer.
> > > > > > > > > >
> > > > > > > > > > **Q1: The authors have devoted Sections 2 and 3 to justify the theoretical advantages of their proposed algorithm. Among them Theorem 4 is crucial to justify why eigencurve is better than geometric decaying stepsize. My concern is that the current Theorem 4 is unfair to geometric decaying stepsize (see my previous comment). If the authors cannot address this issue but simply walk round it, I would have to value less on the theory part of this work (which in my perspective is one of the main focus of this work as it takes takes at least 3-4 pages).**
> > > > > > > > > >
> > > > > > > > > > **A1:** As we certainly mentions that Eigencurve is better than step decay in a wide range of settings, our main focus in this paper is still the theoretical and empirical property of Eigencurve for skewed Hessian spectrums, where Eigencurve removes the $\log T$ factor and achieves good empirical performance. Please refer to our title and abstract, where we didn't mention a single word about Eigencurve being theoretically better than step decay.
> > > > > > > > > >
> > > > > > > > > > Of course we would be happy that Eigencurve can outperform step decay in every possible settings, and would like to discusss more about it. We are certainly not evading answering this question, as we keep clarifying or improving Theorem 4 to address the reviewer's concern during the rebuttal process. We just hope the reviewer could pay more attention to the main contribution of our paper instead of just concentrating on Theorem 4.
> > > > > > > > > >
> > > > > > > > > > **Q2: The assumption $T > \kappa$ is not practical...**
> > > > > > > > > >
> > > > > > > > > > **A2:** The assumption $T > \kappa$ is practical.
> > > > > > > > > >
> > > > > > > > > > First, for a fixed neural networks and dataset that has well-defined $\kappa$'s around stationary points (which are also local optimums since $\kappa$'s are well-defined), once the parameter $w_t$ enters the nearby region of those stationary points, the value of $\kappa$ will be upper bounded by a fixed constant no matter how long we further train the model. So if we train the model long enough, $T$ will always be sufficiently large to surpass $\kappa$. For example, in industry, large tech companies train models on months or even years of collected data, and normally the training process will be extremely long, even without stopping (since higher accuracy in recommendation systems/search engines will directly link to the revenue of those companies). In those cases, $T$ will be very large, and eventually surpass $\kappa$.
> > > > > > > > > >
> > > > > > > > > > Second, for tasks with strong L2 regularizations, or large weight decays (if using common gradient-based optimizers excluding adaptive methods), the value of $\kappa$ will be even smaller.
> > > > > > > > > >
> > > > > > > > > > Third, discussion about $\mu \rightarrow 0$, i.e. $\kappa \rightarrow \infty$ is beyond the scope of quadratic objectives theoretically. It is impossible to discuss $T \le \kappa$ or $T > \kappa$ if $\kappa$ is not well-defined, so the example the reviewer proposed is simply not suitable here. In that case, we will be talking about general smooth convex functions, which is irrelevant to the theory part of our paper. Of course, that is an interesting question, and we would be happy to explore that topic in future researches.
> > > > > > > > > >
> > > > > > > > > > Fourth, as the reviewer suggest us to provide some concrete practical cases where $T > \kappa$, we would like to do it here. Of course, for general non-convex non-smooth deep learning models, it is almost impossible to find such examples and will be an unreasonable demand. So we focus on those datasets that are quadratic or at least strongly convex. Due to time constraints, we have to run these additional experiments on small datasets. In [LIBSVM dataset](https://www.csie.ntu.edu.tw/~cjlin/libsvmtools/datasets/), we estimate the largest eigenvalue $L$ of dataset abalone_scale for ridge regression and a4a for logistic regression around optimum (using gradient descent to train 2M iterations), which is around $L \approx 3.71$ and $L \approx 0.82$ respectively. With L2 coefficient $10^{-3}$, the value of $\kappa$ is at most $3.71 \times 10^3$ and $8.2 \times 10^2$. As a reference, the first dataset has around $4.2 \times 10^3$ samples and the second dataset has around $4.8 \times 10^3$ samples. Running vanilla SGD with at least 1 epoch results in $T > \kappa$. Furthermore, one may run multiple epochs with smaller L2 coefficient or mini-batch SGD and still have $T > \kappa$ satisfied.
> > > > > > > > > >
> > > > > > > > > > By the way, we suppose that when the reviewer says $\kappa \rightarrow 0$, the reviewer means $\kappa \rightarrow \infty$.

---

> > > > > > > > > > > ### Author Response · Authors · 2021-12-01
> > > > > > > > > > > **5th round response - Thanks for the reply (part 2/3)**
> > > > > > > > > > >
> > > > > > > > > > >
> > > > > > > > > > > **Q3.1: I do not think the authors addressed my Q2!!! In particular, can you
> > > > > > > > > > > explain why $\lambda_i (w_{0,j} - w_{\*,j})^2$ is a constant? Or $> \sigma^2 \log T / T$?**
> > > > > > > > > > >
> > > > > > > > > > > **A3.1:** We have addressed Q2 (in 3rd round comments) almost perfectly in an asymptotic sense and would be glad to discuss more details to clarify our statement.
> > > > > > > > > > >
> > > > > > > > > > > Roughly speaking, for a given problem instance with initialization $w_0$, all $\lambda_j (w_{0,j} - w_{\*,j})^2$ are fixed, thus can be treated as constants when compared to $T$ asymptotically.
> > > > > > > > > > >
> > > > > > > > > > > Rigorously speaking, we have proved the following extended Theorem:
> > > > > > > > > > >
> > > > > > > > > > > **Theorem 4 (extended).** Given an instance of objective function $f(x)$ being quadratic, with initialized parameters $w_0$. We run SGD for $T$-steps starting from $w_0$ and a step decay learning rate sequence {$\eta_t$} defined in Algorithm 1 of [1], with $\eta_1 < 1/L$. As long as (1) H is diagonal, (2) The equality in Assumption (1.7) holds, i.e. $E_{\xi}[n_t {n_t}^\top ] = \sigma^2 H$, (3) $\lambda_j (w_{0,j} - w_{\*,j})^2 \neq 0$ for $\forall j = 1, 2, ..., d$, then the final iterate $w_{T+1}$ satisfies,
> > > > > > > > > > >
> > > > > > > > > > >   $\mathbb{E}[f(w_{T+1}) - f(w_*)] = \Omega\left(\frac{d\sigma^2}{T} \cdot \log T\right)$
> > > > > > > > > > >
> > > > > > > > > > > Proof: We can divide all eigenvalues { $ \lambda_j $ } into two groups: 1) $\eta_1 \ge \frac{\log T}{8 \lambda_j T}$; 2) $\eta_1 < \frac{\log T}{8 \lambda_j T}$. The first group are those large eigenvalues that the variance term will finally dominate, and the second group are those small eigenvalues that the bias term will finally dominate, as we have proved in our 3rd round response. To make things clearer, we will prove it here again.
> > > > > > > > > > >
> > > > > > > > > > > Regarding the first group, we can utilize our original proof of Theorem 4, since the only restriction on $\eta_1$ is to assure that the bottleneck interval $\ell$ is well-defined. For this first group, it is indeed the case, since
> > > > > > > > > > >
> > > > > > > > > > > $2^{\ell} \ge \frac{\lambda_j}{L} \cdot \frac{8T}{\log T}$
> > > > > > > > > > >
> > > > > > > > > > > $\Leftrightarrow \frac{\log T}{8 \lambda_j T} \ge \frac{1}{2^{\ell} L}$
> > > > > > > > > > >
> > > > > > > > > > > $\Leftrightarrow \frac{\log T}{8 \lambda_j T} \ge \eta_t  \text{ for some } t$
> > > > > > > > > > >
> > > > > > > > > > > will hold since $\frac{\log T}{8 \lambda_j T} \in \left[\frac{1}{LT}, \eta_1\right] \subseteq [\frac{\eta_1}{T}, \eta_1]$ for $T \ge 2^{16}$. Thus we will have variance terms $\frac{\lambda_j^2 \sigma^2}{2} \cdot v_j \ge \frac{\lambda_j^2 \sigma^2}{2} \cdot \frac{1}{512} \cdot \frac{\log T}{T} \cdot \frac{1}{\lambda_j^2} = \frac{1}{1024} \cdot \frac{\sigma^2}{T} \cdot \log T $, where $v_j$ is defined in the statement above Eqn. (G.20).
> > > > > > > > > > >
> > > > > > > > > > > Regarding the second group, the variance is indeed smaller, but the bias will come to play since the eigenvalue will be too small to reduce bias. According to proof of our Lemma 6, for diagonal $H$, we have the bias term being $\lambda_j (w_{0,j} - w_{\*,j})^2 \cdot \prod_{k=0}^T (1 - \eta_k \lambda_j)^2$ for any scheduler {$\eta_t$}. In [1], the learning rate sequence starts from $\eta_1$ instead of $\eta_0$, so this term is translated to $\lambda_j (w_{0,j} - w_{\*,j})^2 \cdot \prod_{k=1}^T (1 - \eta_k \lambda_j)^2$ accordingly.
> > > > > > > > > > >
> > > > > > > > > > > Since $\eta_1 < \frac{\log T}{8 \lambda_j T}$, we have
> > > > > > > > > > >
> > > > > > > > > > > $bias_j = \lambda_j (w_{0,j} - w_{\*,j})^2 \cdot \prod_{k=1}^T (1 - \eta_k \lambda_j)^2$
> > > > > > > > > > >
> > > > > > > > > > > $\ge \lambda_j (w_{0,j} - w_{\*,j})^2 \cdot (1 - \sum_{k=1}^T 2 \eta_k \lambda_j)$
> > > > > > > > > > >
> > > > > > > > > > > $\ge \lambda_j (w_{0,j} - w_{\*,j})^2 \cdot \frac{1}{2}$,
> > > > > > > > > > >
> > > > > > > > > > > where the first inequality is caused by $(1 - a_1)(1 - a_2) \ge (1 - a_1 - a_2)$ and applying mathematical induction for { $a_n$ } to obtain $\prod_{k=1}^n (1 - a_k) \ge (1 - \sum_{k=1}^n a_k)$ as long as $\sum_{k=1}^n a_k < 1$. The second inequality is entailed by the property of second group, where $\sum_{k=1}^T \lambda_j \eta_k \le \lambda_j \cdot \eta_1 \cdot \frac{T}{\log T} (1 + \frac{1}{2} + ...) < 1/4$ for step decay scheduler in [1].
> > > > > > > > > > >
> > > > > > > > > > > According to Assumption (3) in this extended theorem, $\lambda_j (w_{0,j} - w_{\*,j})^2 \not= 0$, so we have $c \triangleq \min_j bias_j = \min_j \frac{1}{2} \cdot \lambda_j (w_{0,j} - w_{\*,j})^2 > 0$ be a fixed constant. Thus with sufficiently large $T$, we have $c \ge \frac{1}{1024} \cdot \frac{\sigma^2}{T} \cdot \log T$.
> > > > > > > > > > >
> > > > > > > > > > > Combine conclusions of both groups, we have
> > > > > > > > > > >
> > > > > > > > > > > $\mathbb{E}[f(w_{T+1}) - f(w_\*)] = \sum_{j=1}^d bias_j + \frac{\sigma^2}{2} \sum_{j=1}^d \lambda_j^2 v_j$
> > > > > > > > > > >
> > > > > > > > > > > $\ge d \cdot \frac{1}{1024} \cdot \frac{\sigma^2}{T} \cdot \log T$
> > > > > > > > > > >
> > > > > > > > > > > $= \Omega\left(\frac{d\sigma^2}{T} \cdot \log T\right)$.
> > > > > > > > > > >
> > > > > > > > > > > Notice that this proof no longer requires $T / \log T \ge \max(\kappa, 2^{16}, 16)$ since the proof technique can be extended to remove this assumption as well, as long as we are talking about asymptotic cases, where $T$ can be sufficiently large.

---

> > > > > > > > > > > > ### Author Response · Authors · 2021-12-02
> > > > > > > > > > > > **5th round response - Thanks for the reply (part 3/3)**
> > > > > > > > > > > >
> > > > > > > > > > > >
> > > > > > > > > > > > This extended Theorem 4 can be further extended to cases where $w_0$ is not given. For example, with randomly initialized $w_0$ subject to uniform distribution $[-a, a]$ in each dimension, the probability of $(w_{*,j} - w_{0,j})^2 \ge \left(\frac{a \epsilon}{d}\right)^2$ for all dimension $j = 1, 2, ..., d$ will be at least $p=\left(\frac{2a - 2a \epsilon/d)}{2a}\right)^d = \left(1 - \frac{\epsilon}{d}\right)^d \ge 1 - \epsilon$ for $\forall 0 < \epsilon < 1$. So given a fixed $\epsilon$, with probability $p \ge 1 - \epsilon$, $\lambda_j (w_{\*,j} - w_{0,j})^2 \ge \mu \left(\frac{a \epsilon}{d}\right)^2$ holds for $\forall j = 1, 2, ..., d$, which is a fixed constant and will be smaller than $\frac{\sigma^2}{T} \cdot \log T$ with sufficiently large $T$. By choosing $\epsilon$ to be a small constant, we have this asymptotic lower bound hold for step decay with high probability $p\ge 1 - \epsilon$, given randomly initialized $w_0$.
> > > > > > > > > > > >
> > > > > > > > > > > > **Q3.2: First of all, it is possible that $\lambda_i$ is small (e.g., there is
> > > > > > > > > > > > un-useful features that correspond to zero eigenvalue in the covariance).
> > > > > > > > > > > > Second, it is possible that $w_{0,j} - w_{\*,j}$ is small (e.g., $w_0 = 0$ and
> > > > > > > > > > > > the label does not depend on certain feature so the corresponding component in
> > > > > > > > > > > > $w_*$ is zero too).**
> > > > > > > > > > > >
> > > > > > > > > > > > **A3.2:** First, it is possible that $\lambda_i$ is small. But under our quadratic objective context, once the objective is given, $\lambda_i$ becomes a fixed constant when compared to horizon $T$ asymptotically. As for the reviewer's proposed counterexample, it is simply unjustified, since according to Lemma 5 and Lemma 6 in our paper, zero eigenvalues contribute nothing to the loss, no matter in terms of bias or variance. So any well-defined and meaningful examples should have no zero eigenvalues.
> > > > > > > > > > > >
> > > > > > > > > > > > Second, again, it is possible that $w_{0,j} - w_{\*,j}$ is small, but once $w_0$ is given, it becomes a fixed constant when compared to horizon $T$ asymptotically. In addition, the reviewer's counterexample is improper since features that are not updated during the whole training process should be ignored because of their zero contribution to the loss theoretically. By the way, it is possible that the label does not depend on certain features on most samples, given the existence of sparse features in common practical settings. But as long as the feature at least occurs once in some data samples, it will be updated eventually and $w_*$ will no longer be zero.
> > > > > > > > > > > >
> > > > > > > > > > > > **Final Remark:**
> > > > > > > > > > > >
> > > > > > > > > > > > We feel a bit confused about the reviewer's aggressiveness towards our paper and comments. If there is anything unclear, we would be happy to clarify it. Let's discuss it peacefully. Throwing a tantrum won't achieve anything.
> > > > > > > > > > > >
> > > > > > > > > > > > ## Reference
> > > > > > > > > > > > [1]: [Ge 2019] Ge, Rong, et al. "The step decay schedule: A near optimal, geometrically decaying learning rate procedure for least squares." NeurIPS 2019.

---

> > > > > > > > > > > > > ### Comment · Reviewer_qo4b · 2021-12-02
> > > > > > > > > > > > > **Thanks for the reply; Theorem 4 (extended) is now crystal clear**
> > > > > > > > > > > > >
> > > > > > > > > > > > > Dear authors, I am sincerely thankful for the clarification to Theorem 4, which now looks crystal clear to me.
> > > > > > > > > > > > >
> > > > > > > > > > > > > Indeed, the condition
> > > > > > > > > > > > >
> > > > > > > > > > > > > (\*) $\\lambda\_i (w\_{t,i} - w\_{*,i})\^2 \ne 0$ (and is a fixed constant independent of $T$)
> > > > > > > > > > > > >
> > > > > > > > > > > > > should be regarded as an **assumption** instead of part of the conclusion. I was confused in your third round reply mainly due to that you seemed to suggest (\*) holds without any assumptions, which is incorrect in my perspective.
> > > > > > > > > > > > >
> > > > > > > > > > > > > The lastest version of Theorem 4 (extended) addresses my concerns on the theory part. Moreover the authors provided practical motivations to condition (\*). I hope the authors could revise Theorem 4 accordingly, and summarize and include these discussion in their revision. It is, finally, a pleasure of mine to raise the score to 6.

---

> > > > > > > > > > > > > > ### Author Response · Authors · 2021-12-03
> > > > > > > > > > > > > > **Thanks for the positive feedback!**
> > > > > > > > > > > > > >
> > > > > > > > > > > > > > Thanks very much for the recognition on our theoretical contribution! Also, we would like to offer our gratitude for those valuable suggestions the reviewer has proposed during the rebuttal process, which helps us improve Theorem 4 and obtain a better version of it.
> > > > > > > > > > > > > >
> > > > > > > > > > > > > > We are glad that Theorem 4 (extended) has cleared up the reviewer's doubts about our theory part. This extended version of Theorem 4 will definitely be included in our next version, either in replace of Theorem 4 if area chairs think the revision won't affect the decision of accept/reject, or in Appendix otherwise. Other relevant discussions will be summarized and included in the Appendix.
> > > > > > > > > > > > > >
> > > > > > > > > > > > > > Again, it is a pleasure for us to have this discussion with you during the rebuttal process and we would like to thank for the effort the reviewer has put into to help us clarify and improve our contribution.

---

### Official Review · Reviewer_vptL · 2021-11-02

**Correctness:** 3
**Technical Novelty And Significance:** 2
**Empirical Novelty And Significance:** 2
**Recommendation:** 6
**Confidence:** 3

**Main Review:**

Overall, the paper is easy to follow and well-motivated.

Strengths:
- The derived learning rate schedule is theoretically sound and the intuition is well-conveyed. It also provides some justifications for the effectiveness of the cosine decay schedule.

Weaknesses:
- In terms of optimal learning rate schedule for convex-quadratics, it was previously discussed in [1] with a noisy quadratic model. In particular, [1] directly optimized the schedule as they can compute the exact loss using dynamic programming. In principle, one can get an optimal (or near-optimal) learning rate schedule in this model by plugging the Hessian spectrum, then use it to train neural networks. Given that, some discussions or comparisons would be valuable in the main paper.
- The derived theoretical results are all about asymptotic convergence. To be honest, I’m not sure how relevant that would be. Of course, it is an interesting contribution from a theoretical perspective. But in practice, we train neural networks with fixed epochs and I would expect the learning rate schedule to depend on the gradient noise level $\sigma$. It seems that the proposed schedule is independent of $\sigma$. I think the authors might like to add some discussions on that.
- As the authors mentioned in the paper, Polyak averaging is optimal for convex quadratics but is unable to handle nonconvex landscapes in practice. However, I believe the exponential moving averaging method is commonly used for training neural networks, so I encourage the authors to include the comparison with the exponential moving average.
- The 10-epoch results are a bit artificial. With only 10 epochs, it makes little sense to decay the learning rates twice for the step-decay schedule. It seems that the authors used different parameters (e.g. \beta) for the 10-epoch setting. Although the authors argue that \beta is not a hyperparameter, but it is obvious to me that the value of 1.000005 is carefully tuned. Could the authors justify this particular choice?
Overall, the improvement of using the proposed Eigencurve is marginal on image classification tasks, though this is understandable as the step-decay and cosine-decay schedules are very strong baselines.

Minor points:
- The authors could save a lot of space by reorganizing all the figures in Figure 4. And with the saved space, I suggest the authors to include the details of hyperparameter tuning in the main paper.


[1] Which Algorithmic Choices Matter at Which Batch Sizes? Insights From a Noisy Quadratic Model. NeurIPS 2019.

**Summary Of The Paper:**

This paper studies the convergence of SGD with different learning rate schedules and aims to achieve minimax optimal convergence rates on quadratic objectives. To this end, this work proposes a new learning rate schedule (named Eigencurve) based on the Hessian spectrum and provides an optimal last-iterate convergence rate.
The proposed Eigencurve gives rise to slightly improved performance on image classification tasks with deep neural networks. In addition, the introduced schedule is similar to the popular cosine learning rate schedule on some problems, which, to some extent, justifies the effectiveness of the cosine schedule.

**Summary Of The Review:**

The paper proposed an interesting learning rate schedule that achieves an optimal asymptotic rate on strongly-convex quadratics. However, the schedule relies on first computing the Hessian spectrum and the empirical improvement is not significant.

Overall, this is a borderline paper and I'm inclined to reject the paper (though I'm willing to increase my score if the authors can address my comments).

---

> ### Author Response · Authors · 2021-11-17
> **Response to reviewer vptL (part 1/2)**
>
> **Overall:** First, we would like to appreciate for all the constructive comments this reviewer has made.
>
> To answer the reviewer's questions, we have added additional experimental results of exponential moving average and displayed learning rate curves for different choices of beta. Furthermore, extra results in Language Modeling task are available for Elastic Step Decay. We have also added discussions related of Eigencurve's difference from near-optimal schedulers computed by dynamic programming.
>
> The details are available as follows.
>
> **Q1: In terms of optimal learning rate schedule for convex-quadratics...**
>
> **A1:** Yes. Dynamic programming can indeed find numerically near-optimal schedulers given the exact loss. Nevertheless, Eigencurve still possesses several advantages over this kinds of approaches.
>
> First, Eigencurve can be used to find simple-formed schedulers. Compared to schedulers numerically computed by dynamic programming, our method provide an analytic framework so it is able to bypass the Hessian spectrum estimation if we can obtain some useful assumptions of the Hessian spectrum to simplify the form of Eigencurve, such as "power power law".
>
> Second, Eigencurve has a clear theoretical convergence guarantee. Dynamic programming can find near-optimal schedulers, but the convergence property of the computed scheduler is still not clear. Our work fills this gap, at least for problem instances with skewed Hessian spectrums.
>
> The aboved discussion will be added to later versions of our paper.
>
> **Q2: The derived theoretical results are all about asymptotic convergence...**
>
> **A2:** The results are non-asymptotic for quadratic objectives. As for NN, we focus on asymptotic convergence in this paper, since more results may make the paper too long to be easily understood, so we leave it to future researches.
>
> **Q3: As the authors mentioned in the paper, Polyak averaging is optimal for...**
>
> **A3:** Yes. Exponential moving average is commonly used in deep learning, but the performance is normally not on par with a good scheduler. Our additional experimental results turn out to support this statement as well, as shown in following tables. The 10-epoch result is as follows,
>
> || ResNet-18 || GoogLeNet || VGG16 ||
> | -- | -- | -- | -- | -- | -- | -- |
> | Schedule |  Training loss | Test acc (%) | Training loss | Test acc (%) | Training loss | Test acc (%) |
> | Inverse Time Decay | 1.5986 | 79.22 | 2.6162 | 85.16 | 2.2722 | 84.94 |
> | Step Decay         | 1.5684 | 81.07 | 2.5572 | 88.57 | 2.2535 | 85.16 |
> | Exponential Moving Average | 1.3534 | 77.83 | 1.6638 | 90.04| 1.9147 | 88.27 |
> | Cosine Decay       | 1.4181 | 84.58 | 1.9400 | 90.25 | 2.0328 | 87.72 |
> | Eigencurve         | **1.3474** | **85.72** | **1.3494** | **90.33** | **1.8764** | **88.51** |
>
> Along with the 100-epoch result,
>
> || ResNet-18 || GoogLeNet || VGG16 ||
> | -- | -- | -- | -- | -- | -- | -- |
> | Schedule |  Training loss | Test acc (%) | Training loss | Test acc (%) | Training loss | Test acc (%) |
> | Inverse Time Decay | 0.7269 | 90.97 | 0.6821 | 91.86 | 0.6085 | 89.82 |
> | Exponential Moving Average | 0.3012 | 89.62 | 0.3371 | 93.24 | 0.4860 | 92.49 |
> | Step Decay         | 0.2181 | 93.39 | 0.1669 | 94.07 | 0.2334 | 92.81 |
> | Cosine Decay       | 0.1728 | 93.94 | 0.1215 | 94.33 | 0.2016 | **93.21** |
> | Eigencurve         | **0.1360** | **93.99** | **0.1209** | **95.03** | **0.1758** | 93.13 |
>
> We train all models on CIFAR-10, with same settings in Section 4.2. For exponential moving average, we search the constant learning rate in range { 1.0, 0.6, 0.3, 0.2, 0.1 } and decay in { 0.9, 0.95, 0.99, 0.995, 0.999 }. We will release all the code if this paper got accepted.
>
> **Q4: The 10-epoch results are a bit artificial. With only 10 epochs, it makes little sense to decay the learning rates twice for the step-decay schedule...**
>
> **A4:** We presents this result as a proof of concept that Eigencurve can indeed outperform cosine decay when their learning rate curves differ a lot, since under normal settings they share extremely similar shapes.
>
> Also, there has been applications such as finetuning [1] or Neural Architecture Search [2], which only requires a small number of epochs. So Eigencurve does have adanvtanges under those scenarios.

---

> > ### Author Response · Authors · 2021-11-17
> > **Response to reviewer vptL (part 2/2)**
> >
> > **Q5: It seems that the authors used different parameters (e.g. \beta) for the 10-epoch setting. Although the authors argue that \beta is not a hyperparameter, ...**
> >
> > **A5:** Actually we just picked some random value near 1 without too much thought and certainly did not tune this value carefully. The value is chosen to be near 1 since we want to make the learning rate curve smoother.
> >
> > As a proof, we have plotted the learning rate curve of beta in { 1.0005, 1.00005, 1.000005, 1.0000005, 1.00000005 }, the figure is available in [Link to the Figure](https://github.com/opensource12345678/why_cosine_works/blob/main/figures/beta-choices/different-beta.png), along with separated curves under [Link to the Folder](https://github.com/opensource12345678/why_cosine_works/blob/main/figures/beta-choices/). One can observe that all of them share a similar learning rate curve, especially when beta's are close to 1.
> >
> > **Q6: ...Overall, the improvement of using the proposed Eigencurve is marginal on image classification tasks..."**
> >
> > **A6:** To further demonstrate Eigencurve's potential of discovering good schedulers, we add an extra experiment in language modeling task with LSTM + Penn Treebank for Elastic Step Decay. Results support our claims as well. We follow the setting of [3], except the number of epochs, where we set the number of epochs to 30 instead of 55 for large-regularized LSTM since we observe the model starts to overfit after 30 epochs. Please refer to the following table.
> >
> > | Schedule |  Validation perplexity | Test perplexity |
> > | ----- | ----- | ----- |
> > | Inverse time decay |   115.9   |   113.6   |
> > | Step Decay         |   82.3    |   79.1    |
> > | Baseline in [3]    |   82.2    |   78.4    |
> > | Cosine Decay       |   82.0    |   78.2    |
> > | Elastic Step Decay | **80.9**  | **77.2**  |
> >
> > Here all scheduler (except baseline) goes through hyperparameter grid search and are selected with best validation perplexity, especially for step decay since its common version is unknown in language modeling. The number of intervals are searched in { 3, 4, 5, [logT], [logT] + 1 } with decay factor searched in { 2, 5, 10 }. Other details will be added in the revised version of our paper.
> >
> > **Q7: The authors could save a lot of space by reorganizing all the figures in Figure 4...**
> >
> > **A7:** Thanks for this suggestion! We will certain reorganize our paper accordingly in the revised version.
> >
> > ## Reference
> > [1] Jacob Devlin, et al. "BERT: Pre-training of Deep Bidirectional Transformers for Language Understanding". arXiv preprint arXiv:1810.04805 (2018).
> >
> > [2] Barret Zoph, et al. "Learning transferable architectures for scalable image recognition". arXiv preprint arXiv:1707.07012 (2017).
> >
> > [3] Wojciech Zaremba, Ilya Sutskever, and Oriol Vinyals. "Recurrent neural network regularization". arXiv preprint arXiv:1409.2329 (2014).

---

> > > ### Comment · Reviewer_vptL · 2021-11-28
> > > **Thank you for the reply**
> > >
> > > I thank the authors for carefully addressing my concerns and for providing some useful clarifications. I have appreciated the effort in improving the manuscript, which is now more solid. I'm happy to increase my score to 6.

---

> > > > ### Author Response · Authors · 2021-11-29
> > > > **Thanks for the positive feedback!**
> > > >
> > > > Thanks very much for the positive feedback! We really appreciate it.

---

### Official Review · Reviewer_7X3r · 2021-11-02

**Correctness:** 4
**Technical Novelty And Significance:** 4
**Empirical Novelty And Significance:** 1
**Recommendation:** 6
**Confidence:** 4

**Main Review:**

## Writing

The quality of the writing is generally good.
Some sections have grammatical issues (see "Minor Comments" for a few examples), but these could be easily corrected by carefully proof-reading the draft.

## Theory

**Correctness**: I checked all of the theoretical derivations in the Appendix and believe them to be sound.
Some of the proofs are fairly dense and written with complex notation, so it possible that I missed a mistake.
With that said, I commend the authors on their attention to detail in the appendices.

**Novelty**: I believe the theoretical contributions to be novel.
While nice, the improvement in convergence rate of step-decay from a $\\log(T)$ to $\\log(\\kappa)$ factor is does not seem like a major contribution.
For instance, depending on $\\kappa$ and $T$, it is still possible for step-decay to have a faster theoretical rate of convergence than Eigencurve.

Eigencurve does attain the idea minimax complexity for SGD on convex quadratics when the "power power law" condition holds.
However, it is not clear that this means Eigencurve is optimal in a rigorous sense.
Is Eigencurve minimax optimal for the restricted class of quadratic problems satisfying the power power law?
This restricted class may be "easier" than the set of all quadratic functions due to clustering of the eigenvalues.
It seems to me that a lower bound for this class is necessary for the claim that Eigencurve is optimal to be accurate, since it is clearly not minimax optimal for the full class of quadratics.

Eigencurve also requires knowledge of the complete eigenvalue spectrum.
In general, computing this spectrum is as computationally intensive as minimizing the quadratic objective in the first place.
It isn't clear from the theoretical analysis if Eigencurve is robust to miss-estimation of the eigenvalues.
For tractability, extensions to approximate estimation of the spectrum or perhaps partial (e.g. only the top $n$ eigenvalues) spectra are attractive, but maybe hard to prove.
Given this issue, I think many practitioners will be content with step-decay.

Finally, it is not at all clear to me how to apply Eigencurve to non-quadratic optimization problems.
The discussion in the Appendix notes that, for non-convex optimization, an approximation of the Hessian at an approximate first-order critical point is used to compute the eigenvalue spectrum.
Why does this choice make sense?
Firstly, the FO point may be a saddle and the Hessian will be indefinite.
In this case, why is using the magnitude of the spectrum justified?
Secondly, why should the Hessian at a FO point be a good descriptor of the global function curvature?
The eigenvalues around a stationary point can be arbitrarily scaled for non-convex, non-smooth, non-Lipschitz models like neural networks [1].
I think these choices must be justified more carefully than they are in the submission.


## Experiments

The experimental evaluation in the submission is not ready for publication.

The first issue is that nearly all figures in the paper are illegible
For example, the labels for Figures 3, 4, and 5 are barely visible when reading the paper at a normal zoom-level.
Moreover, the columns and rows of Figure 4 are not labeled, meaning that the figure is still difficult to interpret when zoomed-in.
Note that the additional experiments in the appendix suffer from these same problems.
I don't think that figures need to be perfect at submission time, but they must be legible.

Another major issue of the evaluation is that there do not appear to be any repeats over random seed.
I searched Appendix B.1 where the experimental experimental methodology is given and did not find any mention of repetitions.
Moreover, distribution is not reported in any figure or table in the submission.
When evaluating stochastic optimization methods, it is absolutely essential that multiple repetitions be performed and distribution information included with the experimental results.
This is necessary for readers to judge whether or not results are a meaningful trend or a low-probability outcome.
For example, table 2 is just not meaningful without standard deviations or inter-quartile range.

Finally, there is a problem of fairness in the comparison to step-decay.
The authors report in Appendix B.1 that they use the "common settings" for step-decay on CIFAR-10/100.
No reference is provided to justify that the given settings are standard.
Moreover, the authors report that estimating the eigenvalue spectrum requires 1-2 days, which is an order of magnitude more time than required to train a single model.
This gives Eigencurve a huge advantage in computational budget over step-decay.
To be fair, I think a similar (or least substantial) attempt must be made to tune the parameters of step-decay.


## Minor Comments

- Remark 1: "coordinately" -> "coordinate-wise".
- Line after Eq. 1.7: In fact, no real justification or explanation of this assumption is given in Appendix G.5. The only comment made there is that previous work also make the same assumption.
- (Page 3) "Moreover, in practice, $\\kappa$ can be very big for large neural networks": (i) this remark requires some justification, (ii) to be picky, $\\kappa$ is not defined for neural networks because they are non-convex and typically non-smooth.
- Proposition 1: The proof of proposition 1 is not given in Lemma 15 as stated in the text. Instead, it is given in Appendix E on page 22.
- Page 12: what is $\\beta$? I don't see a description of this parameter in Eq. 3.3. Edit: I see that this is defined in Appendix B.1 *after* it is first used.
- Figure 6: It looks like the Largest eigenvalue is not within the bounds of the left-hand plot (log-scale).
- Appendix A.1: what is the batch size for this experiment? I don't see it reported anywhere.
- Page 20: I don't understand why the GPU memory limit prevents you from using multiple batches to estimate the Hessian spectrum. Isn't that the point of using batching in the first place?
- Page 20: There is no guarantee that the optimizer converges to a local minimum. It may converge to a saddle-point, resulting in the negative eigenvalues.
- Appendix B.4: This is not correct. The README.md of the indicated repository explicitly states "the CIFAR-10 are not my property. If used in a paper, you'll need to cite the reference paper, as indicated in the official website."
- Appendix C: Again, referencing a GitHub repository for code using ImageNet doesn't mean that the ImageNet dataset is distributed under the same license  as that repo.
    Licensing rules for ImageNet are given here: https://www.image-net.org/download.
- Appendix G.1: Use the restatable environment from thmtools (https://ctan.org/pkg/thmtools?lang=en) to avoid creating a new Lemma (i.e. Lemma 15) when proving Lemma 1.

## References:

[1] Dinh, Laurent, et al. "Sharp minima can generalize for deep nets." International Conference on Machine Learning. PMLR, 2017.

**Summary Of The Paper:**

# === Update === #

I appreciate the thorough response to my review provided by the authors. I have looked through the newest version of the submission and find it to be much improved. The inclusion of repeats for the train loss / test accuracy comparison is great to see and the updated figures are mostly fixed. The experiments now make a meaningful argument for Eigencurve as a (non)-convex optimization method.

I have increased my score to 6 to reflect the improvements and the author response.

# ====== #

This submission proposes a new step-size schedule for stochastic gradient descent which leverages the eigenvalue spectrum of Hessian to speed-up convergence.
The approach, dubbed Eigencurve, is shown to achieve the minimax optimal convergence rate for stochastic gradient descent on quadratic functions under an additional condition that the eigenspectrum of the Hessian decays according to a power law.
When this decay condition is not satisfied, Eigencurve still improves upon the popular step-decay schedule; in this case, it is sub-optimal by a factor of $\\log(\\kappa)$ rather than the $\\log(T)$ factor of step-decay.
A lower bound is also provided verifying that the $\\log(T)$ sub-optimality of step-decay is tight and cannot be improved for quadratics.
The submission concludes with an empirical investigation of Eigencurve for (non-convex) optimization of several popular neural network architectures on the CIFAR-10 and ImageNet datasets.

**Summary Of The Review:**


This is an interesting submission with a very novel (at least, to my knowledge) approach to choosing the step-size for SGD.
Effective step-size selection for stochastic optimizers is a challenging problem with potential for great impact on current machine learning practice.
As such, I think this work is a good fit for ICLR.
Theoretically, the submission is sound and I enjoyed reviewing the theoretical arguments (see "Theory" below).
However, empirically, the work is weak and, in my opinion, essentially unfinished.
I found the decision to focus the experiments on heuristic extensions of Eigencurve to non-convex optimization to be strange, given the theory does not even apply to general convex functions.
The majority of figures are completely unreadable and all results are reported without distribution information; it is to judge the significance of the experiments as a result.
See "Experiments" for more details.

Although interesting, I recommend that the submission be rejected given the unfinished state of the empirical evaluation.

---

> ### Author Response · Authors · 2021-11-21
> **Response to reviewer 7X3r (part 1/4)**
>
> First, we would like to offer our sinere thanks for the time and effort this reviewer has put into, given all the comprehensive and detailed comments this reviewer has written. We would also like to appreciate for all the constructive suggenstions and recognition on our contribution, which certainly cheers us up and helps us improve our paper concretely.
>
> **Overall:** We have revised the experimental parts in our main paper according to the reviewer's suggestion, including reorganizing Figures, adding multi-trial experiments and hyperparameter search for step decay schedulers. The revised version is available now. Those results are also provided and explained in following responses.
>
> Besides those experiments, more additional results are available in the revised version of our paper, such as language modeling for Elastic Step Decay. One may also refer to our responses to other reviewers. Due to time constraints, multi-seed experiments for results in the Appendix and additional results in this rebuttal cycle will be provided in the camera-ready version if this paper got accepted.
>
> Here we will address all the raised questions one by one.
>
> ## Theory
>
> **Q1: The improvement in convergence rate of step decay from a $\log(T)$ to $\log(\kappa)$...**
>
> **A1:** In worst case, yes, Eigencurve will indeed have a same convergence rate as step decay [1], but there are still several points worth noticing.
>
> First, when $T \le \kappa$, the variance term will no longer be the dominated term, hence comparing the resultant $\log(T)$ and $\log(\kappa)$ will be meaningless. In that case, the bias terms have to be taken into account, which are large constants for both step decay and Eigencurve. Therefore theoretically it is more reasonable to focus on the case of $T > \kappa$.
>
> Second, $\log(\kappa)$ is only the worst convergence rate for Eigencurve. The extra term $(\sum_{i=0}^{I_{\max}-1} \sqrt{s_i})^2$ has a size dependent on the given Hessian spectrum, and will be a constant if the Hessian spectrum is highly skewed, which is quite common for neural networks near their minimizers or second-order stationary points. In those cases, Eigencurve will be much better than step decay.
>
> Third, it is possible to obtain the same convergence rate by the mathematical trick of ignoring very small eigenvalues. Let us consider only those eigenvalues $\lambda_j > L/(T+1)$ and truncating other small eigenvalues, then our variance bound $O(\log(\kappa)) = O(\log(T))$ will hold. As for those eigenvalues $\lambda_j \le L/(T+1)$, each of them will create a variance no greater than $\sigma^2 \cdot \lambda_j^2 \sum_{k=0}^T \eta_k^2 \le \sigma^2 / (T+1)$ (Lemma 5) since $\eta_k \le 1/L$ (Eqn. 3.3). Combine those two parts together and assuming that the first part occupies $p \cdot d$ eigenvalues, we have the variance being $O((p \log(T) + (1-p)) \cdot d \sigma^2 / T) = O(\log T \cdot d \sigma^2 / T)$. Of course, for large $\kappa$, bias will become a bottleneck instead, which is the same for step decay [1], as they use the same trick in Theorem 13. We just want to demonstrate that it is mathematically possible to convert the variance bound from $O(\log(\kappa))$ to $O(\log(T))$. Furthermore, our proof techniques can be extended to smooth cases with the same trick. One just have to replace the bias reduction lemma with Lemma 10 in [1]. In that case semi-definite Hessians can be handled by Eigencurve as well.
>
> **Q2: Eigencurve does attain the idea minimax complexity for SGD...Is Eigencurve minimax optimal for the restricted class of quadratic problems satisfying the power power law?...**
>
> **A2:** This is indeed a good question. As Eigencurve has elimianted the $O(\log T)$ term from step decay, it is still unclear whether the introduced extra constant $C(\alpha)$ in Corollary 2 can be further improved to a constant independent from $\alpha$. We conjectured that it might be impossible, since there does exist hard problem instances with uniform densities across eigenvalue groups, which corresponds to cases when $\alpha \rightarrow 1$. However, the proof of lower bound is not that easy, so we leave it to future work.

---

> > ### Author Response · Authors · 2021-11-21
> > **Response to reviewer 7X3r (part 2/4)**
> >
> > **Q3: Eigencurve also requires knowledge of the complete eigenvalue spectrum...**
> >
> > **A3:** Theoretically, yes, there is currently no rigorous guarantee in our paper showing that Eigencurve is robust to miss-estimation of eigenvalues. However, it is possible to prove that. First, since the interval length is $\Delta_i = \frac{\sqrt{s_i}}{Z} \cdot T$ where $Z = \sum_{i=0}^{I_{\max}-1} \sqrt{s_i}$ (Eqn 3.5), any Hessian spectrums with similar { $s_i$ } will yield similar Eigencurve scheduler schemes. As the variance is dominated by the interval length corresponds to each group of eigenvalues, we conjecture that tranferring Eigencurve for { $s_i$ } to a new distribution { $s'_i$ } will yield an extra term around $O(\text{poly}(\max_i \frac{s_i}{s'_i}, \max \frac{s'_i}{s_i}))$. Second, it is possible to extend our proof tenchniques to smooth cases by ignoring very small eigenvalues, so there is indeed a way to have some theoretical guarantees of the robustness.
> >
> > Empirically, we have presented evidence in Appendix A.3, showing that Eigencurve can be robust to Hessian spectrums for models in that same task. Furthermore, we here present an additional empirical result for Eigencurve, showing Eigencurve's learning rate curve under different choices of $\kappa$, with the originally estimated eigenvalues linearly mapped to $[L / \kappa, L]$. Please refer to [Link to the folder](https://github.com/opensource12345678/why_cosine_works/tree/main/figures/kappa-choices). Results show that when $\kappa$ is above a threshold, the learning rate curve of Eigencurve "converges" and doesn't change much, which again reflects Eigencurve's robustness in practice.
> >
> > Regarding the expensive cost of Hessian spectrum estimation, we recommend two ways to use Eigencurve for practitioners. First, for models or tasks that differ a lot from past settings, one can run Hessian spectrum estimation once to obtain some insights about the new setting. This is an onetime effort and this Hessian spectrum can be reused for similar models or tasks under the same settings, just like how we use cosine decay for CV tasks. Second, one may consider directly using Eigencurve's derivatives, such as Elastic Step Decay. Accordingly to results in Appendix A.4, it has a very simple form and surpasses step decay as well.
> >
> > **Q4: Finally, it is not at all clear to me how to apply Eigencurve to non-quadratic optimization problems...**
> >
> > **A4:** This is indeed a good question. However, since our main focus in this paper are quadratic objectives, we would like to leave the theoretical part of non-convex objectives to future works.
> >
> > ## Experiments
> >
> > **Q5: The first issue is that nearly all figures in the paper are illegible...**
> >
> > **A5:** Definitely. We have reorganized all the figures. The revised version will be avaiable in 1-2 days.
> >
> > **Q6: Another major issue of the evaluation is that there do not appear to be any repeats over random seed...**
> >
> > **A6:** Thanks for pointing this out. We have conducted multi-seed experiments for results in the main paper. Please refer to the result in Q7 & A7. The paper has been revised accordingly. Due to time constraints, multi-seed experiments for results in the Appendix and additional results in this rebuttal cycle will be provided in the camera-ready version if this paper got accepted.

---

> > > ### Author Response · Authors · 2021-11-21
> > > **Response to reviewer 7X3r (part 3/4)**
> > >
> > > **Q7: Finally, there is a problem of fairness in the comparison to step-decay...**
> > >
> > > **A7:** The is certainly a reasonable suggestion. We have added hyperparameter search for step decay in the revised paper, with interval numbers in { 3, 4, 5, [$\log T$], [$\log T$] + 1 } and decay factor in { 2, 5, 10 }, along with the same search range for $\eta_0$ as other schedulers. We also display results of the step decay scheduler proposed in [1], which has a fixed interval number $[ \log T ]$ and decay factor $2$.
> > >
> > > We run all schedulers with 5 different random seeds and report their means and standard deviations. The CIFAR-10 10-epoch result is as follows,
> > >
> > > || ResNet-18 || GoogLeNet || VGG16 ||
> > > | -- | -- | -- | -- | -- | -- | -- |
> > > | Schedule |  Training loss | Test acc (%) | Training loss | Test acc (%) | Training loss | Test acc (%) |
> > > | Inverse Time Decay | $1.58\pm0.02$ | $79.45\pm1.00$ | $2.61\pm0.00$ | $86.54\pm0.94$ | $2.26\pm0.00$ | $84.47\pm0.74$ |
> > > | Step Decay in [1]  | $1.82\pm0.04$ | $73.77\pm1.48$ | $2.59\pm0.02$ | $87.04\pm0.48$ | $2.42\pm0.45$ | $82.98\pm0.27$ |
> > > | Step Decay         | $1.52\pm0.02$ | $81.99\pm0.35$ | $1.93\pm0.03$ | $88.32\pm1.32$ | $2.14\pm0.42$ | $86.79\pm0.36$ |
> > > | Cosine Decay       | $1.42\pm0.01$ | $84.23\pm0.07$ | $1.94\pm0.00$ | $90.56\pm0.31$ | $2.03\pm0.00$ | $87.99\pm0.13$ |
> > > | Eigencurve         | $\textbf{1.36}\pm\textbf{0.01}$ | $\textbf{85.62}\pm\textbf{0.28}$ | $\textbf{1.33}\pm\textbf{0.00}$ | $\textbf{90.65}\pm\textbf{0.15}$ | $\textbf{1.87}\pm\textbf{0.00}$ | $\textbf{88.73}\pm\textbf{0.11}$ |
> > >
> > > Along with the corresponding 100-epoch result,
> > >
> > > || ResNet-18 || GoogLeNet || VGG16 ||
> > > | -- | -- | -- | -- | -- | -- | -- |
> > > | Schedule |  Training loss | Test acc (%) | Training loss | Test acc (%) | Training loss | Test acc (%) |
> > > | Inverse Time Decay | $0.73\pm0.00$ | $90.82\pm0.43$ | $0.62\pm0.02$ | $92.05\pm0.69$ | $1.32\pm0.62$ | \*$76.24\pm13.77$ |
> > > | Step Decay in [1]  | $0.26\pm0.01$ | $91.39\pm1.03$ | $0.28\pm0.00$ | $92.83\pm0.15$ | $0.59\pm0.00$  | $91.37\pm0.20$ |
> > > | Step Decay         | $0.17\pm0.00$ | $93.97\pm0.21$ | $0.13\pm0.00$ | $94.18\pm0.18$ |  $0.20\pm0.00$ | \*$92.36\pm0.46$ |
> > > | Cosine Decay       | $0.17\pm0.00$ | $94.04\pm0.21$ | $0.12\pm0.00$ | $94.62\pm0.11$ | $0.20\pm0.00$ | $\textbf{93.17}\pm\textbf{0.05}$ |
> > > | Eigencurve         | $\textbf{0.14}\pm\textbf{0.00}$ | $\textbf{94.05}\pm\textbf{0.18}$ | $\textbf{0.12}\pm\textbf{0.00}$ | $\textbf{94.75}\pm\textbf{0.15}$ | $\textbf{0.18}\pm\textbf{0.00}$ | $92.88\pm0.24$ |
> > >
> > > \*: Loss explodes for one seed among all 5 seeds.
> > >
> > > We can observe that Eigencurve still consistently outperforms step decay, and in most cases by a large margin. When we compare the performance of step decay in [1] and Eigencurve, this gap widens and becomes quite significant.
> > >
> > > ## Minor Comments
> > >
> > > **Q8.1: Remark 1: "coordinately" -> "coordinate-wise".**
> > >
> > > **A8.1:** Thanks! we will fix this typo.
> > >
> > > **Q8.2: Line after Eq. 1.7: In fact, no real justification or explanation of this assumption...**
> > >
> > > **A8.2:** Yes. Our intention in the paper is to to make it clear that this is a normal assumption. It is in fact an intermediate assumption entailed by the common assumptions of strongly convex least square regressions. Hence our theoretical results hold under that setting as well.
> > >
> > > **Q8.3: (Page 3)...(i) this remark requires some justification, (ii) to be picky, $\kappa$ is not defined for neural networks because they are non-convex and typically non-smooth.**
> > >
> > > **A8.3:** (i) Please refer to the row for Inverse Time Decay in Table 4, Appendix A.2, where the computed kappa value for different models/tasks are provided.
> > >
> > > (ii) Indeed. There is still a theoretical gap between our theory and the real application. However, as the main focus of our paper are quadratic objectives, we would like to view it as a starting point and use the estimated Hessian just to check if Eigencurve works in practice. Empirically, we follow this assumptions of quadratic objectives and estimate the Hessian spectrum after sufficiently training the neural network, where the details are presented in Appendix B.2.
> > >
> > > As our theory mainly focus on quadratic cases, our experiments on non-convex and non-smooth objectives majorly demonstrate the practical value and extensibility of Eigencurve, showing that it works not only in theory, but also in practice.
> > >
> > > **Q8.4: Proposition 1: The proof of proposition 1 is not given in Lemma 15...**
> > >
> > > **A8.4:** Lemma 15 gives the proof of the equality in Proposition 1, not the whole Proposition 1. We have improved the writing to make it less confusing.
> > >
> > > **Q8.5: Page 12: what is $\beta$?...**
> > >
> > > **A8.5:** Thanks for pointing that out! We will remove that in Appendix A.1 to make it less confusing.

---

> > > > ### Author Response · Authors · 2021-11-21
> > > > **Response to reviewer 7X3r (part 4/4)**
> > > >
> > > > **Q8.6: Figure 6: It looks like the Largest eigenvalue is not within the bounds of the left-hand plot...**
> > > >
> > > > **A8.6:** Thanks for noticing that! The largest eigenvalue is 12.5477734895, which is not presented in the right-hand plots. We have fixed it in the revised version of our paper.
> > > >
> > > > **Q8.7: Appendix A.1: what is the batch size for this experiment?...**
> > > >
> > > > **A8.7:** The batch size is 1. We will emphasize it to make it clearer.
> > > >
> > > > **Q8.8: Page 20: I don't understand why the GPU memory limit prevents you from using multiple batches...**
> > > >
> > > > **A8.8:** You are right. As shown in Appendix B.2, GPU memory only limits the largest batch size, not the number of batches. In fact, in our paper, we never mention the reason of choosing single batch for estimation. The phrasing here may be a bit misleading and we will improve it in the revised version. As the question is raised, we would like to present the reasons here.
> > > >
> > > > The main reason of not using multiple batches is actually time cost, not space. As the high-precision estimation process is already time-consuming, adding more batches will be computationally unaffordable. We hope that more efficient tools for estimating Hessian spectrums can be invented in the future.
> > > >
> > > > On top of that, single-batch estimation is already good enough. We have observed that estimation with a single batch yields similar results as full-batch versions presented in PyHessian's paper [5], so we decide to use single batch for Hessian spectrum estimation in our experiments.
> > > >
> > > > **Q8.9: Page 20: There is no guarantee that the optimizer converges to a local minimum...**
> > > >
> > > > **A8.9:** This is indeed true. But as we mentioned in Q4 & A4, our main focus in this paper are quadratic objectives and we view it as a startinging point. For possible extensions to non-convex objectives, we think that the theoretical backups provided in [2][3][4] may be useful.
> > > >
> > > > Since our current theory majorly focus on the quadratic case, we leave the theoretical part of non-quadratic objectives to furture works.
> > > >
> > > > **Q8.10: Appendix B.4: This is not correct. The README.md of the indicated repository explicitly states...**
> > > >
> > > > **A8.10:** Thanks for pointing that out! However, it seems that the author of CIFAR-10 does not provide the license explicitly on the [CIFAR-10 official website](https://www.cs.toronto.edu/~kriz/cifar.html), so we decide to remove this part of the description.
> > > >
> > > > **Q8.11: Appendix C: Again, referencing a GitHub repository for code using ImageNet...**
> > > >
> > > > **A8.11:** Thanks for providing this useful information! We have revised our paper correspondingly to correct this issue.
> > > >
> > > > **Q8.12: Appendix G.1: Use the restatable environment from thmtools...**
> > > >
> > > > **Q8.12:** Thanks for this helpful suggestion! We have revised our paper accordingly.
> > > >
> > > > ## Summary
> > > > **Q.9: ...I found the decision to focus the experiments on heuristic extensions of Eigencurve to non-convex optimization to be strange, given the theory does not even apply to general convex functions...**
> > > >
> > > > **A.9:** Our intention is to show that Eigencurve can have real applications in practice, where it can help the deep learning community discover new types of schedulers and lead to concrete improvements for different tasks with good theoretical guarantees. This is the most exciting part of Eigencurve.
> > > >
> > > > ## Reference
> > > >
> > > > [1] Ge Rong, et al. "The step decay schedule: A near optimal, geometrically decaying learning rate procedure for least squares." arXiv preprint arXiv:1904.12838 (2019).
> > > >
> > > > [2] Cong Fang, et al. "Sharp Analysis for Nonconvex SGD Escaping from Saddle Points". arXiv preprint arXiv:1902.00247 (2019).
> > > >
> > > > [3] Jason D. Lee, et al. "Gradient Descent only Converges to Minimizers". COLT 2016.
> > > >
> > > > [4] Sanjeev Arora, et al. "On exact computation with an infinitely wide neural net". NeurIPS 2019.
> > > >
> > > > [5] Zhewei Yao, et al. "PyHessian: Neural Networks Through the Lens of the Hessian". arXiv preprint arXiv:1912.07145 (2019).

---

> > > > > ### Comment · Reviewer_7X3r · 2021-11-28
> > > > > **Author Response**
> > > > >
> > > > > Thank you for the extremely thorough response to my review.
> > > > >
> > > > > I have looked at the revised submission and find it to be significantly improved. Well done! Some specific comments follow:
> > > > > - The updated figures look much better, although an even larger font could be used for the labels, ticks, etc.
> > > > >
> > > > > - The repeats over seed in Table 2 are excellent. Now we can see that Eigencurve has better performance than cosine decay for ResNets when Epochs = 100, but about the same performance on GoogLeNet.
> > > > >
> > > > > - **Regarding A8.8**: Thanks for clarifying this. Of course, multiple batches will extend the time to estimate the spectrum and may be too slow --- this isn't an issue for me.
> > > > >
> > > > > - **Regarding A9**: Yes, I understand the intention behind these experiments and behind the elastic step-decay schedule. However, I think this focus short-sells the contributions you have made to quadratic minimization, which is an interesting problem in its own right. Neural networks are a "hot topic", but we, as a community, don't have to cast every optimization method as a contribution to neural network training. Personally, I think a principled extension to approximate eigenvalue estimation, etc, is more interesting than yet another heuristic for non-convex optimization.
> > > > >
> > > > > - **Regarding A3**: I suspect that the reason Eigencurve is resilient to the use of different eigenspectrums for the same task is the choice of $\\beta \\approx 1$, which leads to a very smooth schedule. Choosing $\\beta = 2$ may lead to more instability. Maybe an ablation study could be performed later, but it's not critical at the moment.
> > > > >
> > > > > Thanks again for the obvious effort that has gone into the author response. I'm glad to see the submission has improved as a result of the review process. I will update my review to reflect the updates.

---

> > > > > > ### Author Response · Authors · 2021-11-29
> > > > > > **Thanks for the positive feedback**
> > > > > >
> > > > > > Thanks very much for the positive feedback! That means a great deal to us. We have certainly learned a lot through the rebuttal process, given the constructive advices provided by the reviewer.
> > > > > >
> > > > > > Regarding the remaining minor issues, we would like to address them as follows.
> > > > > >
> > > > > > **Q: The updated figures look much better, although an even larger font could be used for the labels, ticks, etc.**
> > > > > >
> > > > > > **A:** No problem. We will enlarge them accordingly in the camera-ready version if this paper got accepted.
> > > > > >
> > > > > > **Q: Regarding A9...**
> > > > > >
> > > > > > **A:** Thanks very much for this instructive comment! We will definitely try to follow this advice in our future researches.
> > > > > >
> > > > > > **Q: Regarding A3:...**
> > > > > >
> > > > > > **A:** This is indeed a good question. We will try to add ablation study to check it out and report the result if time and resource allows.

---

### Decision · Program_Chairs · 2022-01-20

**Decision:**

Accept (Poster)

**Comment:**

The paper proposes a method to learning rate scheduling that uses information form the eigenvalues of the Hessian. It shows that this scheduler obtains the minimax optimal rate on the noisy quadratic problem; and, empirically, this scheduler demonstrates faster convergence on CIFAR-10 and ImageNet, when the number of epochs is small.  Using Hessian information in direct and indirect ways is of interest to the community, and the paper does a nice job illustrating that in a context of interest.